# MFSD1 with its accessory subunit GLMP functions as a general dipeptide uniporter in lysosomes

Katharina Esther Julia Jungnickel [1,2,12], Océane Guelle [3,12], Miharu Iguchi[4,5,6], Wentao Dong [4,5,6], Vadim Kotov[1,2], Florian Gabriel[1,2], Cécile Debacker[3], Julien Dairou[7], Isabelle McCort-Tranchepain[7], Nouf N. Laqtom[4,5,6], Sze Ham Chan[8], Akika Ejima[9], Kenji Sato[9], David Massa López[10], Paul Saftig [10], Ahmad Reza Mehdipour[11], Monther Abu-Remaileh[4,5,6], Bruno Gasnier [3,13] ✉, Christian Löw [1,2,13] ✉ & Markus Damme [10,13] ✉

The lysosomal degradation of macromolecules produces diverse small metabolites exported by specific transporters for reuse in biosynthetic pathways. Here we deorphanized the major facilitator superfamily domain containing 1 (MFSD1) protein, which forms a tight complex with the glycosylated lysosomal membrane protein (GLMP) in the lysosomal membrane. Untargeted metabolomics analysis of MFSD1-deficient mouse lysosomes revealed an increase in cationic dipeptides. Purified MFSD1 selectively bound diverse dipeptides, while electrophysiological, isotope tracer and fluorescence-based studies in *Xenopus* oocytes and proteoliposomes showed that MFSD1–GLMP acts as a uniporter for cationic, neutral and anionic dipeptides. Cryoelectron microscopy structure of the dipeptide-bound MFSD1–GLMP complex in outward-open conformation characterized the heterodimer interface and, in combination with molecular dynamics simulations, provided a structural basis for its selectivity towards diverse dipeptides. Together, our data identify MFSD1 as a general lysosomal dipeptide uniporter, providing an alternative route to recycle lysosomal proteolysis products when lysosomal amino acid exporters are overloaded.

Lysosomes degrade various macromolecules, including extracellular and intracellular proteins internalized or sequestered by endocytosis, phagocytosis and autophagy[1,2]. Lysosomal proteolysis prevents the build-up of old or damaged proteins and protein aggregates under basal conditions and supplies recycled amino acids under starvation[3]. A set of ~15 relatively promiscuous lysosomal proteases mediates this hydrolysis, yielding short peptides and free amino acids, which are eventually exported from the lysosomal lumen to the cytoplasm by specific transport systems[4–9]. Lysosomes also play a critical role in intracellular nutrient sensing and the recruitment and activation of the mTOR complex at the outer lysosomal surface[3].

Recently, the export of amino acids from lysosomal proteolysis has received increasing attention[3]. Although several underlying transporters have been identified[7–11], many are still missing. Various regulatory mechanisms of lysosomal amino acid transport have been discovered[12–14], some transporters have been implicated in nutrient sensing[3,15–17] and transporter structures have been characterized[18–20]. In contrast, lysosomal peptide transporters have received less attention, although it has been known for decades that in lysosomes, specific peptides are not completely proteolytically degraded to single amino acids and that lysosomal peptide transporters must exist[21–24]. Two members of the proton-coupled oligopeptide transporter (POT)

family, PHT1/SLC15A4 and PHT2/SLC15A3, localize to endosomes and lysosomes[25,26]. They transport carnosine, muramyl dipeptide, tri-DAP, glycylsarcosine (Gly-Sar) by PHT1 and His–Leu by PHT2 (refs. 6,27). Both may transport histidine, though evidence varied greatly across cell lines[28–30]. However, due to their close relationship to the extensively studied bacterial and mammalian POT members[31–35], including PepT1/SLC15A1 and PepT2/SLC15A2, they are expected to transport a broad spectrum of dipeptides and tripeptides.

To help elucidate orphan lysosomal transporters, we recently investigated the major facilitator superfamily domain containing 1 (MFSD1) protein, which we and others identified by mass spectrometry in isolated lysosomes[36,37]. Members of the major facilitator superfamily (MFS) typically mediate the import and export of water-soluble molecules through a rocker-switch mechanism[38–41]. However, MFSD1 substrate(s) remain unknown[42]. MFSD1 is ubiquitously expressed in mouse tissues, where it localizes in lysosomes[43]. In contrast to most lysosomal transmembrane proteins, MFSD1 is not N-glycosylated[43]. However, it forms a heterodimeric complex with the glycosylated lysosomal membrane protein (GLMP)[43], an extensively N-glycosylated single-pass type I transmembrane protein. Without one subunit, the other is rapidly degraded, suggesting a chaperone function and protective effect towards lysosomal proteases[43]. The remaining MFSD1 is retained in the Golgi apparatus in GLMP-deficient cells, indicating an additional role of GLMP in transporting the complex from the Golgi apparatus to lysosomes[44].

In this Article, we used metabolomics, electrophysiology and fluorescence- or tracer-based uptake assays to elucidate the transport activity of MFSD1 and show that it acts as a dipeptide-specific uniporter with broad dipeptide promiscuity. We determined the structure of the MFSD1–GLMP complex by cryoelectron microscopy (cryo-EM). Together with molecular dynamic (MD) simulations, we obtained a detailed molecular picture of how lysosomal dipeptides are recognized and transported, providing a structural basis for its role as a general dipeptide transporter.

## Results

### Dipeptides accumulate in MFSD1-deficient lysosomes

To identify substrate(s) potentially transported by MFSD1, we enriched lysosomes from wild-type (WT) and *Mfsd1*-knockout mice (*Mfsd1*$^{tm1d/tm1d}$)[43] by differential centrifugation and a sucrose density gradient (Fig. 1a). This procedure yields fractions highly enriched for the lysosomal markers LAMP1 and cathepsin D, with little contamination from other organelles (Fig. 1b)[37]. These fractions were analysed by untargeted mass spectrometry-based metabolomics. Two metabolites significantly increased above the defined thresholds ($P \leq 0.05$, fold change $\geq 2$) and were tentatively identified as Arg–Pro (or Pro–Arg) and Pro–Lys dipeptides (Fig. 1c and Supplementary Table 1). The extracted ion chromatograms of the first metabolite, with $m/z$ (M + H) 272.1717, matched that of an Arg–Pro or Pro–Arg chemical standard (Fig. 1d). Tandem mass

spectrometry (MS/MS) analysis against spectral libraries confirmed the identity of both dipeptides (Extended Data Fig. 1a). Quantification of Pro–Lys and Arg–Pro and targeted analysis of additional dipeptides (Arg–hydroxyPro and anserine) revealed a pronounced increase in *Mfsd1*$^{tm1d/tm1d}$ lysosomes (Fig. 1e). Quantification of different dipeptides in different organ (liver, spleen and lung) lysates (Extended Data Fig. 1b) showed an increase of anserine, Arg–Pro, Pro–Arg and Arg–hydroxyPro in spleen but not other organs from *Mfsd1*$^{tm1d/tm1d}$ mice.

### Recombinant MFSD1 binds dipeptides

The metabolomics data prompted us to test whether MFSD1 is involved in lysosomal peptide transport. MFSD1 was transiently expressed in Expi293F cells and purified to homogeneity in dodecyl-β-D-maltopyranoside (DDM)/cholesterol hemisuccinate (CHS) detergent solution (Fig. 1f and Extended Data Fig. 1c). To screen for peptide binding, MFSD1 was subjected to thermal shift experiments using differential scanning fluorimetry (nanoDSF) (Fig. 1g–j). Upon interaction with a substrate, the protein is stabilized, resulting in an increased melting temperature ($T_m$). Initial nanoDSF experiments at a 5 mM ligand concentration showed stabilization of MFSD1 by Leu–Ala, Lys–Val and Pro–Arg but not Ala–Ala (Fig. 1g). We performed a larger nanoDSF screen covering 18 amino acids, 68 di- and tripeptides, two tetrapeptides, five sugars and seven drugs (Fig. 1h). The strongest stabilization was observed for neutral dipeptides (for example, Leu–Leu, changes in the melting temperature ($\Delta T_m$) of 14 °C) and dipeptides with at least one positively charged residue (for example, Pro–Arg, $\Delta T_m$ of 12.1 °C and His–Lys, $\Delta T_m$ of 12 °C). No, or only small, thermal shift changes were detected for any other compound classes, indicating that MFSD1 primarily binds dipeptides. Titration experiments with His–Ala, His–Lys, Leu–Ala, Lys–Val or Pro–Arg yielded dissociation constants ($K_D$) of 6.7 ± 0.55 mM, 765 ± 136 μM, 2.2 ± 0.42 mM, 4.3 ± 0.6 mM and 318 ± 66.7 μM (Fig. 1i,j and Extended Data Fig. 1d), respectively. These $K_D$ values are within the range of reported binding affinities of other MFS peptide transporters[35,45–48].

### Uptake of dipeptides by MFSD1 and GLMP

Next, we tested whether MFSD1 not only binds but also transports dipeptides using a whole-cell transport assay in *Xenopus* oocytes. In this approach, the lysosomal transporter is misrouted to the plasma membrane by mutating its lysosomal sorting motif(s), replacing the poorly tractable lysosomal export with whole-cell import. The transport reaction is started by adding the substrate in an acidic extracellular medium (mimicking lysosomal pH)[5,49]. Expression of an MFSD1 sorting mutant (MFSD1$_{L11A/L12A}$)[43] fused to emerald-green fluorescent protein (EmGFP) in oocytes showed limited localization to the plasma membrane, as determined by cell surface biotinylation. However, co-expression of a GLMP sorting mutant[43], GLMP$_{Y400A}$-mKate2, increased the surface level of MFSD1 by approximately tenfold (Fig. 2a). Fluorescence microscopy confirmed this effect and showed colocalization of the

**Fig. 1 | *Mfsd1*-knockout mice accumulate cationic dipeptides in liver lysosomes, and recombinant MFSD1 binds various dipeptides.**
**a**, A schematic representation of lysosome enrichment by ultracentrifugation and untargeted metabolomics. **b**, Immunoblot analysis of PNS, mitochondria and lysosome-enriched fractions and the final lysosome-enriched fraction from WT and *Mfsd1*-knockout mice for markers of various cellular compartments. ER, endoplasmic reticulum. **c**, Volcano plot of differential metabolites between liver lysosomes of WT and *Mfsd1*-knockout mice (two-sided one-way analysis of variance with Tukey's post hoc test, adjustment for multiple testing).
**d**, Extracted ion chromatogram (EIC) for the chemical standard Pro–Arg (yellow, 100 nM) and representative samples from WT (red) and *Mfsd1*-knockout mice (blue). Pro–Arg is detected as a peak eluting at a retention time (RT) of 8.44 min. **e**, Relative abundance of Pro–Lys, Arg–Pro and anserine between WT and *Mfsd1*-knockout mice. The abundance was normalized to the isotopically labelled arginine levels, which showed no differences between the two genotypes

in the untargeted metabolomic analysis (two-tailed unpaired *t*-tests). The data are means ± s.e.m. *N* = 5 for the animals/genotype (\**P* ≤ 0.05 and \*\*\**P* ≤ 0.001).
**f**, Coomassie-stained SDS–PAGE gel of purified MFSD1 with a Twin-Strep-tag that was transiently expressed in Expi293F cells and purified to homogeneity in DDM/CHS detergent solution. **g**, Unfolding traces of MFSD1 in the absence and presence of Ala–Ala, Pro–Arg, Leu–Ala and Lys–Val at a concentration of 5 mM. **h**, Thermal stability of MFSD1 in the presence of a compound library at a 5 mM final ligand concentration. The $\Delta T_m$ of MFSD1 are given as a difference to the melting temperature of apo MFSD1 ($T_{m(apo)}$). The data are means ± s.e.m. (*n* = 3 for the independent samples). **i,j**, Examples of $K_D$ measurements are based on changes in the thermal stability of MFSD1 in the presence of varying concentrations of the dipeptides His–Ala (red) (**i**) or Pro–Arg (blue) (**j**). The $K_D$ values were determined using Moltenprot[67]. **h**-**j**, Data are shown as mean ± s.d. The source numerical data and unprocessed blots are available in the source data.

EmGFP and Kate2 signals at or near the surface of co-injected oocytes (Fig. 2b). We, thus, used oocytes co-expressing MFSD1$_{L11A/L12A}$-EmGFP and GLMP$_{Y400A}$-mKate2 ('MFSD1–GLMP oocytes') for the transport assays.

The oocytes were recorded under two-electrode voltage clamp (TEVC) at −40 mV, and the dipeptides were applied at extracellular pH (pH$_{out}$) 5.0 to test them for electrogenic transport (Fig. 2c).

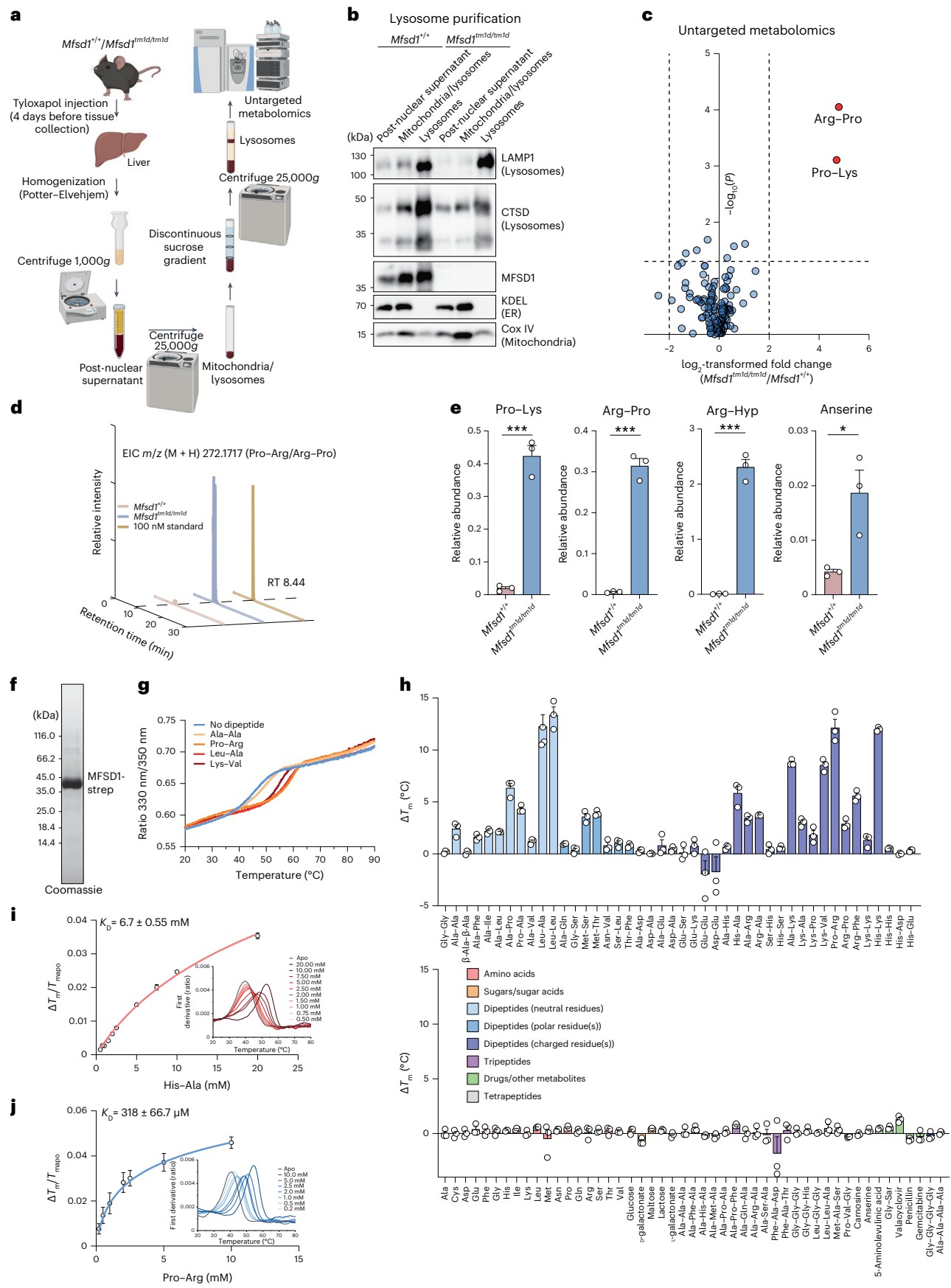

Lys–Ala evoked a robust inward current (−300 ± 50 nA) in MFSD1–GLMP oocytes but not in mock (non-injected) oocytes nor oocytes expressing only GLMP$_{Y400A}$-mKate2, while it evoked at best, a very low current (−9.2 ± 4.8 nA) in oocytes expressing only MFSD1$_{L11A/L12A}$-EmGFP. The Lys–Ala current was dose-dependent with a $K_M$ of 2.6 ± 0.4 mM ($n$ = 3) (Fig. 2d). It was approximately threefold stronger at pH$_{out}$ 5.0 than pH$_{out}$ 7.0 (Fig. 2e) but did not depend on Na$^+$ (Extended Data Fig. 2a). Single cationic amino acids (His, Lys or Arg) and the tripeptide Lys–Ala–Ala (10 mM) did not evoke any current in MFSD1–GLMP oocytes (Fig. 2f), in agreement with the nanoDSF data. Among dipeptides, several cationic compounds such as Ala–Lys, Arg–Ala, His–Ser, Arg–Pro and, to a lesser extent, Lys–Pro and Pro–Arg, evoked a robust current, whereas neutral dipeptides (Leu–Ala and Ala–Ala) and an anionic dipeptide (Glu–Ser) had no effect (Fig. 2g and Extended Data Fig. 2b). We performed competition experiments to test whether neutral or anionic dipeptides interact with MFSD1–GLMP in oocytes. Leu–Ala (20 mM) applied simultaneously with Lys–Ala (3 mM) abolished the Lys–Ala current (Extended Data Fig. 2c), while Ala–Ala (20 mM) and Glu–Ser (10 mM) inhibited it by 66 ± 3% ($n$ = 6) and 26 ± 3% ($n$ = 3), respectively (Extended Data Fig. 2d,e). We concluded that MFSD1–GLMP interacts with diverse dipeptides in the oocyte membrane and transports cationic dipeptides in an electrogenic manner.

As an alternative in vitro approach, the transport activity was characterized using purified WT MFSD1 (MFSD1$_{WT}$) (Fig. 3a). To monitor possible proton-coupling by MFSD1$_{WT}$ as observed for other lysosomal transporters[5,49,50], liposomes were loaded with the pH-sensitive dye pyranine[51]. A membrane potential of approximately −100 mV was applied using valinomycin (val) (Fig. 3b). We used liposomes devoid of MFSD1 ('empty liposomes') as negative controls. Time-dependent uptake assays in the presence of the dipeptide His–Ser highlight that only MFSD1-containing liposomes exhibit a decrease in fluorescence ($F_{norm}$). All other traces remained stable over a time period of 10 min (Fig. 3c). Since this method monitors the uptake of protons, we screened a similar set of dipeptides than in the oocyte assay (Fig. 3d) and determined the Michaelis–Menten kinetics for His–Ala and His–Ser. The $K_M$ values were 119.1 ± 59.3 µM and 24.4 ± 13.5 µM, respectively, with $v_{max}$ of −0.001731 ± 0.00045 $\Delta F_{norm}$ s$^{-1}$ and −0.001586 ± 0.00035 $\Delta F_{norm}$ s$^{-1}$, respectively (Fig. 3e). Intriguingly, uptake was exclusively observed for peptides containing at least one histidine residue, with Glu–Lys being the only exception (Fig. 3d). Although the liposome and oocyte activities shared common features (strong His-Ser signal and lack of response to neutral and anionic dipeptides), they diverged for a subset of cationic dipeptides, such as Lys–Ala, Ala–Lys, Lys–Val and L-anserine, which evoked a robust inward current in the oocyte assay, yet had no effect in the liposome assay.

## MFSD1 operates as a dipeptide uniporter

To clarify this discrepancy, we examined whether MFSD1 co-transports protons, as initially postulated, using combined TEVC and intracellular pH (pH$_{in}$) recording of MFSD1–GLMP oocytes (Fig. 4a). We used two approaches to check the sensitivity of the pH$_{in}$ microelectrode impaled in the oocyte. First, we co-expressed MFSD1–GLMP with the

lysosomal uniporter for cationic amino acids PQLC2 (sorting mutant PQLC2$_{L290A/L291A}$-enhancedGFP (EGFP)) to serve as a positive control[7,13]. Uptake of cationic histidine by PQLC2 induces intracellular acidification, reflecting the release of its side chain proton (pK$_a$ of 6.0) when the substrate faces the cytosol (pH of 7.2)[13]. As PQLC2 does not respond to Lys–Ala (Extended Data Fig. 3a), the MFSD1–GLMP and PQLC2 activities can be monitored independently. Sequential application of Lys–Ala and His to MFSD1–GLMP + PQLC2 oocytes showed that Lys–Ala uptake by MFSD1–GLMP does not evoke any intracellular acidification under conditions where the pH$_{in}$ microelectrode detects a slower flux of cationic histidine through PQLC2 (Fig. 4a and Extended Data Fig. 3b), ruling out an H$^+$ symport mechanism for MFSD1–GLMP (Fig. 4b). Second, we compared the responses of MFSD1–GLMP oocytes with Lys–Ala and His-containing dipeptides. Similar to His uptake by PQLC2, His-containing dipeptides should release their side chain proton within the oocyte if MFSD1–GLMP transports them in cationic form. Indeed, His–Ala and His–Ser but not Lys–Ala evoked an intracellular acidification in MFSD1–GLMP oocytes (Fig. 4c,d). To quantify this acidification, we normalized the current and pH$_{in}$ signal (initial slope) evoked by each substrate to those evoked by His–Ala in the same oocyte. As the acidification rate is proportional to proton influx above an ~100 nA current threshold[13], the ratio between the normalized acidification and the normalized current provides a rough estimate of the number of protons released per elementary charge during substrate translocation (Fig. 4e,f). This analysis yielded ratios of 1.2 ± 0.1 for His–Ser ($n$ = 4) and −0.05 ± 0.05 ($n$ = 4) for Lys–Ala, in agreement with the concept of cytosolic acidification caused by the release of proton(s) bound to the translocated substrate. To test this model further, we measured the responses of MFSD1–GLMP oocytes to His–Glu. This dipeptide exists in four protonation states: a zwitterionic form (His$^+$–Glu$^-$), which predominates in the perfusion medium (pH$_{out}$ of 5.0, one unit above the Glu side chain pK$_a$ of 4.1); a cationic form, His$^+$–Glu$^0$, with a protonated Glu residue; an anionic form, His$^0$–Glu$^-$, with a deprotonated His residue; and low amounts of the neutral form, His$^0$–Glu$^0$. His–Glu evoked both an inward current and intracellular acidification with, remarkably, an acidification/current ratio of 2.4 ± 0.3 ($n$ = 4) instead of ~1 (Fig. 4c,e,f). MFSD1–GLMP thus substantially transports His–Glu in cationic form in our experimental conditions (pH$_{out}$ of 5.0, $V_m$ = −40 mV) since this form must release two protons per elementary charge when it reaches the cytosol (Fig. 4g). Finally, we tested the dipeptide Glu–Lys, which stood out as an atypical substrate in the proteoliposome assay. Glu–Lys evoked both an inward current and intracellular acidification in MFSD1–GLMP oocytes, with an acidification/current ratio of 1.2 ± 0.2 identical to His–Ala, in agreement with its entry in protonated, cationic state Glu$^0$–Lys$^+$ (Extended Data Fig. 3c–e). Additional uptake in the predominant zwitterionic form, Glu$^-$–Lys$^+$, may also occur but cannot be detected by the dual TEVC/pH$_{in}$ recording technique.

These data show that MFSD1 transports cationic dipeptides with or without concomitant acidification, whose presence and intensity depend on the number of titratable side chains. The simplest interpretation is that MFSD1 is not intrinsically coupled to protons, as initially thought, but operates instead as a dipeptide uniporter (for the

**Fig. 2 | Cationic dipeptides evoke an inward current in MFSD1–GLMP-expressing oocytes. a**, Surface biotinylation analysis of *Xenopus* oocytes expressing MFSD1$_{L11A/L12A}$-EmGFP and/or GLMP$_{Y400A}$-mKate2. The oocytes expressing EGFP in the cytosol validated the selectivity of surface labelling in streptavidin-bound fractions. The western blots are representative of three independent experiments. **b**, Fluorescence micrographs of representative oocytes ($n$ = 7 for either GLMP or MDFS1 alone and $n$ = 25 for MFSD1 + GLMP). The arrowheads show MFSD1–GLMP colocalization at the plasma membrane. **c**, TEVC recording of oocytes clamped at −40 mV and perfused with 10 mM Lys–Ala at pH 5.0. The traces show representative Lys–Ala-evoked currents of 7–14 oocytes per expression condition. Only 2 out of 14 oocytes expressing only MFSD1$_{L11A/L12A}$-EmGFP responded to Lys–Ala. The $P$ values were calculated using two-sided

Mann–Whitney $U$ tests (***$P$ ≤ 0.001). **d**, Dose-response relationship of the Lys–Ala current in MFSD1–GLMP oocytes. The current follows Michaelis–Menten kinetics with a $K_M$ of 2.6 ± 0.4 mM (mean ± s.e.m. of $n$ = 3 oocytes). **e**, Lys–Ala was applied to each MFSD1–GLMP oocyte at pH 5.0 and pH 7.0 (mean ± s.e.m. of $n$ = 4 oocytes). Two-tailed paired $t$-test, **$P$ ≤ 0.01. **f**, Response of MFSD1–GLMP oocytes to cationic amino acids and to the tripeptide Lys–Ala–Ala (10 mM each) at pH 5.0. The $P$ values were calculated using two-sided Mann–Whitney $U$ tests, *$P$ ≤ 0.05 and **$P$ ≤ 0.01 (mean ± s.e.m. of $n$ = 5 oocytes (Arg, His, Lys and Lys–Ala) and $n$ = 4 oocytes (Lys–Ala–Ala)). **g**, Response of MFSD1–GLMP oocytes to diverse dipeptides compared with Lys–Ala (mean ± s.e.m. of 4–11 oocytes per substrate). The source numerical data and unprocessed blots are available in the source data.

bioenergetical implications, see Discussion). The apparent discrepancy between the proteoliposome and TEVC assays, thus, reflects the inability of the former to detect transport of substrates that do not carry, and subsequently release, a proton bound to their sidechain(s).

## MFSD1 efficiently transports neutral and anionic dipeptides

The conclusion that MFSD1 operates as a uniporter revealed the technical limits of our fluorescence-based and electrophysiological assays for neutral, non-titratable substrates and prompted us to use stable isotope

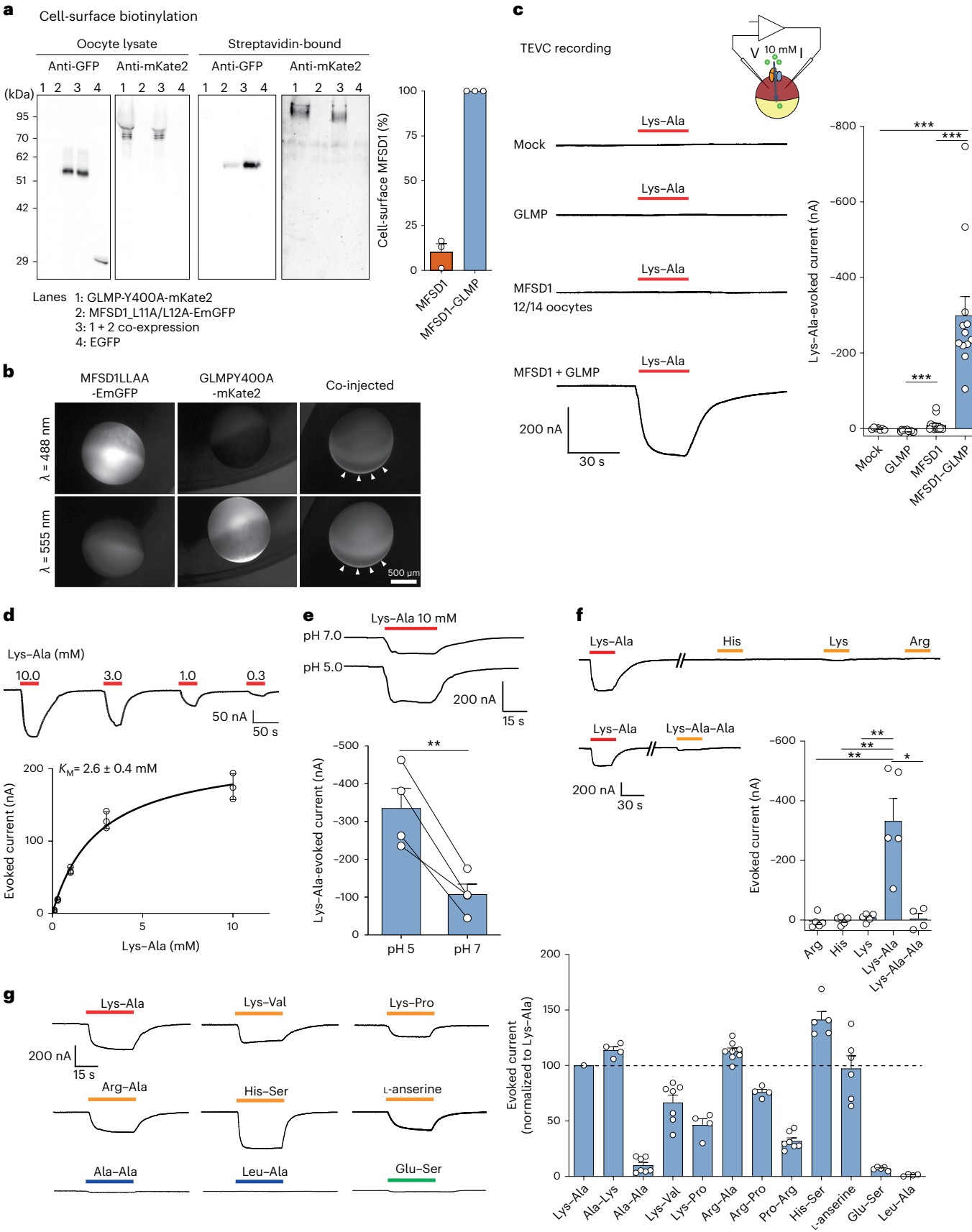

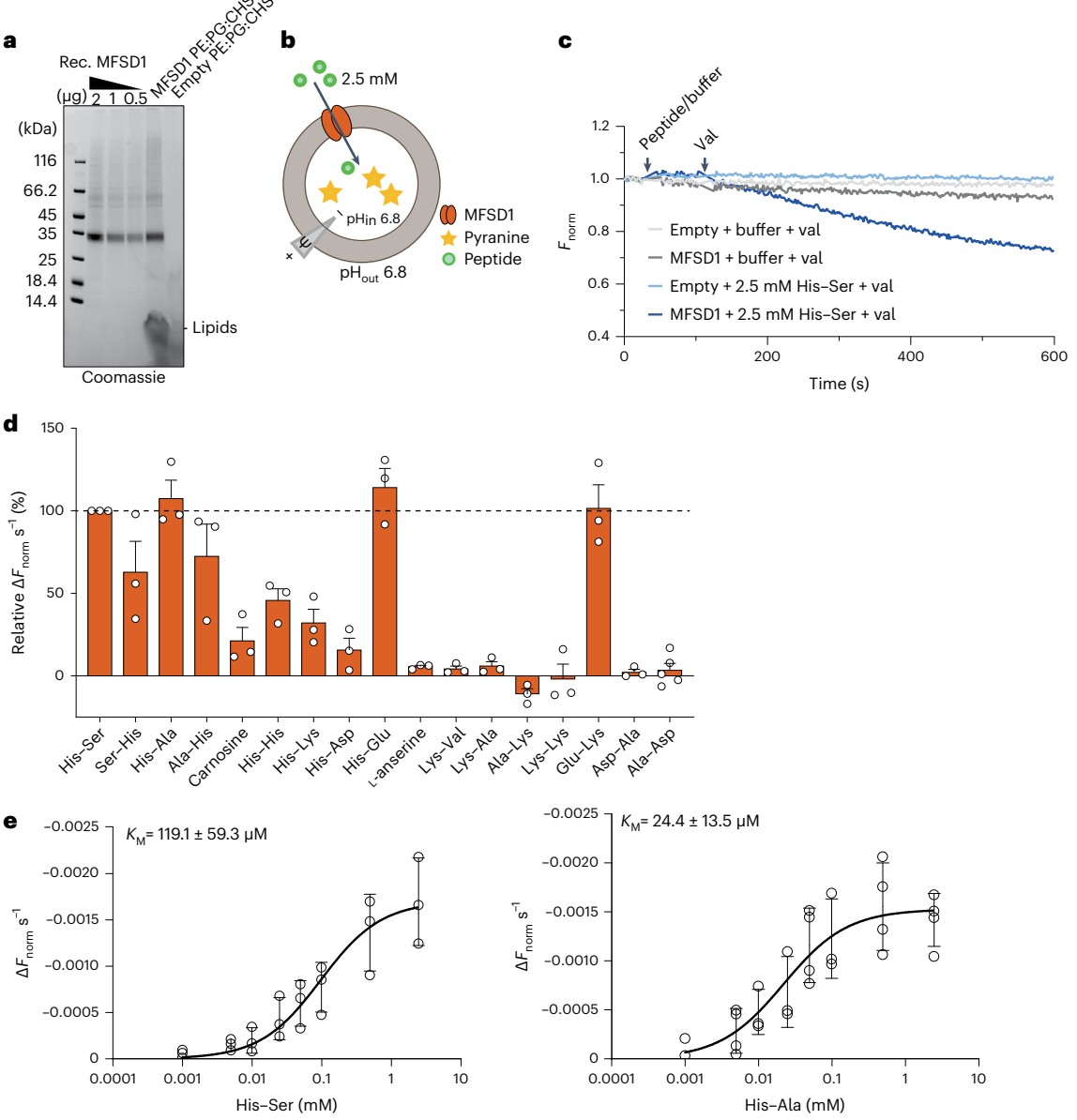

**Fig. 3 | MFSD1 is active as a dipeptide transporter in a liposome-based assay. a**, Coomassie-stained SDS–PAGE gel of MFSD1 after reconstitution into POPE:POPG:CHS liposomes (PE:PG:CHS). The experiment was performed 11 times. Rec., recombinant. **b**, A schematic of the experimental setup of liposome-based transporter assay. **c**, Representative traces of time-course measurements of uptake in the presence of 2.5 mM His–Ser and 1 μM val using MFSD1-containing liposomes (mmMFSD1) and those devoid of protein (empty). The addition of peptide or buffer and val during the measurements is indicated by the arrows.

**d**, Substrate specificity of MFSD1 measured for liposome-based uptake assays. The initial uptake rates for each peptide are given as a percentage of the determined initial uptake rate of His–Ser. The data are shown as mean ± s.d. for $n = 3$. **e**, Michaelis–Menten kinetics of uptake of His–Ser and His–Ala by MFSD1. The $K_M$ and $v_{max}$ values were calculated from three independent experiments using Prism GraphPad. The individual data points are plotted as mean ± s.d. The source numerical data are available in the source data.

tracing and targeted liquid chromatography (LC)–MS/MS analysis to monitor dipeptide transport. Leu–Ala, a good binder of MFSD1 both in vitro (Fig. 1h) and in cellula (Extended Data Fig. 2c), was synthesized in deuterated form (Leu($d_3$)–Ala) and applied at 10 mM to MFSD1–GLMP or mock oocytes for 20 min at pH 5.0. The oocyte extracts were then analysed by targeted LC–MS/MS (Fig. 5a). Leu($d_3$)–Ala showed little, yet significant, accumulation in MFSD1–GLMP oocytes. In contrast, these oocytes but not mock oocytes dramatically accumulated deuterated leucine (Leu($d_3$)) (Fig. 5b,c), showing that Leu($d_3$)–Ala is transported by MFSD1, yet quickly cleaved by intracellular peptidases. Accordingly, MFSD1–GLMP oocytes incubated with Leu($d_3$)–Ala also accumulated 'light' alanine over its endogenous level. Leu($d_3$) accumulation was dose dependent with a $K_M$ for Leu($d_3$)–Ala of 5.6 ± 1.6 mM ($n = 3$) (Extended

Data Fig. 4a). To compare the rate of Leu–Ala transport with that of electrogenic substrates, we performed absolute quantification of the Leu($d_3$) and Ala signals during the time-dependent linear phase of Leu($d_3$)–Ala uptake (Extended Data Fig. 4b). This yielded a Leu–Ala transport rate of 1.32 ± 0.14 pmol s⁻¹ and 1.52 ± 0.20 pmol s⁻¹ per MFSD1–GLMP oocyte for the Leu($d_3$) and Ala signals, respectively (Fig. 5d), a value about half that of Lys Ala (3.11 ± 0.52 pmol s⁻¹ per oocyte ($n = 12$); Figs. 2c and 4d) despite the lack of electric driving force with Leu–Ala. The quantification also showed that Leu($d_3$) and Ala are released at equimolar levels (Ala/Leu($d_3$) ratio = 1.09 ± 0.09, $n = 3$) following Leu($d_3$)–Ala import (Fig. 5e and Extended Data Fig. 4c).

Next, we took advantage of the Ala signal to compare the uptake of diverse Ala-containing dipeptides. MFSD1–GLMP oocytes highly

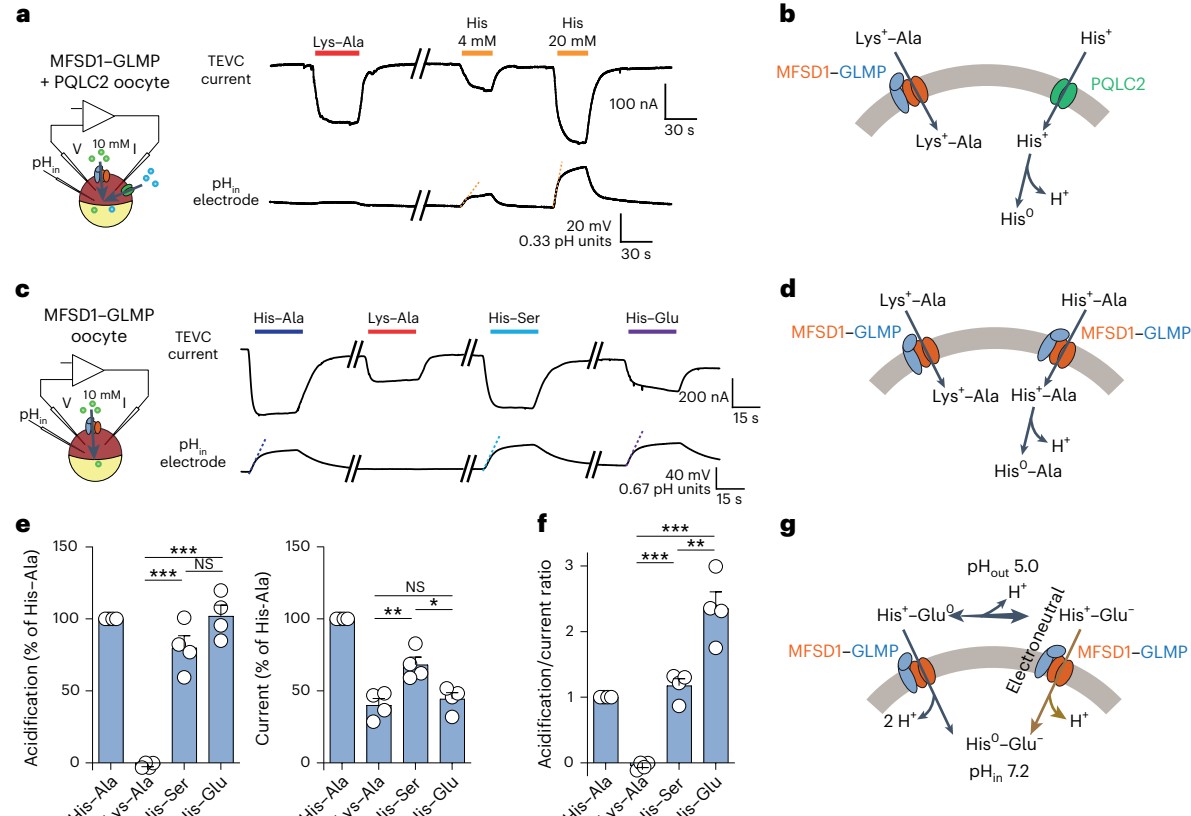

**Fig. 4 | MFSD1 is a dipeptide uniporter. a**, Combined TEVC and pH$_{in}$ recording of oocytes expressing both MFSD1–GLMP and PQLC2 (sorting mutant L290A/L291A) at their surface. His, but not Lys–Ala, applied at pH 5.0, induces intracellular acidification (orange dotted lines). The traces are representative of five oocytes shown in Extended Data Fig. 3b. **b**, A model for the acidification induced by His following its release from PQLC2. **c**, Combined TEVC and pH$_{in}$ recording of an MFSD1–GLMP oocyte perfused with the indicated dipeptides (10 mM) at pH 5.0. **d**, A model accounting for the selective acidification by His-containing dipeptides. **e**, The experiment in **c** was repeated on four MFSD1–GLMP oocytes. The data are means ± s.e.m. of the acidification and current responses normalized to His–Ala (two-tailed unpaired *t*-test). *$P \leq 0.05$, **$P \leq 0.01$ and ***$P \leq 0.001$. **f**, Normalized acidification/current ratios provide

the number of protons released per translocated elementary charge for each substrate (two-tailed unpaired *t*-test). Mean ± s.e.m. **$P \leq 0.01$ and ***$P \leq 0.001$. **g**, A model accounting for the high number of protons released by His–Glu. At the tested potential (−40 mV), His–Glu molecules would be taken up by MFSD1–GLMP predominantly in the minor cationic form, His$^+$–Glu$^0$, releasing two protons per elementary charge. The higher acidification/current ratio observed (2.5 ± 0.2) may result either from the non-linear acidification/current relationship (Main) or from simultaneous uptake in the predominant zwitterionic form, His$^+$–Glu$^-$, which would release another proton in an electroneutral manner. The source numerical data are available in the source data.

accumulated Ala and the second amino acid over their endogenous level with all tested neutral, cationic and anionic dipeptides (Fig. 5f,g and Extended Data Fig. 4d–g). The uptake activity reported by Ala showed the highest transport activity for Leu($d_3$)–Ala, Ala–Ala and His–Ala, followed by Lys–Ala and Glu–Ala and, to a lesser extent, Ala–Asp. Another anionic dipeptide, Glu–Ser, is also transported by MFSD1–GLMP (Extended Data Fig. 4d,h,i). We concluded that MFSD1 has a broad dipeptide selectivity.

**Cryo-EM structure determination of GLMP–MFSD1**

To elucidate the molecular mechanism of substrate recognition, we determined the structure of MFSD1–GLMP in the apo- and dipeptide-bound states[43]. To test if the interaction of MFSD1 and GLMP is stable in vitro, MFSD1 and GLMP were individually or co-expressed, and a pull-down assay confirmed that MFSD1 interacts with GLMP even after detergent-extraction (Extended Data Fig. 5a). We also designed a fusion construct connecting GLMP with MFSD1 via a glycine/serine linker (GLMP–MFSD1).

The GLMP + MFSD1 co-complex and the GLMP–MFSD1 construct (Extended Data Fig. 5b,e) exhibited similar stabilization effects by dipeptides as MFSD1$_{WT}$ (Extended Data Fig. 5c,f). They were more thermostable than MFSD1$_{WT}$ ($T_m$ of 40 °C), though the transport activity

in proteoliposomes was reduced for the fusion, whereas the purified complex was as active as MFSD1$_{WT}$ (Extended Data Fig. 5d,g). Since GLMP–MFSD1 could be purified at higher yields, we used this construct for structure determination. We obtained three-dimensional reconstructions for the apo- and substrate (His-Ala)-bound structures (GLMP–MFSD1$_{apo}$ and GLMP–MFSD1$_{His–Ala}$) at nominal resolutions of 4.2 Å and 4.1 Å (Extended Data Fig. 6 and Supplementary Table 4), though the luminal domain of GLMP and core parts of MFSD1 reach a local resolution up to 3.43 Å. Given the slightly higher resolution of the GLMP–MFSD1$_{His–Ala}$ dataset, we used this reconstruction for model building. The EM map resolved most of both proteins, including *N*-glycans (Asn85, Asn94, Asn157, Asn228 and Asn331) of GLMP (Fig. 6a–c and Extended Data Fig. 7a–c). For GLMP, the missing regions include residues 1–35, 99–100, 135–141, 178–181 and 392–404. For MFSD1, residues 1–35, 446–464 and the inter-domain loop region (residues 241–260) could not be modelled.

MFSD1 is captured in an outward-open conformation where the binding site is accessible from the lysosomal lumen (Fig. 6a–c). The transmembrane (TM) domains of MFSD1 adopt the canonical MFS fold formed by 12 TM helices organized in two six-helix bundles (N-domain by TM1-6 and C-domain by TM7-12) with both termini facing the cytoplasm (Fig. 6b,c)[40,52,53]. For GLMP, the luminal domain and

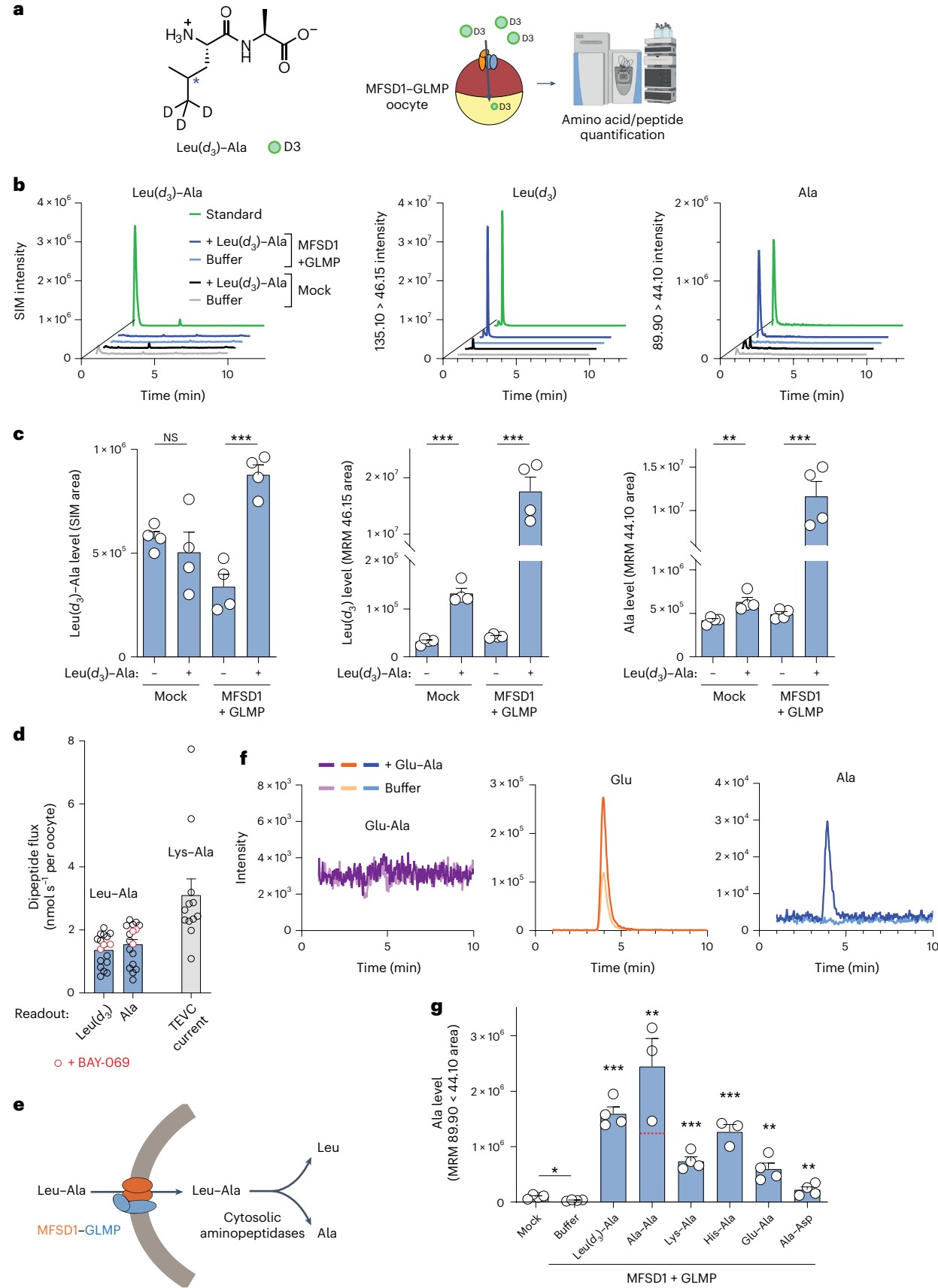

its single-span TM helix could be resolved (Fig. 6a–c and Extended Data Fig. 7b). The TM helix of GLMP is located directly adjacent to the C-domain of MFSD1. We could identify five of the six N-linked glycosylation sites present in a previous X-ray structure of the luminal domain

(Protein Data Bank (PDB) 6NYQ) and confirmed in vivo[44] (Extended Data Fig. 7c). The luminal domain of GLMP adopts a β-sandwich fold (Fig. 6b) that is structurally similar to a dimerization domain found in a cellodextrin phosphorylase from *Clostridium thermocellum* (PDB 5NZ7)[54].

**Fig. 5 | MFSD1 has a broad dipeptide selectivity. a**, Heavy isotope tracer approach used to monitor Leu–Ala transport. **b**, Representative LC–MS chromatograms of ≥5 independent experiments. The amount of standard (green lines) was 3.9 pmol for Leu($d_3$)–Ala and 15.6 pmol for Leu($d_3$) and Ala. **c**, Relative quantification of the chromatographic peak area of Leu($d_3$)–Ala, Leu($d_3$) and Ala in extracts from mock and MFSD1–GLMP oocytes, incubated or not, with, 10 mM Leu($d_3$)–Ala for 20 min at pH 5.0. The data are means ± s.e.m. of four oocytes from a representative example of three independent experiments (two-tailed unpaired $t$-tests). *$P \le 0.05$, **$P \le 0.01$ and ***$P \le 0.001$. **d**, Absolute quantification of Leu–Ala uptake. The data are means ± s.e.m. of 17 oocytes from two oocyte batches. In one experiment, some oocytes were treated with the branched-chained amino acid transaminase inhibitor BAY-069. The Lys–Ala currents from Fig. 2c were divided by the Faraday constant and plotted with the same scale (grey bar) to allow comparison with Leu–Ala uptake. **e**, A model accounting for the LC–MS/MS data. **f**, Representative LC–MS chromatograms of eight MFSD1–GLMP oocytes from two batches incubated for 23 min at pH 5.0 with 10 mM Glu–Ala. **g**, Quantification of Ala in oocytes incubated for 23 min with the indicated dipeptides (10 mM). The means ± s.e.m. of three to four oocytes are depicted. The red dotted line at mid-height of the Ala–Ala bar is shown for comparison with other substrates. Two-tailed unpaired $t$-tests relative to MFSD1–GLMP oocytes incubated in dipeptide-free buffer; *$P \le 0.05$, **$P \le 0.01$ and ***$P \le 0.001$. The source numerical data are available in the source data.

## The substrate binding site of MFSD1

A comparison of the three-dimensional reconstructions of both data-sets revealed an additional density for GLMP–MFSD1$_{His–Ala}$ (Fig. 6d) in the cavity between the two helical bundles (Fig. 6c). This potential binding site is located approximately halfway into the membrane-spanning region and is formed by TM1 (Tyr56 and Tyr59), TM4 (Glu150) and TM5 (Gln176 and Arg181) of the N-domain and TM7 (Tyr276 and Phe280), TM10 (Tyr365 and Trp373) and TM11 (Gln393, Gln396, Asn397 and Leu400) of the C-domain (Fig. 6d). The cavity exhibits a bipolar sur-face character mainly caused by residues Glu150 and Arg181(Fig. 6e).

Unambiguous placement of the His–Ala peptide was impossible owing to its insufficiently resolved density (Fig. 6d). To further inves-tigate peptide binding, we performed MD simulations in a simple lipid bilayer reflecting that of the liposomes in the presence of differ-ent dipeptides. The dipeptides Leu–Ala, Lys–Ala and His–Ala (with either a neutral or positively charged histidine, His$^0$–Ala or His$^+$–Ala, respectively) were placed in two different orientations based on the peptide density observed in the cryo-EM reconstruction of GLMP–MFSD1$_{His–Ala}$. The peptides in peptide orientation 1 (PO1) had their C-terminus positioned towards a patch of polar residues (Gln393, Gln396 and Asn397). The side chain of the first dipeptide residue is pointing towards Arg181 (Extended Data Fig. 8a and Extended Data Fig. 9). For peptide orientation 2 (PO2), the dipeptide's N-termini and C-termini are near residues Glu150 and Arg181, respectively (Extended Data Fig. 8a–g). After 500 ns of simulation time, the peptides starting from PO2 deviate less from their starting pose while peptides in PO1 flipped so that their N-termini and C-termini interact with Glu150 and Arg181 (Extended Data Fig. 8b–g). In two simulations with a peptide starting in orientation PO1, Leu–Ala$_{pose1,run1}$, and His$^0$–Ala$_{pose1,run2}$, the corresponding peptides diffused from the binding cavity (Extended Data Fig. 8a). For the substrate His–Ala in the protonated state, the histidine side chain is close to residue Asp60, though in two simula-tions, the C-termini of the peptides lost their interaction with Arg181. The neutral His–Ala peptide displays more flexibility of the histidine side chain in the binding site, while the peptide remains sandwiched between Arg181 and Glu150 (Extended Data Fig. 8a).

On the basis of the MD and cryo-EM data, we hypothesize that the peptide orientation at the end of the MD simulation from PO2 (Fig. 6f and Extended Data Fig. 8b–g) represents the most probable dipeptide binding mode. In comparison with other peptide-bound structures of the POT family (PepT1 or DtpB[35]), it is striking that MFSD1 displays a similar recognition pattern, even though MFSD1 does not share any of the POT signature motifs or their coupling mechanism (Extended Data Fig. 8f).

To validate our peptide recognition and transport findings, we mutated selected highly conserved peptide binding-site residues (Fig. 6g) with Gln176 showing greater variability among different organ-isms (Supplementary Fig. 1). Most mutants, except for MFSD1$_{E150R}$, MFSD1$_{W373F}$ and MFSD1$_{Y56F}$, could be expressed and purified (Extended Data Fig. 10a,b). Peak fractions of the remaining mutants were used for nanoDSF experiments and liposome-based transport assays (Fig. 6g and Extended Data Fig. 10c). MFSD1$_{D60A}$, MFSD1$_{E150A}$, MFSD1$_{R181A}$ and MFSD1$_{R181E}$ did not exhibit a characteristic thermal unfolding trace and could not be analysed further (Extended Data Fig. 10d,e). MFSD1$_{W373A}$ had a higher melting temperature ($T_m$ of 46.6 °C) than MFSD1$_{WT}$ ($T_m$ of 40 °C), which did not increase upon peptide addition. The remaining mutants could still interact with dipeptides. The stabilization pattern across selected peptides differed from MFSD1$_{WT}$ for MFSD1$_{Q176K}$, where Pro–Arg, Arg–Pro and Lys–Val had no effect. On the basis of these results, MFSD1$_{Y56A}$, MFSD1$_{D60A}$, MFSD1$_{E150A}$, MFSD1$_{Q176K}$, MFSD1$_{R181A}$ and MFSD1$_{W373A}$ were selected for liposome-based uptake assays of His–Ala and His–Ser. Most MFSD1 mutants lost their transport activity. For MFS-D1$_{Y56A}$, transport of His–Ala and His–Ser was still detectable, although the signal was reduced by ~50% compared with MFSD1$_{WT}$ (Fig. 6g). While MFSD1$_{Q176K}$ binds peptides, it did not transport them. Residue Gln176 is close to the ligand density identified in the cryo-EM map of GLMP–MFSD1$_{His–Ala}$ (Fig. 6d) but is oriented away from the peptides screened in MD simulations (Fig. 6f and Extended Data Fig. 8a). Nevertheless, this residue is probably crucial for the transport mechanisms but less for peptide binding. As expected, mutating D60, E150 and R181 had greater implications on the stability of the protein and its ability to transport peptides, implying that these residues are critical for the interaction of the dipeptide with MFSD1 (Fig. 6e–g and Extended Data Figs. 8a and 10d,e). The putative transport cycle model is shown in Fig. 6h.

## Gating mechanism of MFSD1 using conformational predictions

The transition from the outward-open to the inward-open state is essen-tial for substrate translocation across the lysosomal membrane. For

**Fig. 6 | The outward-open structure of GLMP–MFSD1. a**, Cryo-EM map of GLMP–MFSD1$_{His–Ala}$. The N- and C-domain of MFSD1 are coloured yellow and orange, respectively. GLMP is coloured blue. **b**, Topology diagram of MFSD1 and GLMP. The N- and C-termini are labelled, and the secondary structure elements are numbered. **c**, Cartoon representation of GLMP–MFSD1 with top view of MFSD1. The numbering of TMs is indicated. Sugar modifications (acetylglucosamine (NAG)) identified on GLMP are coloured pink. **d**, Additional binding-site density was found for the GLMP–MFSD1 data set in the presence of the dipeptide with His–Ala (MFSD1$_{His–Ala}$) compared with the apo dataset (MFSD1$_{apo}$). The map of MFSD1$_{His–Ala}$ is shown as light blue surface and that of the apo dataset as grey mesh (light grey). Both the maps are depicted at $\sigma = 6$. The residues surrounding the extra density are labelled. **e**, The electrostatic surface potential (expressed as kT/e, with k, Boltzmann constant; T, temperature in K; and e, elementary charge), calculated with the APBS plugin in PyMol, highlights the bipolar character of the binding site. The residues that were mutated in this study are framed in bold black. **f**, Binding of the protonated dipeptide His$^+$–Ala (green) as observed after 500 ns of MD simulations. Hydrogen bonds are indicated as dashed black lines, and residues used for mutational studies are framed in bold black. **g**, Effect of mutations of binding-site residues on uptake of His–Ala or His–Ser compared with MFSD1$_{WT}$. The uptake rates are given as mean ± s.d. for $n = 8$ (MFSD1$_{WT}$) or $n = 4$ (mutants) of independent experiments. **h**, A schematic of transport of dipeptides (blue (N-terminus) and red (C-terminus) sticks) by the GLMP–MFSD1 complex. The cytoplasmic gate formed by residues N157, F173, W373 and Y369 is shown (shown as a grey bar) as well as residues E150 and R118 involved in peptide coordination. The source numerical data are available in the source data.

MFS transporters, the alternating access to the binding site is mediated by the movement of the N- and C-domains against each other, also known as the rocker-switch model[40,41]. Though the experimental structure of MFSD1 represents the outward-open state only, we used two additional conformations (representing the inward-open and outward-occluded state) derived from AlphaFold2 (ref. 55) predictions (Supplementary Fig. 2) to analyse the conformational transitions

occurring during a transport cycle. Therefore, we aligned the N-domain and C-domain of the outward-occluded and inward-open models to the outward-open cryo-EM structure, termed MFSD1$_{out}$. Overall, the two domains do not differ greatly when superimposed individually onto the N- or C-domain of MFSD1 (root mean square deviation of Cα atoms (r.m.s.d.$_{Cα}$) range of 0.85–1.27 Å, Supplementary Fig. 2a,b and Supplementary Video 1). However, the superposition of the full-length

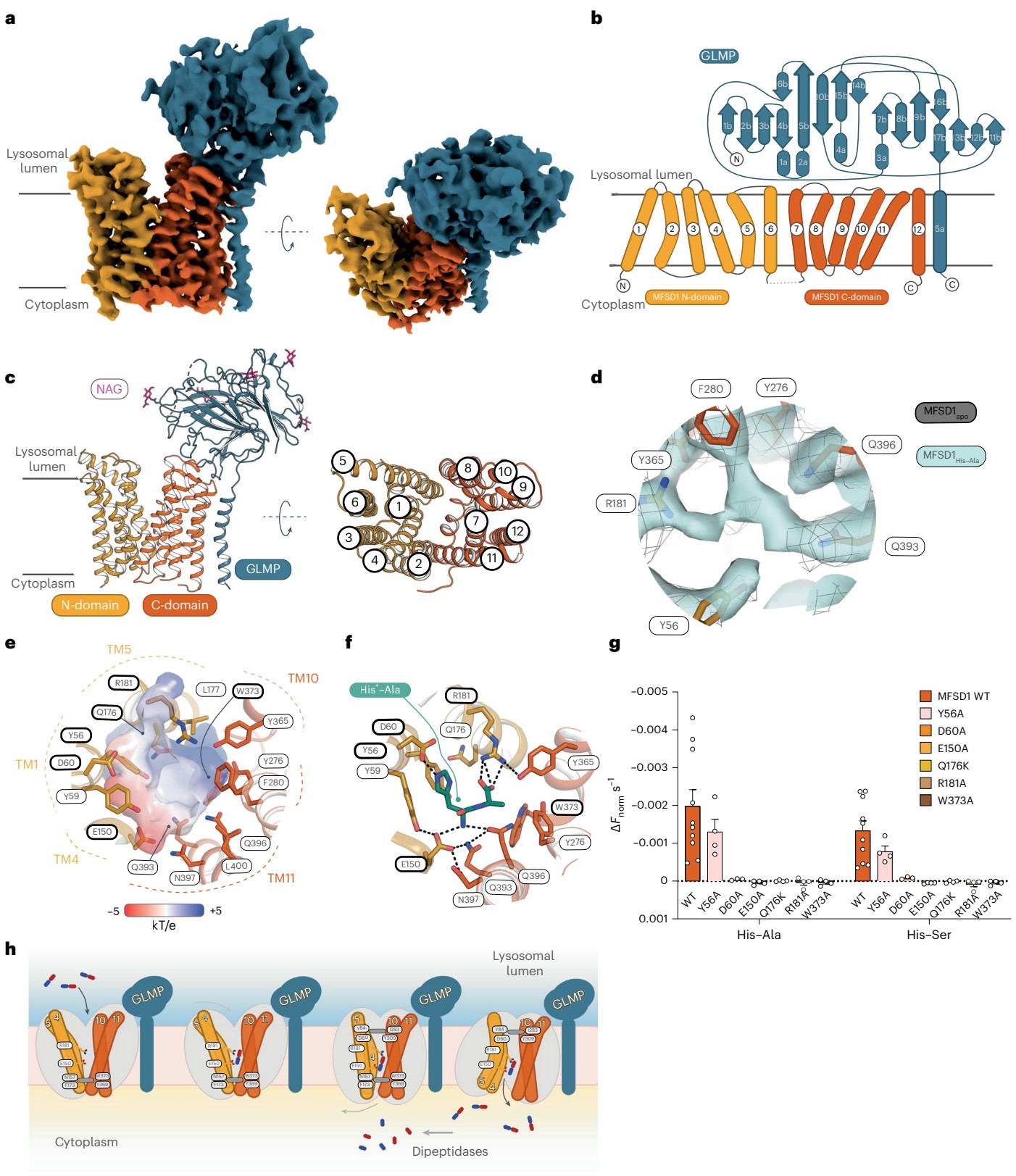

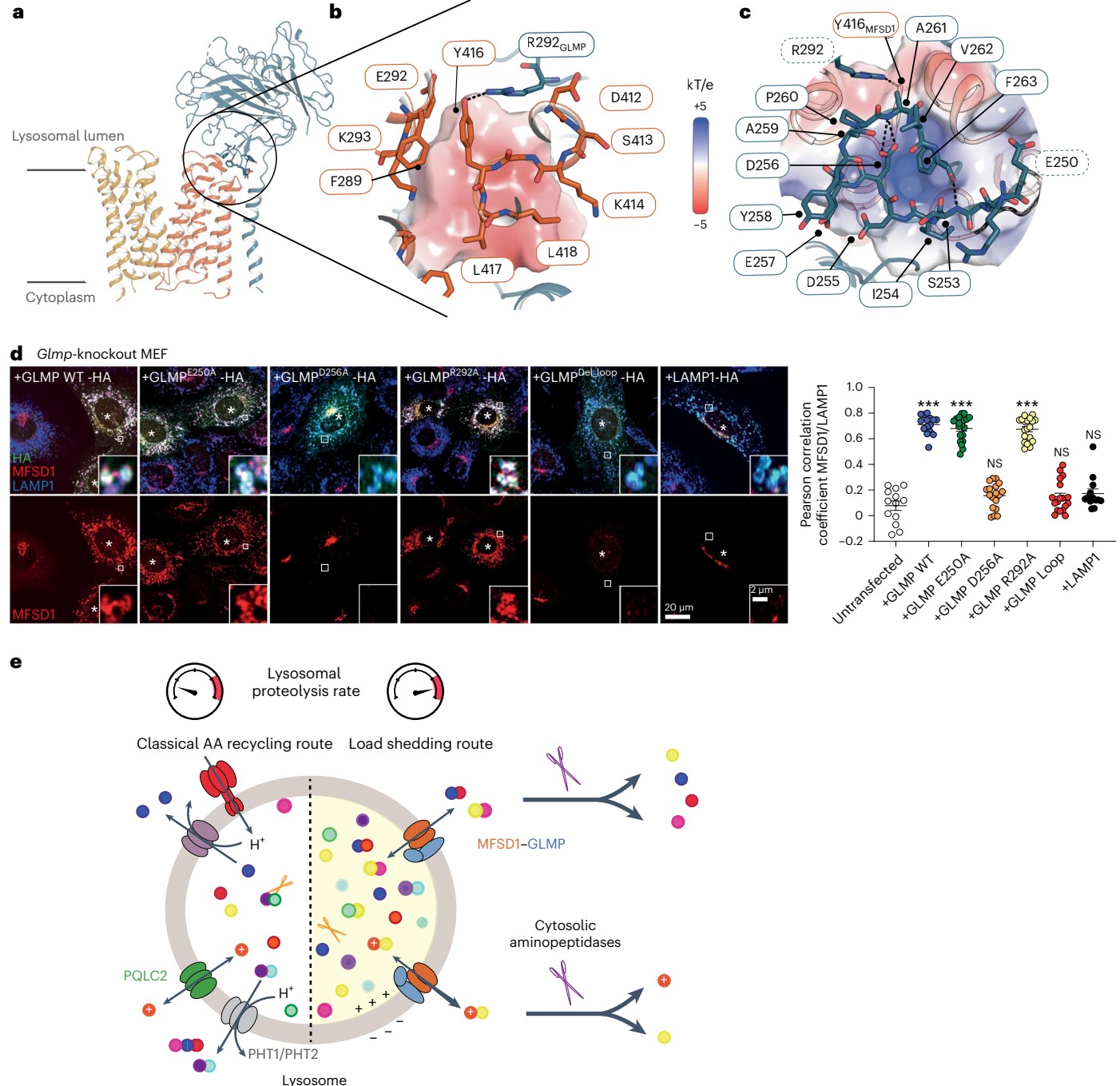

**Fig. 7 | Interaction of GLMP with MFSD1. a**, Cartoon representation of GLMP in complex with MFSD1. The interaction site of GLMP with MFSD1 is highlighted in stick representation. **b**, Zoom in on the interaction of MFSD1 to GLMP as viewed from MFSD1. The electrostatic surface of GLMP is shown. Y416 (MFSD1) is in hydrogen-bond (H-bond) distance to R292 (GLMP) and is highlighted as a black dotted line. **c**, Zoom in on the interaction of GLMP to MFSD1 as viewed from GLMP. The electrostatic potential surface of MFSD1 is highlighted, indicating complementarity to the GLMP surface. Besides the salt bridge between residues Y416 (MFSD1) and R292 (GLMP), residue D256 (GLMP) is at an H-bond distance from the backbone amide of A261 (GLMP), shown as black dotted lines. The loop region spanning residues 253 to 260 was mutated (blue border). The single-point mutants are highlighted in bold. **d**, Immunofluorescence-staining of endogenous MFSD1 (red) after transfection with hemagglutinin (HA)-tagged GLMP, GLMP mutants and LAMP1 (green) in *Glmp*-knockout MEFs. The endogenous LAMP1 is shown in blue. The transfected cells are marked with an asterick. The Pearson correlation coefficient for MFSD1/endogenous LAMP1 is shown in the right panel. The means ± s.e.m. for *n* = 13–20 cells are shown over two independent experiments (two-tailed unpaired *t*-tests). *$P \leq 0.05$, **$P \leq 0.01$ and ***$P \leq 0.001$. **e**, Cellular model for the role of MFSD1 in the recycling of amino acids (AA) derived from lysosomal proteolysis. Owing to its broad selectivity and low affinity for dipeptides, MFSD1 provides an alternative recycling route when the lysosomal breakdown of proteins exceeds the capacity of lysosomal amino acid exporters. Fast cleavage of the released dipeptides by cytosolic aminopeptidases drives MFSD1 activity in the export direction and provides amino acids for biosynthetic pathways. The narrow selectivity of MFSD1 for dipeptides (in contrast with PHT1 and PHT2 transporters) prevents competition by single amino acids and protects this load-shedding route from amino acid overload. The source numerical data are available in the source data.

proteins (r.m.s.d.$_{C\alpha}$(out-occluded) of 3.5 Å, r.m.s.d.$_{C\alpha}$(outward-open–inward-open) of 4.87 Å) revealed that the N-domain undergoes a larger helical rearrangement in both predicted states, compared with the C-domain (Supplementary Fig. 2c–f). During the outward-open to the outward-occluded transition, the N-domain folds onto the substrate cavity, thereby closing it off from the lysosomal lumen, while the cytoplasmic bottom of the transporter stays static (Supplementary Fig. 2g,h). The cytoplasmic gate of MFSD1$_{out}$ is formed by residues Asn57 (TM4), Phe173 (TM5), Trp373 (TM10) and Tyr389 (TM11). Mutating Trp393 to alanine stabilized MFSD1 but interfered with peptide binding (Extended Data Fig. 10d,e) and transport (Fig. 6f). Further interactions between the N- and C-domain retained by Glu150 with Asn397, Arg181 with Tyr365 and a pi-cation interaction of Lys287 with Phe378 on the cytoplasmic side stabilize the outward-open conformation (Supplementary Fig. 2i).

Interactions on the cytoplasmic side are similar between the outward-occluded AF2 model and MFSD1$_{out}$. However, access to the binding cavity from the lysosomal lumen is blocked by residues Tyr59 (TM1), Asp60 (TM1), Met81 (TM2), Tyr84 (TM2), Ile283 (TM7) and Tyr309 (TM8), forming the lysosomal gate. During the transition from the outward-occluded to the inward-open state, the cytoplasmic gate opens by a swinging motion of the bottom half of the N-domain away from the C-domain (Supplementary Fig. 2j). This disrupts the cytoplasmic gate to open the cavity and thus facilitates the release of the substrate. The luminal gate remains closed and is formed by the same residues as observed in the outward-occluded state (Supplementary Fig. 2h,j). The conformation is further stabilized through interactions between the side chain of Lys287 with the backbone carbonyl of Ala64 and between the Gln66 side chain and the backbone amide of Val288 (Supplementary Fig. 2j). On the basis of the analysis between the experimental outward-open structure and the two AF2 models in the occluded and inward-open states, it becomes apparent that the positions of the peptide-binding residues Glu150 and Arg181 move towards the cytoplasmic side (Supplementary Fig. 2k) and thus might push the dipeptide coordinated between both residues towards the cytoplasmic opening of MFSD1 to facilitate substrate release (Fig. 6h).

### The interaction of GLMP with MFSD1

Previous in vivo studies highlighted that GLMP is crucial to protect MFSD1 from degradation[43,44]. All our data show that GLMP and MFSD1 form a stable complex. Based on the analysis of the cryo-EM structure, we identified a loop region of GLMP (residues 250-263) near the luminal region of the C-domain of MFSD1. This region seems to be pivotal for the interaction between both proteins (Fig. 7a,b) and was not resolved in the X-ray structure of GLMP (Extended Data Fig. 7c), though it is conserved in GLMP homologues (Supplementary Fig. 3). The electrostatic surface of MFSD1 in this region is positively charged, while it is negative for GLMP, indicating an interaction via polar interactions (Fig. 7b). Arg292(GLMP) is in hydrogen bonding distance with Tyr416(MFSD1), and the loop is further stabilized by an intra-loop interaction of Asp256(GLMP) with the backbone amide of Ala261(GLMP) (Fig. 7c). To evaluate the role of this interaction in a cellular context, we used *Glmp*-knockout mouse embryonic fibroblasts (MEFs), in which endogenous MFSD1 is strongly reduced, and the remaining MFSD1 localizes to the Golgi apparatus. Re-expression of GLMP rescues the lysosomal localization of MFSD1[43]. We exchanged the interaction-surface loop in HA-tagged GLMP (253–263) with four alanine residues and generated constructs with individual amino acid exchanges (E250A, D256A, and R292A) to test if these constructs can still rescue lysosomal MFSD1 localization (Fig. 7d). HA-tagged LAMP1 served as a negative control. Re-expression of GLMP$_{WT}$ efficiently restored the levels and localization of endogenous MFSD1 in *Glmp*-knockout MEFs. In contrast, the construct with the deleted interaction-surface loop did not restore lysosomal MFSD1. Two point

mutants (Glu250Ala and Arg292Ala) fully restored lysosomal MFSD1, while Asp256Ala did not, indicating this amino acid is most critical in the interaction between MFSD1 and GLMP (Fig. 7d). These data confirm the interaction surface between MFSD1 and GLMP in the loop between 250–263 in vivo.

## Discussion

In this study, we provide compelling evidence that MFSD1 functions as a general, low-affinity uniporter for dipeptides. Some cationic dipeptides accumulated in MFSD1-deficient lysosomes, providing a clue to elucidate its transport activity. Studies of purified MFSD1 and the MFSD1–GLMP complex showed that MFSD1 binds and efficiently transports diverse cationic, neutral and anionic dipeptides but not single amino acids or longer peptides. Our combined cryo-EM and MD simulation data provided a structural basis for this substrate selectivity since a highly conserved glutamate (Glu150) and arginine (Arg181) residue clamps the N- and C-termini, respectively, of the dipeptide in an extended conformation. The substrate binding site of MFSD1 thus acts as a 'molecular ruler' that dictates the strict selectivity for dipeptides while accommodating diverse side chains, explaining its promiscuity among dipeptides. This binding mode is reminiscent of the POT family[35,56,57], although MFSD1 lacks any typical POT signature motifs. A similar molecular ruler principle applied to cystine, the oxidized form of cysteine, underlies the narrow substrate selectivity of cystinosin, the lysosomal transporter defective in cystinosis[18,19].

From a lysosomal physiology perspective, MFSD1 differs from PHT1 and PHT2 in several respects. First, MFSD1 is ubiquitously expressed[43], whereas the expression of PHT1 and PHT2 strongly varies across mammalian organs and tissues[58]. Second, it has a strict selectivity towards dipeptides, while the SLC15 members transport dipeptides and tripeptides. Third, MFSD1 affinities range from 24 μM to 4 mM depending on the dipeptides, whereas PHT1 and PHT2 operate in the 10–100 μM range[34,58]. Therefore, lysosomal export of dipeptides by MFSD1 may intervene when there is a build-up of intralysosomal dipeptides, for instance, when cathepsin C, which has dipeptidyl peptidase activity[59], is more active or, more generally, when the overall endopeptidase activity of the lysosomal lumen exceeds its exopeptidase activity.

MFSD1 also differs from POT family members and many lysosomal transporters by its bioenergetical properties since it is not intrinsically coupled to protons. Indeed, luminal protons (extracellular protons in our oocyte assay) were co-transported exclusively with a subset of substrates harbouring a side chain (His, Glu) with a pK$_a$ relatively close to the luminal pH but not with substrates such as Lys–Ala (side chain pK$_a$ of 10.5) or Leu–Ala. The simplest interpretation is that protons are carried by the dipeptide's titrable side chain rather than through an MFSD1 proton pathway.

MFSD1 is, thus, most probably a uniporter, that is, it transports a single solute. Therefore, in contrast with intrinsically proton-coupled lysosomal exporters (proton symporters), which are governed by the steep pH gradient of the lysosome, MFSD1 is prone to reverse direction in the lysosomal membrane, explaining the old paradoxical observation that high concentrations of dipeptides enter and burst purified lysosomes more efficiently than single amino acids[23]. However, in a cellular context, three forces drive MFSD1 in the export direction (Fig. 7e). The first, general one is the efficient hydrolysis of dipeptides by cytosolic aminopeptidases[60,61], as highlighted by the full cleavage of Leu($d_3$)–Ala and other dipeptides into single amino acids after their discharge into the cytosol. The second driving force, restricted to cationic dipeptides, is the positive-inside polarization of the lysosomal membrane[62,63]. This polarization selectively accelerates the lysosomal export of cationic dipeptides, presumably explaining why this dipeptide subclass stood out in our initial metabolomics profiling of MFSD1-deficient lysosomes. Finally, for titratable dipeptides, the proton carried by these substrates indirectly couples their transport to the pH gradient. These dipeptides should, thus, be actively

exported against their concentration gradient if MFSD1 prefers their protonated forms.

Taken together, these features (ubiquitous expression and broad selectivity among dipeptides) strongly suggest that MFSD1 provides an alternative route to supply amino acids for biosynthetic pathways when the 'classical' route mediated by lysosomal amino acid transporters and PHT1 and PHT2 is overloaded (Fig. 7e). Moreover, the strict selectivity of MFSD1 for dipeptides protects this load-shedding route from competition by single amino acids or longer peptides.

MFSD1 and GLMP critically interdepend on each other[44]. This interdependence is also highlighted in our experiments, in which only the co-expression of MFSD1 and GLMP led to detectable MFSD1 at the oocyte plasma membrane and a transport current. Under these conditions, the system did not allow for the analysis of how GLMP affects the substrate translocation activity of MFSD1. However, our in vitro liposome reconstitution experiments allowed a direct comparison of the MFSD1 activity alone or with GLMP as a fusion construct or in complex. The reconstituted complex of GLMP and MFSD1 exhibited similar uptake rates compared with MFSD1$_{WT}$ only, whereas the transport activity for the fusion protein was reduced. This is probably due to the linker approach used to connect both proteins, which has been beneficial for cryo-EM studies but reduced the conformational flexibility crucial for transport activity. Our cryo-EM data revealed a crucial loop within GLMP interacting with the lysosomal surface of the C-terminal domain of MFSD1, confirmed by mutagenesis. The MFSD1–GLMP structure illustrates that the N-glycosylated GLMP shields the luminal loops and the surface of the non-glycosylated MFSD1 from proteases, supporting the presumed function as a 'protector' similar to OSTM1 for the lysosomal chloride channel CLCN7 (ref. 64).

During the revision of our manuscript, another study identified MFSD1 as a lysosomal dipeptide uniporter based on the accumulation of dipeptides with at least one cationic residue in MFSD1-defective lysosomes and the electrogenic transport of such dipeptides[65]. The authors concluded that MFSD1 is highly specific for this subset of dipeptides. However, they did not test whether other dipeptides are transported in an electroneutral manner nor whether they compete with cationic dipeptides in the electrophysiological assay. Therefore, their diverging conclusion about the substrate selectivity of MFSD1 merely reflects the positive-inside polarization of lysosomes and the bias of the electrophysiological assay towards cationic dipeptides[66].

## Online content

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

[1]Centre for Structural Systems Biology, Hamburg, Germany. [2]European Molecular Biology Laboratory Hamburg, Hamburg, Germany. [3]Saints-Pères Paris Institute for the Neurosciences, Université Paris Cité, Centre National de la Recherche Scientifique, Paris, France. [4]Department of Chemical Engineering, Stanford University, Stanford, CA, USA. [5]Department of Genetics, Stanford University, Stanford, CA, USA. [6]The Institute for Chemistry, Engineering and Medicine for Human Health, Stanford University, Stanford, CA, USA. [7]Laboratoire de Chimie et Biochimie Pharmacologiques et Toxicologiques, CNRS UMR 8601, Université Paris Cité, Paris, France. [8]Department of Pharmacology, University of Virginia School of Medicine, Charlottesville, VA, USA. [9]Graduate School of Agriculture, Kyoto University, Kyoto, Japan. [10]Institute of Biochemistry, Christian-Albrechts-University Kiel, Kiel, Germany. [11]UGent Center for Molecular Modeling, Ghent University, Ghent, Belgium. [12]These authors contributed equally: Katharina Esther Julia Jungnickel, Océane Guelle. [13]These authors jointly supervised this work: Bruno Gasnier, Christian Löw, Markus Damme. ✉e-mail: bruno.gasnier@u-paris.fr; christian.loew@embl-hamburg.de; mdamme@biochem.uni-kiel.de

## Methods

The research in this manuscript complies with relevant ethical regulations. Mouse and *Xenopus* work was approved by the local German and French authorities, respectively.

### Chemicals

The peptides were purchased from Bachem or Sigma-Aldrich. All amino acids used belong to the L series. Most of the charged peptides were obtained as salts with the following counterions: hydrochloride (Ala–Lys, Lys–Pro and Lys–Val), hydrobromide (Lys–Ala), acetate (Arg–Ala, Lys–Ala–Ala and Pro–Arg) and nitrate (anserine); the chemicals and reagents were purchased, if not otherwise indicated, from Sigma-Aldrich. A complete list of peptides is depicted in Supplemental Table 2. Hydroxyproline-bound 2-chlorotrityl chloride (Barlos) resin and *N*-α-(9-fluorenylmethoxycarbonyl)-*N*-ω-(4-methoxy-2,3,6-trimethylbenzenesulfonyl)-L-arginine were obtained from Watanabe Chemicals. 6-aminoquinolyl-*N*-hydroxysuccinimidyl carbamate (AccQ) was purchased from Toronto Research Chemicals). Arginyl–hydroxyproline was synthesized according to the *N*-α-9-fluorenylmethoxycarbonyl strategy using a PSSM-8 peptide synthesizer (Shimadzu). Synthesized arginyl–hydroxyproline was purified by reversed-phase high-performance liquid chromatography (HPLC) using a Cosmosil 5C18-MS-II column (10 mm × 250 mm, Nacalai). A binary gradient was used with 0.1% formic acid (solvent A) and 0.1% formic acid containing 80% acetonitrile (solvent B) at a flow rate of 2.0 ml min$^{-1}$. The chemicals for leucine-5,5,5-$d_3$–alanine (Leu($d_3$)–Ala) synthesis, *tert*-butoxycarbonyl-leucine-5,5,5-$d_3$ (98%), HCl.alanine-O*t*Bu (99%) and (benzotriazol-1-yloxy)tripyrrolidinophosphonium hexafluorophosphate (PyBOP) were purchased from Cambridge Isotope Laboratories, Sigma and Novabiochem, respectively. The leucine-5,5,5-$d_3$ standard was from Cambridge Isotope Laboratories. BAY-069 was from MedChemExpress.

### Synthesis of Leu($d_3$)–Ala

Dipeptide Leu($d_3$)–Ala hydrochloride, as a mixture of two diastereoisomers, was synthesized in two steps by coupling Boc-Leu-5,5,5-$d_3$-OH with HCl.Ala-O*t*Bu using PyBOP as the coupling reagent[68], followed by the deprotection of the protecting groups in acidic conditions, as shown in the following scheme:

### General synthesis protocol for Leu($d_3$)–Ala

All reactions were carried out under an argon atmosphere with anhydrous solvent and were monitored by thin layer chromatography (TLC) with silica gel Merck 60 F254 on aluminium sheets. Automated flash chromatography was performed using a Biotage apparatus with evaporative light scattering detection and ultraviolet detectors using a Buchi FlashPure silica column. The solvent systems were given according to (s/s: v/v). The $^1$H (500.16 MHz), $^{13}$C (125.78 MHz) and nuclear magnetic resonance (NMR) spectra were recorded on a 500 Bruker spectrometer equipped with a sensitivity-optimized measurement head (cryoprobe). Chemical shifts ($\delta$, ppm) are given with reference to deuterated solvents for $^1$H and $^{13}$C NMR, respectively: CDCl$_3$ (7.24, 77.23) and D$_2$O (4.78). The signal multiplicity is described as follows: s (singlet), d (doublet), t (triplet), q (quartet), quin (quintuplet) and m (multiplet). Broad signals are described as br. The coupling constants ($J$) are given in hertz. The greek letters are used as locants for NMR attributions, which were established on the basis of $^{13}$C using $^1$H decoupled spectra as well as COSY, HSQC and HMBC.

### Synthesis of *tert*-butyl (*tert*-butoxycarbonyl)-leucyl-5,5,5-$d_3$-alaninate

To a cooled solution of Boc-leucine-5,5,5-$d_3$ (469.0 mg, 2.0 mmol, 1.0 equiv.), HCl.alanine-O*t*Bu (550.50 mg, 3.0 mmol, 1.5 equiv.),

PyBOP (1.25 g, 2.4 mmol, 1.2 equiv.) in dimethylformamide (9.6 ml), *N*,*N*-diisopropylethylamine (1.4 ml, 8.0 mmol, 4 equiv.) was added slowly. The reaction mixture was stirred at room temperature overnight, diluted with ethyl acetate (10 ml for 1 ml of dimethylformamide (DMF)) and then extracted with a cooled solution of 5% aqueous KHSO$_4$ (2×), saturated NaHCO$_3$ (2×) and brine (2×). The organic layer was then dried with Na$_2$SO$_4$, filtered and evaporated under vacuum to give the product after purification by flash chromatography (cyclohexane/ethyl acetate: 90/10) as a white solid in 51% yield (370 mg, 1.02 mmol). $^1$H NMR in CDCl$_3$ showed the presence of two rotamers due to the Boc group (80/20).

$^1$H NMR (500 MHz, CDCl$_3$) $\delta$: 7.06 (d, $J$ = 6.0 Hz, 0.8H, NH-Ala), 6.74 (brs, 0.2H, NH-Ala), 5.69 (brs, 0.2H, NH-Boc), 5.32 (d, $J$ = 8.5 Hz, 0.8H, NH-Boc), 4.28 (quin, $J$ = 7.0 Hz, 1H, Hα-Ala), 4.12 (m, 0.8H, Hα-Leu), 3.89 (brs, 0.2H, Hα-Leu), 1.59 (m, 1H, Hγ-Leu), 1.55–1.38 (m, 2H, Hβ-Leu), 1.34 (s, 9H, CO$_2$*t*Bu), 1.31, 1.30 (2 s, 9H, Boc), 1.22 (d, $J$ = 7.5 Hz, 3H, Hβ-Ala), 0.82, 0.80 (2d, $J$ = 6.0 Hz, 3H, Hδ-Leu); $^{13}$C NMR (500 MHz, CDCl$_3$) $\delta$:172.6, 172.5 (CONH), 171.9 (CO$_2$*t*Bu), 155.9 (CO-NH-Boc), 81.5, 81.4 (Cq-NH-Boc), 79.5 (Cq-CO$_2$*t*Bu), 52.9 (Cα-Leu), 48.6 (Cα-Ala), 41.5 (Cβ-Leu), 28.3, 27.9 (CH$_3$-*t*Bu), 24.4 (Cγ-Leu), 23.0, 21.8 (Cδ-Leu), 18.0, 17.9 (Cβ-Ala).

### Synthesis of leucyl-5,5,5-$d_3$-alanine hydrochloride (LSP11-280723)

To a solution of Boc-Leu($d_3$)–Ala-O*t*Bu (120.0 mg, 0.33 mmol) in dioxane (0.25 ml) at 0 °C was added slowly a solution of HCl 4 M in dioxane (2.5 ml). After 30 min at this temperature, the reaction mixture was stirred at room temperature overnight. Evaporation of the solvent under vacuum and recrystallization with MeOH–Et$_2$O afforded HCl·leucyl-5,5,5-$d_3$-alanine as a white solid (66.5 mg, 0.275 mmol) in 83% yield.

$^1$H NM (500 MHz, D$_2$O) $\delta$: 4.36 (q, $J$ = 7.0 Hz, 1H, Hα-Ala), 3.93 (t, $J$ = 7.0 Hz, 1H, Hα-Leu), 1.77–1.61 (m, 3H, Hγ-Leu, Hβ-Leu), 1.40 (d, $J$ = 7.0 Hz, 3H, Hβ-Ala), 0.92, 0.90 (2d, $J$ = 6.0 Hz, 3H, Hδ-Leu); $^{13}$C NMR (500 MHz, D$_2$O) $\delta$:176.0 (CO$_2$H), 170.1 (CONH), 51.6 (Cα-Leu), 48.8 (Cα-Ala), 39.7 (Cβ-Leu), 23.4 (Cγ-Leu), 21.5, 20.9 (Cδ-Leu), 15.9 (Cβ-Ala).

### Cell lines, mouse strains and antibodies

*Mfsd1*-knockout mice (C57Bl/6N-Mfsd1$^{tm1dHhtg/Damme}$; age 6 months, male and female) were described previously[43]. Mice were housed under standard laboratory conditions with a 12 h light–dark cycle and constant room temperature and humidity. Food and water were available ad libitum. Expi293F cells were purchased from Thermo Fisher (A14527). MEFs from *Glmp*-knockout mice were described previously[43]. The cell lines were not authenticated, and no commonly misidentified cell line was used.

The antibodies used throughout the study included: LAMP1 clone 1D4B (rat monoclonal, Developmental Studies Hybridoma Bank; 1:1,000); LAMP1 clone 1D4B (rat monoclonal, conjugated to AlexaFluor 647, BioLegend; 1:25); HA clone 3F10 (rat monoclonal, Sigma-Aldrich/Merck); HA clone 3F10 (rat monoclonal, conjugated to fluorescein isothiocyanate, Sigma-Aldrich/Merck; 1:50), GFP (mouse monoclonal, Roche Molecular Biochemicals; 1:1,000), mKate2 (rabbit polyclonal, Origene), KDEL (mouse monoclonal, Enzo Life Sciences; 1:500), Cox IV (rabbit polyclonal, ab16056, Abcam; 1:1,000) and Golgin 97 (clone CDF4, mouse monoclonal, Thermo Scientific Fisher; 1:500). The antibody against cathepsin D was custom made against a synthetic peptide (CKSDQSKARGIKVEKQIFGEATKQP) and immunization of rabbits, followed by affinity purification against the immunization peptide and used in a 1:2,000 dilution. The custom-made MFSD1- and GLMP-specific antibodies were described before[43] and used in a 1:3,000 dilution (MFSD1) or 1:1,000 dilution (GLMP).

### Cell culture and transfection of eukaryotic cells

For transfection of MEF cells, 1–5 µg of DNA were incubated with polyethylenimine in Dulbecco's modified Eagle medium (without antibiotics nor foetal bovine serum) for 15 min at room temperature.

The mix was applied to the culture of cells, and after ~6 h, the medium was exchanged. The transfected cells were analysed 48 h post-transfection.

## Cloning of cDNA constructs for oocyte expression

Lysosomal sorting motif mutations, Y400A and L11A/L12A, were introduced into mouse GLMP and MFSD1 plasmids, respectively, using the Q5 Site-Directed Mutagenesis Kit (New England Biolabs). The whole coding sequence was verified by automated sequencing. mGLMP$_{Y400A}$-mKate2 and mMFSD1$_{L11A/L12AA}$-EmGFP cDNAs were then subcloned into the pOX(+) vector for *Xenopus* oocyte expression. In this vector, the cDNA of interest is flanked by the 5′- and 3′-noncoding sequences from *Xenopus laevis* β-globulin mRNA to increase expression.

## Cloning, expression and purification of MFSD1, GLMP and GLMP–MFSD1-fusion protein for recombinant expression

The gene encoding mouse MFSD1 (Uniprot Q9DC37) was cloned into a pXLG vector[69] containing an N-terminal Twin-Strep-tag followed by a human rhinovirus 3C cleavage site, referred to as MFSD1-strep. The encoding sequence of mouse GLMP (Uniprot Q9JHJ3) was cloned into the pXLG vector containing a C-terminal tobacco etch virus cleavage site and GFP tag, followed by an 8×histidine (8×His)-tag, termed GLMP-Ct-His-GFP. A fusion construct of mouse GLMP and mouse MFSD1 connected by a linker region (GSAGSAAGSGEF), termed GLMP–MFSD1-strep, was inserted into a pXLG vector with a C-terminal 3C-protease cleavage site followed by a Twin-Strep-tag. The Expi293F cells were transiently transfected as described elsewhere[70], and the cells were collected 48 h post-transfection. MFSD1-strep, co-expressed MFSD1-strep and GLMP-Ct-His-GFP, referred to as GLMP + MFSD1, and GLMP–MFSD1-strep, referred to as GLMP–MFSD1, proteins were directly purified from the cell pellet by standard affinity purification. Briefly, the cell pellets were solubilized for 1 h at 4 °C in buffer containing 1× phosphate-buffered saline (PBS) pH 7.4, 150 mM NaCl, 5 mM MgCl$_2$, 5% glycerol, 1% (w/v) *n*-DDM (Anatrace) detergent, 0.1% (w/v) CHS (Anatrace), 20 U ml$^{-1}$ DNase I and EDTA-free protease inhibitors (Roche). The sample was centrifuged for 30 min at 35,000*g*, and the supernatant was directly applied to Strep-TactinXT beads (IBA), incubated for 1 h at 4 °C and loaded onto a gravity column. The beads were washed with 20 column volumes (CV) of washing buffer (1× PBS pH 7.4, 150 mM NaCl, 0.03% DDM and 0.003% CHS) before elution with 3 CV of size exclusion (SEC) buffer (20 mM HEPES pH 7.5, 150 mM NaCl, 0.03% DDM and 0.003% CHS) containing 10 mM desthiobiotin.

For GLMP + MFSD1, the elution fraction from the strep-tactin purification was incubated with Ni-NTA beads for 1 h at 4 °C and loaded onto a gravity column. The beads were washed with 10 CV of SEC buffer before elution with 3 CV of SEC buffer containing 250 mM Imidazole. Tobacco etch virus protease was added to the elution fraction, and the mixture was dialysed against SEC buffer. The dialysed sample was again incubated with Ni-NTA beads for 30 min at 4 °C and loaded onto a gravity column, and the flow-through was collected and combined with that of one washing step of 2 CV of SEC buffer. The sample was then concentrated, as were the elution fractions of strep-tactin affinity purification of MFSD1-strep and GLMP–MFSD1-strep. The concentrated samples were applied onto either a Superose 6 increase 3.2/300 (Cytiva), in the case of GLMP–MFSD1 and GLMP + MFSD1, or a Superdex200 5/150 (Cytiva) column for MFSD1 sample. For all samples, the columns were equilibrated in SEC buffer (20 mM HEPES pH 7.5, 150 mM NaCl, 0.03% DDM and 0.003% CHS). For cryo-EM sample preparation, the SEC buffer contained 20 mM HEPES pH 7.5, 150 mM NaCl, 0.015% DDM and 0.0015% CHS.

## Cloning and characterization of MFSD1 mutants for recombinant expression

Binding-site mutations within the MFSD1 gene were generated via amplification of the mMFSD1 gene in combination with primers carrying the respective mutations, followed by SLiCE cloning[71] of the amplified gene into a pXLG vector. For initial expression tests, the mutants and wildtype MFSD1 were cloned with an additional N-terminal 8×His and GFP tag. Expression levels of each mutant were assessed by fluorescent SEC chromatography in comparison with the expression level of wildtype MFSD1. For this, the cell pellet of a 10 ml Expi293F culture overexpressing MFSD1 wildtype or mutant was solubilized in 1× PBS pH 7.4, 150 mM NaCl, 5 mM MgCl$_2$, 5% glycerol, 1% (w/v) DDM detergent, 0.1% (w/v) CHS, 20 U ml$^{-1}$ Dnase I and EDTA-free protease inhibitors (Roche) for 1 h at 4 °C. This was followed by ultracentrifugation at 100,000*g* for 1 h at 4 °C using a MLA130 rotor. The supernatant was then loaded onto a Superose 6 5/150 home-packed column, equilibrated in SEC buffer, monitoring the EGFP-fluorescence at $\lambda_{excitation}$ = 488 nm/$\lambda_{emission}$ = 510 nm. Based on the expression and solubilization screening results, the selected mutants were cloned into the pXLG vector carrying only an N-terminal Twin-Strep-tag. The mutants were expressed and purified as wildtype MFSD1.

## LC–MS/MS-based analysis of dipeptides from tissues

**Sample preparation.** An aliquot of the liver (approximately 150 mg) was homogenized with PBS (150 μl) in a Biomasher II (Nippi, Tokyo, Japan). The homogenate was mixed with 900 μl of ethanol. The ethanol (75%) suspension was centrifuged at 10,000*g* for 5 min after strong agitation. The supernatants were used for further analysis.

## Derivatization with AccQ

Aliquots (100 μl) of 75% ethanol soluble fractions and peptide standards (1 mM and 20 μl) were dried under vacuum and dissolved into 80 μl of 50 mM sodium borate buffer, pH 8.8. Then, 20 μl of AccQ acetonitrile solution (0.3%; AccQ powder dissolved in acetonitrile giving 3 mg ml$^{-1}$) was added and kept at 50 °C for 10 min. The reaction mixture was mixed with 100 μl of 5 mM sodium phosphate buffer, pH 7.5, and used as a sample for LC–MS/MS. For the standard, the reaction mixture was further diluted to 1/10.

## LC–MS/MS analyses

Aliquots (10 μl) of AccQ derivatives of standard peptide were injected into an electron spray ionization tandem mass spectrometer (LCMS-8040, Shimadzu, Kyoto, Japan) without using a column. Multiple-reaction monitoring (MRM) conditions for each AccQ-peptide were optimized using LaboSolution LCMS v5.5 (Shimadzu) after the detection of singly and doubly charged ions.

Each peptide was determined by reversed-phase high-performance liquid chromatography-electron spray ionization tandem mass spectrometer equipped with an Inertsil ODS 3 column (2.1 mm × 250 mm, GL Science). A binary gradient was carried out at a flow rate of 0.2 ml min$^{-1}$. The gradient program was as follows: 0–15 min, 0–50% B; 15–20 min, 50–100% B; 20–25 min, 100% B; 25.01–35, 0% B. Detection was carried out in MRM mode. For the sample and standard, 20 and 1 μl were injected, respectively.

## Thermal stability measurements

The unfolding of individual target proteins was followed by the nanoDSF method[72]. Purified wildtype and mutant MFSD1 or GLMP–MFSD1 and GLMP + MFSD1 was diluted to 0.2 mg ml$^{-1}$ into nanoDSF buffer containing 100 mM HEPES pH 7.5, 150 mM NaCl, 0.03% DDM and 0.003% CHS. 50 mM ligand stock solutions were prepared in 100 mM HEPES pH 7.5 buffer. The transporter was incubated at a ligand concentration of 5 mM at room temperature for 30 min before starting the nanoDSF measurement using a Prometheus NT.48 device. The measurements were performed in a temperature range from 20 °C to 95 °C in 1 °C min$^{-1}$ increments. The melting temperatures were determined by the Nanotemper software and plotted using GraphPad Prism. Estimation of $K_D$ was performed as described in Kotov et al.[35].

## Reconstitution of MFSD1 into liposomes

For the liposome-based uptake assays, GLMP–MFSD1, GLMP + MFSD1 wildtype MFSD1 and MFSD1 mutants were reconstituted into liposomes containing 1-palmitoyl-2-oleoyl-sn-glycero-3-phosphoethanolamine (POPE, Avanti Polar Lipids), P1-palmitoyl-2-oleoyl-sn-glycero-3-phospho-1′-rac-glycerol (POPG, Avanti Polar Lipids) and CHS (Anatrace) in a 3:1:1 (w/w) ratio. Lipids were mixed in chloroform, and the solvent was removed using a rotary evaporator. The dried lipids were washed twice with pentane, followed by solvent removal. The lipid film was resuspended in reconstitution buffer (50 mM potassium phosphate, pH 7.0) to a final lipid concentration of 20 mg ml$^{-1}$. On the day of the reconstitution, the lipids were diluted to 5 mg ml$^{-1}$ in reconstitution buffer and extruded through a 400 nm filter unit (Avanti). The preformed liposomes were disrupted with a final concentration of 0.075 % (w/w) Triton X-100 and incubated for 10 min at room temperature. Protein at a concentration of 0.5 mg ml$^{-1}$, or similar amounts of SEC buffer (empty control), was added to the lipids to reach a protein:lipid ratio of 1:60 (w/w), and the mixture was incubated at 4 °C for 1 h. The detergent was removed by sequentially adding Bio-Beads SM-2 (Bio-Rad) and incubating overnight at 4 °C. The mixture was collected, and the liposomes were resuspended in a reconstitution buffer, flash-frozen three times in liquid nitrogen and then stored at −80 °C until further use.

## Liposome-based pyranine assays

For the liposome-based uptake assays[73], the liposomes were thawed and collected using a total amount of 5 μg of protein per experiment. The pelleted liposomes were resuspended in uptake buffer 1 (5 mM HEPES pH 6.8, 150 mM KCl and 2 mM MgSO$_4$) containing 1 mM pyranine. The resuspended liposomes were subjected to seven freeze–thaw cycles in liquid nitrogen before being extruded through a 400 nm filter unit (Avanti Polar Lipids) and then collected. Excess pyranine was removed using a G-25 spin column (Cytiva) equilibrated in uptake buffer 1. The liposomes were again collected and resuspended in uptake buffer 1 to a final volume of 4 μl per experiment.

Pyranine-loaded liposomes were diluted 1:50 into uptake buffer 2 (5 mM HEPES pH 6.8, 120 mM NaCl and 2 mM MgSO$_4$) in a black, chimney-style, flat-bottom 96-deep-well plate (Greiner). The fluorescence of pyranine was measured at excitation wavelengths of 415 nm and 460 nm, with an emission wavelength of 510 nm for both excitations using a TECAN Spark2000 operating at 22 °C. A peptide or buffer was added after a short equilibration period to a final concentration of 2.5 mM. The uptake reaction was initiated after the addition of val at a final concentration of 1 μM. For analysis, the fluorescent counts at $\lambda_{ex}$ = 415 nm/$\lambda_{em}$ = 510 nm were divided by the fluorescent counts at $\lambda_{ex}$ = 460 nm/$\lambda_{em}$ = 510 nm. The average value of the first 25 s after the addition of peptide was used for normalization, and the normalized counts were plotted against the assay time using Prism GraphPad. For bar graphs and $K_M$ measurements, the initial uptake velocity in the linear range of the uptake curve after the addition of val was determined by linear regression using Prism GraphPad.

## Expression of MFSD1 and GLMP in *Xenopus* Oocytes

*Xenopus* oocytes were either purchased from Ecocyte Bioscience or prepared from frogs housed in the local animal facility in compliance with the European Animal Welfare regulations (ethical agreement APAFiS no. 14316-2017112311304463 v4). Ovarian lobes were extracted from *Xenopus laevis* females under anaesthesia, and the oocyte clusters were incubated on a shaker in OR2 medium (85 mM NaCl, 1 mM MgCl$_2$, 5 mM Hepes-K$^+$ pH 7.6) containing 2 mg ml$^{-1}$ of collagenase type II (GIBCO) for 1 h at room temperature. The defolliculated oocytes were sorted and kept at 19 °C in Barth's solution (88 mM NaCl, 1 mM KCl, 2.4 mM NaHCO$_3$, 0.82 mM MgSO$_4$, 0.33 mM Ca(NO$_3$)$_2$, 0.41 mM CaCl$_2$ and 10 mM Hepes–Na$^+$ pH 7.4), supplemented with 50 μg ml$^{-1}$ of gentamycin.

Capped mRNAs were synthesized in vitro from the linearized pOX(+) plasmids using the mMessage-mMachine SP6 kit (Invitrogen). Unless stated otherwise, defolliculated oocytes were injected with both mGLMP$_{Y400A}$-mKate2 mRNA and mMFSD1$_{L11A/L12A}$-EmGFP mRNA (25 ng each at 1 μg μl$^{-1}$). For co-expression with PQLC2, the oocytes were injected with these two mRNAs and an mRNA-encoding rat PQL-C2$_{L11A/L12A}$-EGFP[13] at 16 ng each. The non-injected oocytes were used as negative controls.

## Cell surface biotinylation

Two days after injection, five oocytes were washed twice with ice-cold PBS and biotinylated for 20 min at 4 °C using 2.5 mg ml$^{-1}$ of the membrane-impermeable, cleavable reagent sulfo-NHS-SS-biotin EZ-LinkTM (Thermo Scientific). After four washes, the oocytes were lysed for 30 min in 500 μl lysis buffer (150 mM NaCl, 5 mM EDTA, 50 mM Tris–HCl pH 7.5, 0.1% SDS, 1% Triton X-100 and Halt Protease Inhibitor Cocktail). The cell lysates were clarified by sedimentation at 14.000$g$ for 10 min, and the supernatant was incubated for 2 h at 4 °C with 150 μl streptavidin-agarose beads (Sigma-Aldrich) under gentle agitation. Beads were sedimented at 100$g$ for 30 s. The supernatants (unbound material) were recovered, and the beads were washed three times with 1 ml lysis buffer. Streptavidin-bound material was then eluted in 100 μl Laemmli's sample buffer. Half of the bound proteins were resolved by 10% SDS–polyacrylamide gel electrophoresis (SDS–PAGE). Following electrophoresis, transfer and blocking, the nitrocellulose membrane was incubated with mouse anti-GFP antibodies (1:1,000, Roche Molecular Biochemicals) and rabbit anti-mKate2 antibodies (1:2,000, Origene). The protein bands were obtained using horseradish peroxidase-conjugated antibodies against mouse whole immunoglobulins and horseradish peroxidase-conjugated antibodies against rabbit whole immunoglobulins (1:10,000, Sigma-Aldrich) as secondary antibodies and detection with the Lumi-Light Plus Western Blotting Substrate (Roche). The images were acquired and quantitated with an ImageQuant LAS 4000 chemiluminescence imager (GE Healthcare).

## TEVC recording in *Xenopus* oocytes

Electrophysiological recordings were done at room temperature (20 °C), usually 2 days after complementary RNA injection. For each experiment, mMFSD1$_{L11A/L12A}$-EmGFP expression at the plasma membrane was verified under an Eclipse TE-2000 epifluorescence microscope (Nikon) with a 4× objective focused at the equatorial plane. The voltage clamp was applied with two borosilicate-glass Ag/AgCl microelectrodes filled with 3 M KCl (from 0.5 to 3 MΩ tip resistance) connected to an O725C amplifier (Warner Instrument) and a Digidata 1440A interface controlled via Clampex v.11.2 software (Molecular Devices). The currents were filtered using a 10 Hz low-pass filter and sampled at 1 kHz. The solutions were applied with a gravity-fed perfusion system in a Xenoplace recording chamber (ALA Scientific Instruments) with built-in Ag/AgCl reference electrodes. The oocytes were perfused in ND100 medium (100 mM NaCl, 2 mM KCl, 1 mM MgCl$_2$ and 1.8 mM CaCl$_2$) buffered with 10 mM 2-(N-morpholino)ethanesulfonic acid–NaOH to pH 5.0 unless stated otherwise. After recording a stable baseline current, peptides (10 mM unless stated otherwise) were applied in this medium and eventually washed to measure the evoked current. For peptides purchased as hydrochloride salts, the substrate-free solution was supplemented with N-methyl-D-glucamine hydrochloride (Sigma-Aldrich) at the same concentration to avoid interference with the Ag/AgCl reference electrode. For Lys–Ala application, the substrate-free solution was supplemented with the same concentration of sodium bromide (Merck) to avoid an endogenous current artefact induced upon bromide washing. The data were analysed with Clampfit v.11.2 (Molecular Devices).

## Combined TEVC and pH$_{in}$ recording in *Xenopus* oocytes

In these experiments, a third ion-selective electrode connected to an FD223a dual channel differential electrometer (World Precision

Instruments) was impaled into the oocyte. The signals were digitized with the Digidata 1440A of the TEVC setup and acquired via Clampex v.11.2. To prepare this $pH_{in}$ electrode, a silanized micropipette with dichlorodimethylsilane (Sigma) was tip filled with a proton ionophore (hydrogen ionophore I, cocktail B, Sigma-Aldrich) and backfilled with 150 mM NaCl, 40 mM $KH_2PO_4$ and 23 mM NaOH pH 6.8. The two channels of the FD223a electrometer were connected to the pH electrode and the voltage ground electrode of the TEVC setup, respectively. The potential difference between the two inputs tested in diverse buffers (pH range 5.0–7.5) was proportional to pH with a mean slope of $-59 \pm 8.6$ mV ($n = 3$). The relative level of substrate-evoked intracellular acidification was quantified by the slope, in milivolts per second, of the ion-selective electrode voltage trace. The data were analysed with Clampfit v.11.2.

## Leu($d_3$)–Ala uptake into *Xenopus* oocytes

Two days after complementary RNA injection, the oocytes were washed and individually incubated in 200 µl ND100 pH 5.0 medium supplemented, or not, with 10 mM Leu($d_3$)–Ala for 20 min at room temperature. After three washes in 0.55 ml ice-cold ND100 pH 5.0 medium, the oocytes were transferred into 100 µl ice-cold methanol/water (50:50) and homogenized by pipetting up and down with a P1000 tip. After centrifugation for 5 min at 4 °C and 16,000*g*, the supernatants were collected and stored at −20 °C before analysis. In experiments for absolute quantification of the Leu($d_3$)–Ala flux, a subset of MFSD1–GLMP oocytes was treated before (5 min) and during the transport reaction with 10 µM of the branched-chained amino acid transaminase inhibitor BAY-069 (ref. [74]) to prevent metabolization of the accumulated leucine. Quantification of dipeptides and amino acids in oocyte extracts was done by LC–MS/MS. Lysis supernatants were diluted 20-fold in water, and 20 µl of the dilution were injected into a reverse phase column (Phenomenex-C18, 2.1 × 150 mm; 3 µm). For experiments with unlabelled peptides, the supernatants were diluted tenfold to improve the detection of some amino acids. The mobile phases were water with 0.1% formic acid for phase A and acetonitrile with 0.1% formic acid for phase B. Elution was programmed to start at 100% phase A for 3 min, then fall to 20% phase A at 10 min, return to 100% phase A at 11 min and equilibrate for 6 min before the next sample injection. The flow rate was 0.3 ml min⁻¹, and the detection was done using an 8060NX triple quadrupole mass spectrometer (Shimadzu) with an electrospray ion probe operated at 250 °C. The selected ions monitored (SIM) and MRM are listed with the retention times in Supplementary Table 3. Quantification was done by integrating the chromatographic peak area using Labsolution v.5.118 software (Shimadzu). For absolute quantification, a calibration curve was established with various known concentrations (from 0.2 to 100 µM) of Leu($d_3$), Ala and Leu($d_3$)–Ala standards. We used the 46.15 and 44.10 MS/MS fragment ions as quantifiers for Leu($d_3$) and Ala, respectively.

## Cryo-EM sample preparation and data collection

Gel filtration peak fractions containing GLMP–MFSD1 were used for cryo-EM sample preparation. For the apo state structure, grids at a concentration of 3.33 mg ml⁻¹ purified GLMP–MFSD1 were prepared. For GLMP–MFSD1 in the presence of the dipeptide His–Ala, termed GLMP–MFSFD1_His–Ala, purified GLMP–MFSD1 at 3 mg ml⁻¹ was dialysed over night against buffer containing 150 mM NaCl, 0.015% (w/v) DDM and 0.0015% (w/v) CHS supplemented with 20 mM His–Ala. A total of 3.6 µl of purified protein were applied onto glow-discharged holey carbon-coated grids (Quantifoil R1.2/1.3 Au 300 mesh) and blotted for 3.5 s with a blot force of 0 at 100% humidity and 4 °C before being frozen in liquid ethane using a Vitrobot Mark IV (Thermo Fisher Scientific). The data were collected in counted super-resolution mode, with a binning of 2, on a Titan Krios (Thermo Fisher Scientific) equipped with a K3 camera and a BioQuantum energy filter (Gatan) set to 15 eV. For the two datasets, movies were collected at a nominal magnification

of ×81,000 with a pixel size of 1.1 Å using the EPU software (Thermo Fisher Scientific). For the GLMP–MFSD1_apo structure, two separate data sets were collected, consisting of 3,179 and 2,551 movies. For the GLMP–MFSD1_His–Ala structure, one data set consisting of 3,193 movies was collected. For both GLMP–MFSD1_apo and GLMP–MFSD1_His–Ala, data were collected at a dose rate of 15 e⁻ per pixel per second, with an exposure time between 4 and 4.2 s to reach a total dose of 55 e⁻ Å⁻².

## Cryo-EM data processing and modelling

The data processing was performed in cryoSPARC[75]. The collected movies were subjected to patch motion correction with a maximum alignment resolution of 4 Å. After CTF estimation using CTFFIND4 (ref. [76]), micrographs were curated based on CTF fit resolution and total full-frame motion. The particles were selected using Blob picker with a minimum particle diameter of 100 Å and a maximum particle diameter of 200 Å, followed by manual inspection and adjustment of the NCC score (>0.49) and local power to reduce duplicate particle picks and picking of ethane contaminations on the sample. The particles were extracted with a 256-pixel (GLMP–MFSD1_apo) and 300-pixel (GLMP–MFSD1_His–Ala) box size, followed by several rounds of two-dimensional classification. Particles of the final two-dimensional classification were subjected to ab initio reconstruction of four classes. Upon visual inspection, two reconstructions, one representing the 'model class' and the other one a 'decoy class', depicting a corrupted model, were selected for heterogeneous refinement of the whole particle stack used in the previous ab initio reconstruction. The resulting 'model class' after heterogeneous refinement was then subjected to non-uniform refinement[77], and the resulting reconstruction was used for another round of heterogeneous refinement while the 'decoy class' stayed the same. After several rounds of these two steps, per particle local motion correction[78] was performed, followed by one more non-uniform refinement step that resulted in maps of a final global resolution of 4.2 Å and 4.1 Å for GLMP–MFSD1_apo and GLMP–MFSD1_His–Ala, respectively.

The initial model fitting was performed in UCSF ChimeraX (ref. [79]). A first model of MFSD1, representing an inward-open conformation in complex with GLMP, was obtained by AlphaFold2 Multimer[80], using both protein sequences as input. First, the model of MFSD1 was manually placed into the experimental density, and the fit was refined in UCSF ChimeraX (ref. [79]). Then, the model was refined in Cartesian space using the Rosetta/StarMap workflow[81] with the map resolution set to 7.5 Å. Next, the GLMP model was manually placed into the density, followed by fit refinement in UCSF ChimeraX, and the complex model was refined again with the same settings in Rosetta/StarMap. The model with the highest iFSC metric of 0.64 (ref. [82]) was selected for downstream analyses. The model was further fit into the map with ISOLDE[83]. The subsequent model building and refinement were iteratively performed in Coot[84] and PHENIX[85]. The figures were generated using PyMOL and UCSF ChimeraX. The electrostatic surfaces were generated using the APBS plugin provided in PyMOL[86].

## MD simulation of ligand-bound MFSD1

The MFSD1 structures were placed in a heterogeneous bilayer composed of POPE (20%), 1-palmitoyl-2-oleoyl-glycero-3-phosphocholine (30%), cholesterol (30%) and *N*-palmitoyl-sphingomyelin (20%) using CHARMM-GUI scripts[87]. The protonation states of titratable residues were determined using the MCCE program[88]. For the substrates, both termini are assigned charged. In the case of the dipeptide His–Ala, both neutral and charged side chains were simulated. All systems were hydrated with 150 mM NaCl electrolyte. The all-atom CHARMM36m force field was used for lipids, ions, cofactors and protein with TIP3P water. MD trajectories were analysed using visual MDs[89] and MDAnalysis[90].

All simulations were performed using GROMACS 2021.3. A description of the dipeptide simulations performed for this study is provided in Extended Data Fig. 8a. The conditions and substrates for MD analyses

are summarized in Supplementary Table 5. The starting systems were energy-minimized for 5,000 steepest descent steps and equilibrated initially for 500 ps of MD in a canonical (NVT) ensemble and later for 7.5 ns in an isothermal–isobaric (NPT) ensemble under periodic boundary conditions. During equilibration, the restraints on the positions of non-hydrogen protein atoms of initially 4,000 kJ mol$^{-1}$ nm$^{-2}$ were gradually released. Particle-mesh Ewald summation with cubic interpolation and a 0.12 nm grid spacing was used to treat long-range electrostatic interactions. The time step was initially 1 fs and was increased to 2 fs during the NPT equilibration. The LINCS algorithm was used to fix all bond lengths. The constant temperature was established with a Berendsen thermostat, combined with a coupling constant of 1.0 ps. A semi-isotropic Berendsen barostat was used to maintain a pressure of 1 bar. During production runs, a Nosé–Hoover thermostat and a Parrinello–Rahman barostat replaced the Berendsen thermostat and barostat. The analysis was carried out on unconstrained simulations.

### Indirect immunofluorescence and microscopy
Semi-confluent cells were grown on glass coverslips and fixed 48 h after transfection for 20 min with 4% (w/v) paraformaldehyde at room temperature. The cells were permeabilized, quenched and blocked with normal goat serum before incubation with directly fluorophore-conjugated primary antibodies overnight at 4 °C. After washing, the coverslips were washed four times and mounted on microscope slides with mounting medium including 4-,6-diamidino-2-phenylindole. An Airyscan2 980 laser scanning microscope (Zeiss) equipped with a 63× oil immersion objective (numerical aperture (NA) of 1.40) was used for microscopy. The images were acquired and processed with the Zen 3.2 (Blue edition) software. The Pearson correlation coefficient was calculated with the Zen 3.2 (Blue edition) software.

### SDS–PAGE and immunoblotting
SDS–PAGE and immunoblotting were performed according to standard protocols. The protein lysates were transferred to the nitrocellulose membrane by semi-dry blotting. For MFSD1-immunoblotting, lysates were denatured for 10 min at 55 °C before SDS–PAGE. The protein bands were detected using horseradish peroxidase-conjugated and detection with the Lumi-Light Plus Western Blotting Substrate (Roche). The luminescence was detected with an ImageQuant LAS 4000 chemiluminescence imager (GE Healthcare).

### Enrichment of lysosomal fractions from the mouse liver
Liver lysosome enrichment was performed according to a previously published method[37,43]. All animal experiments were approved by the local authorities: Ministerium fur Energiewende, Klimaschutz, Umwelt und Natur (V242-13648/2018). A total of 4 days before the experiment, the mice were injected intraperitoneally with 4 µl g$^{-1}$ body weight with 17% (v/v) tyloxapol diluted in 0.9% NaCl. The mice were killed in a $CO_2$-flooded chamber. The liver was removed immediately and homogenized in three volumes of isotonic 250 mM sucrose solution in a Potter–Elvejhem and a glass homogenizer (B. Braun type 853202) with five strokes. The homogenate was centrifuged for 10 min at 1,000g to remove unbroken cells and nuclei. The pellet was re-extracted in the same volume of 250 mM sucrose solution in the Potter–Elvejhem and centrifuged again. The supernatants were pooled (post-nuclear supernatant (PNS)) and transferred to ultracentrifugation tubes. In the first differential centrifugation step, the lysosomes and mitochondria were enriched by centrifugation of the pooled PNS at 56,500g for 7 min at 4 °C (Beckman–Coulter, 70 Ti fixed-angle rotor). The supernatant was removed, and the pellet was resuspended in 250 mM sucrose solution. The resuspended solution was centrifuged again for 7 min at 56,500g, and the supernatant was carefully discarded. The differential centrifugation was followed by a discontinuous sucrose gradient. The final pellet was resuspended in a volume of 3.5 ml sucrose solution with a density of $\rho$ = 1.21 and transferred into a new ultracentrifugation tube. This fraction

was carefully layered with a sucrose solution of a density of $\rho$ = 1.15 (3 ml), $\rho$ = 1.14 (3 ml) and $\rho$ = 1.06 (0.5 ml). The gradient was centrifuged for 2.30 h at 4 °C and 111,000g in a swing-out rotor (Beckman–Coulter, SW41). The brownish lysosome-enriched fraction (~1 ml) was collected from the interface between the $\rho$ = 1.14 and $\rho$ = 1.06 sucrose layers.

### Untargeted metabolomics and targeted metabolite quantitation
Three replicates of lysosome-enriched samples from each genotype were submitted for untargeted metabolomics. The polar metabolites were extracted using cold 80% methanol (v/v) with isotopically labelled amino acids (Cambridge Isotope Laboratories MSK-A2-S) as internal standards and profiled using a Thermo Fisher Scientific ID-X Tribrid mass spectrometer with an electrospray ion probe. Metabolite separation before mass spectrometry was achieved through HILIC, conducted using a MilliporeSigma SeQuant ZIC-pHILIC 150 mm × 2.1 mm column (1504600001) along with a 20 mm × 2.1 mm guard (1504380001). The mobile phases consisted of 20 mM ammonium carbonate and 0.1% ammonium hydroxide dissolved in 100% LC–MS-grade water (phase A) and 100% LC–MS-grade acetonitrile (phase B). The chromatographic gradient involved a linear decrease from 80% to 20% of phase B from 0 to 20 min, followed by a linear increase from 20% to 80% from 20 to 20.5 min and maintaining at 80% from 20.5 to 29.5 min. The LC flow rate and injection volume were set to 0.15 ml min$^{-1}$ and 1 µl, respectively. The solvent blanks were also injected. The mass spectrometer settings included Orbitrap resolution of 120,000, positive and negative ion voltages of 3,000 V and 2,500 V, respectively, an ion transfer tube temperature of 275 °C, a vaporizer temperature of 350 °C, an RF lens at 40%, an AGC target of $1 × 10^6$ and a maximum injection time of 80 ms. A full scan mode with polarity switching at an $m/z$ 70–1,000 was executed. The gas flowrates include: sheath, 40 U; aux, 15 U; and sweep 1 U. The internal calibration was achieved by EasyIC.

The metabolite samples were pooled by combining replicates for quality control and data-dependent MS/MS collection. The orbitrap resolution was set at 240,000; higher-energy collisional dissociation (HCD) stepped energies at 15%, 30% and 45%; an isolation window at 1 $m/z$; intensity threshold at $2 × 10^4$ and exclusion duration at 5 s; AGC target at $2 × 10^6$; and maximum injection time at 100 ms. Both isotope and background exclusions were enabled, with background exclusion being performed via AcquireX (ThermoFisher Scientific). TraceFinder (Thermo Fisher Scientific) was used in combination with a library of known metabolite standards (MSMLS, Sigma-Aldrich) for targeted metabolite quantification. The mass tolerance for extracting ion chromatograms was set at 5 ppm.

### Statistics and reproducibility
GraphPad Prism 9.3.1 (GraphPad Software) was used for data representation and calculation of statistic testing. The statistic test applied for each graph is indicated in the figure legends. For most panels, a two-tailed paired $t$-test, a two-tailed unpaired $t$-test or a non-parametric Mann–Whitney $U$ test was used. The statistical differences in the graphs were generally depicted as ns, not significant and *$P ≤ 0.05$, **$P ≤ 0.01$ and ***$P ≤ 0.001$. The error bars in the graphs represent the standard error of the mean (s.e.m.) or standard deviation (s.d.), as indicated in the figure legends. If representative images are shown, the numer of replicates is given in the figure legends. No statistical method was used to predetermine sample size. The data were only excluded if obvious technical problems occurred during the experiments. Generally, no data were excluded. The samples were not randomized for this study because all experiments were internally controlled. The investigators were not blinded to allocation during experiments and outcome assessment.

### Reporting summary
Further information on research design is available in the Nature Portfolio Reporting Summary linked to this article.

## Data availability

The electron microscopy data and fitted models for GLMP–MFSD1 have been deposited in the Electron Microscopy Data Bank under accession code EMD-19005 and the PDB under accession code 8R8Q. The raw data used for data plotting are available as a supplementary table (numerical source data). The crystal structure of GLMP used for comparative analysis in this study can be found in the PDB under accession code 6NYQ AlphaFold2 predictions of MFSD1 as well as the models of MFSD1 and GLMP–MFSD1 after 500 ns of MD simulations and metabolomics raw data were deposited to Zenodo (Alphafold models: https://doi.org/10.5281/zenodo.10276738; MFSD1 apo/with ligands in initial poses and after 500 ns MD: https://doi.org/10.5281/zenodo.10276760). All protein sequences used in this study are publicly available at Uniprot (https://www.uniprot.org/). The metabolomics data are available at https://doi.org/10.5281/zenodo.10839783. Source data are provided with this paper. All other data supporting the findings of this study are available from the corresponding authors on reasonable request.

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

## Acknowledgements

S. Held is acknowledged for excellent technical assistance. We thank the BioMedTech animal facility at Université de Paris (CNRS UMS2009, INSERM US36) for housing the frogs and the 'Service de spectrométrie de masse de l'UMR8601' for mass spectrometry analysis. BioRender was used to create parts of the figures. M.A.-R. is a Stanford Terman Fellow and a Pew-Stewart Scholar for Cancer Research, supported by the Pew Charitable Trusts and the Alexander and Margaret Stewart Trust. K.E.J.J. and V.K. were supported by a research fellowship from the European Molecular Biology Laboratory interdisciplinary post-doctoral programme under Marie Curie Cofund Actions MSCA-COFUND-FP (grant agreement 847543). We thank the Sample Preparation and Characterisation facility of the European Molecular Biology Laboratory Hamburg and the Cryo-EM facility of the Centre for Structural Systems Biology for their support, technical assistance and advice. The groups of H. Tidow, C. Utrecht and H. Sondermann are acknowledged to grant access to their equipment necessary for this project. We thank M. Killer, S. Mostafavi and G. Chojnowski for their scientific input on cryo-EM data processing and J. Steinke for their additional assistance in liposome reconstitution experiments. This work was in part supported by the Deutsche Forschungsgemeinschaft to M.D. (DA 1785–1) and the Agence Nationale de la Recherche to B.G. (grants ANR-18-CE11-0009 and ANR-22-CE11-0008). This study was further supported by a Bundesministerium für Bildung und Forschung (BMBF) grant (number 05K18YEA) to C.L. Part of this work was performed at the Cryo-EM Facility at the Centre for Structural Systems Biology, supported by the UHH and Deutsche Forschungsgemeinschaft (grant numbers INST 152/772-1|152/774-1|152/775-1|152/776-1|152/777-1 FUGG).

## Author contributions

M.D., B.G., C.L., O.G., K.E.J.J., A.R.M. and M.AR. designed the study. M.D., O.G., K.E.J.J., M.I., C.D., K.S., D.M.L., J.D., A.R.M., V.K., F.G., A.E. I.M.-T., N.N.L. and S.H.C performed the experiments. M.D., B.G., C.L., O.G., K.E.J.J., W.D., V.K., F.G. and M.A.-R. analysed the data and/or its significance. M.D., B.G., C.L., O.G. and K.E.J.J. wrote the manuscript with contributions from W.D. and P.S.; and M.D., B.G. and C.L. acquired funding.

## Funding

## Competing interests

The authors declare no competing interests.

## Additional information

**Extended data** is available for this paper at

**Supplementary information** The online version
contains supplementary material available at

**Correspondence and requests for materials** should be addressed to
Bruno Gasnier, Christian Löw or Markus Damme.

**Peer review information** *Nature Cell Biology* thanks Viktor Korolchuk,
Joseph Mindell and the other, anonymous, reviewer(s) for their
contribution to the peer review of this work. Peer reviewer reports are
available.

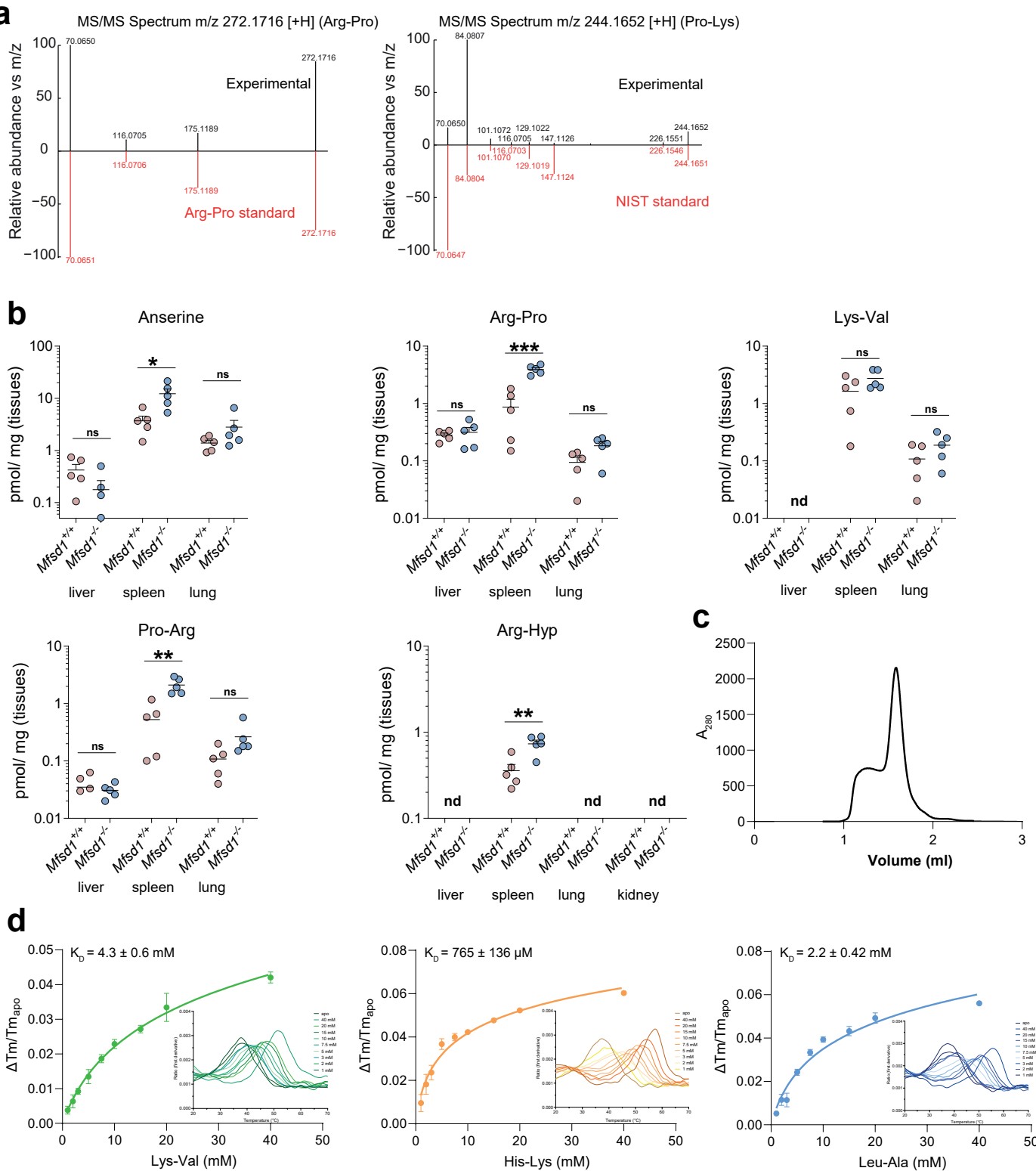

**Extended Data Fig. 1 | Validation of Arg-Pro and Pro-Lys with standards and dipeptide levels in tissues of *Mfsd1^tm1d/tm1d* mice.** (**a**) Mirror plots for the experimental MS/MS spectra of Arg-Pro (left) and Pro-Lys (right) and authentic chemical standards. The individual spectra for the experimentally determined metabolites are shown in black, and the spectra of the chemical standards are shown in red. (**b**) Quantification of the levels of the dipeptides anserine, Arg-Pro, Lys-Val, Pro-Arg, and Arg-Hyp in total tissue lysates of 6-month-old wildtype ad Mfsd1 knockout mice. n = 5 animals/genotype. P-values were calculated using two-tailed paired t-tests. Error bars show the mean ± SEM. (**c**) SEC chromatogram of purified MFSD1 with a Streptavidin-tag (Superdex 200 5/150 increase (Cytiva) column). (**d**) $K_D$ measurements for Lys-Val, His-Lys, Leu-Ala. $K_D$ measurements are based on changes in the thermal stability of MFSD1 in the presence of varying concentrations of the dipeptides Lys-Val (green), His-Lys (orange), and Leu-Ala (blue). N = 3 experiments, data are shown as mean ± SEM. $K_D$ values were determined using Moltenprot (Kotov et al.[67]). Source numerical data are available in source data.

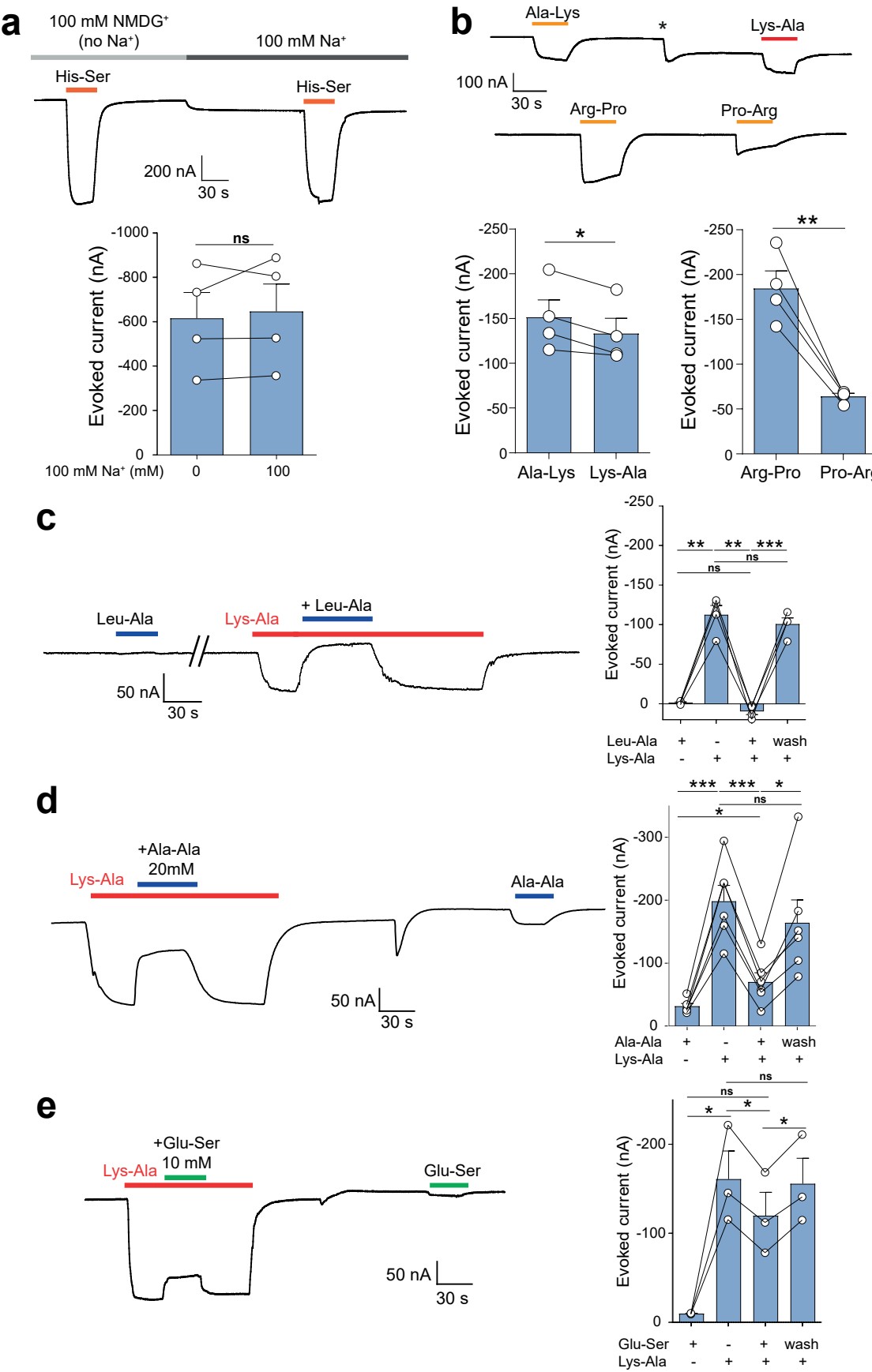

**Extended Data Fig. 2 | See next page for caption.**

**Extended Data Fig. 2 | Dipeptide selectivity of MFSD1/GLMP in the TEVC oocyte assay.** (**a**) The MFSD1/GLMP transport current does not depend on sodium ions. His-Ser (10 mM) was applied to MFSD1/GLMP oocytes at pH 5.0 in the presence of Na$^+$ or NMDG$^+$ as the major cation. *P*-values were calculated using two-tailed paired t-tests. mean + SEM, n = 4 oocytes. (**b**) Residue order effect for two cationic dipeptides. Representative traces and mean TEVC currents ± SEM of four MFSD1/GLMP oocytes. *P*-values were calculated using two-tailed paired

t-tests. mean ± SEM, n = 4 oocytes. (**c, d**) Competition of the Lys-Ala current by neutral dipeptides. Lys-Ala (3 mM) and Leu-Ala or Ala-Ala (20 mM) were applied separately or simultaneously to MFSD1/GLMP oocytes at pH 5.0. Representative traces and mean currents ± SEM n = 4 (Leu-Ala) n = 6 (Ala-Ala) oocytes. (**e**) The competition experiment was repeated with the anionic dipeptide Glu-Ser (10 mM). *P*-values were calculated using two-tailed paired t-tests, mean ± SEM, n = 3 oocytes. Source numerical data are available in source data.

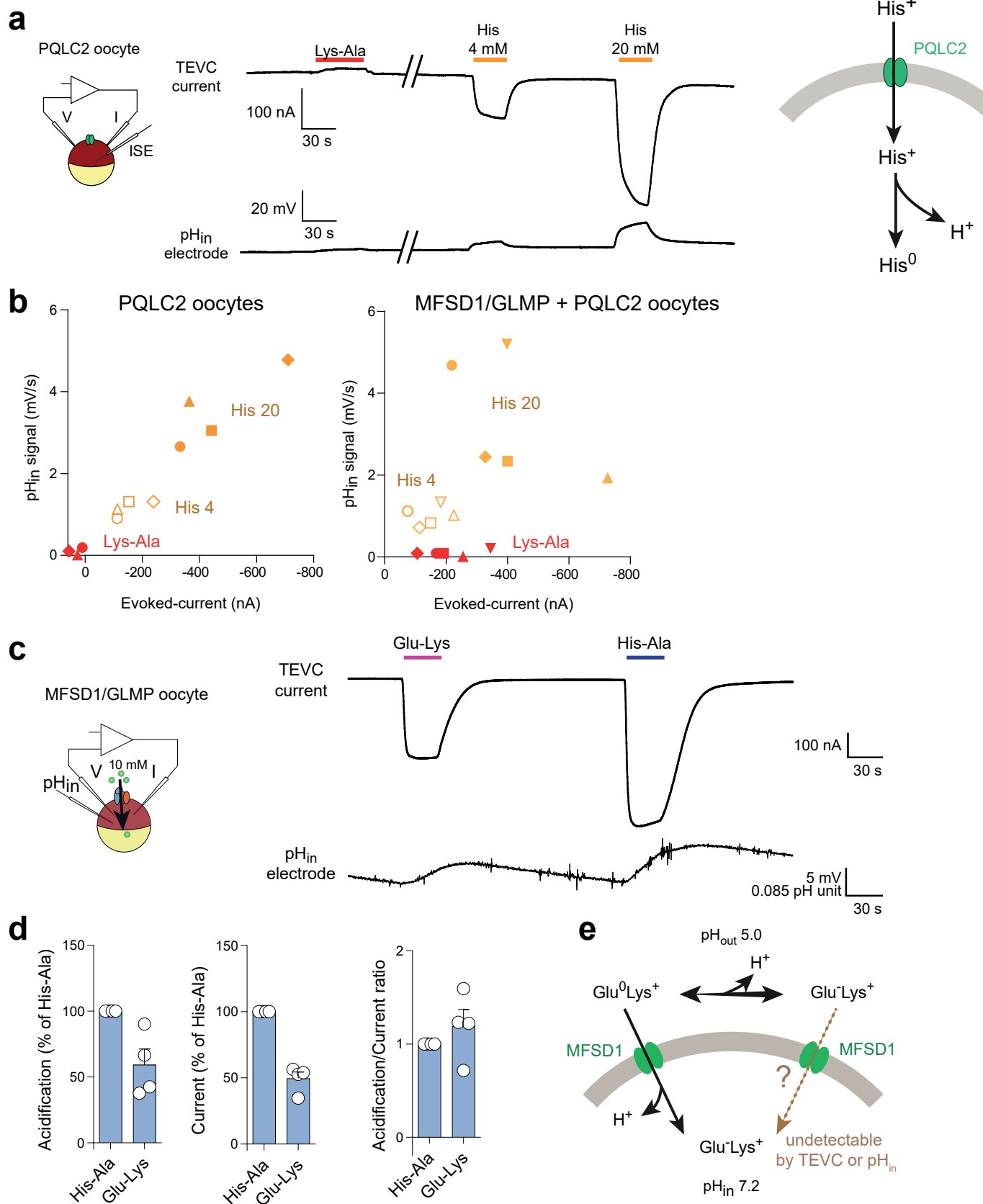

**Extended Data Fig. 3 | See next page for caption.**

**Extended Data Fig. 3 | Additional evidence for the uniporter model uptake of Glu-Lys into MFSD1/GLMP oocytes.** (**a**) Combined TEVC and intracellular pH (pH$_{in}$) recording of oocytes expressing only the sorting mutant of PQLC2. Lys-Ala (20 mM) is not transported by PQLC2. The traces are representative of four PQLC2 oocytes. (**b**) The current/acidification relationship of the experiments is shown in Fig. 4a and Extended Data Fig. 3a. The graphs show individual TEVC and pH$_{in}$ responses to Lys-Ala (10 mM) and His (4 or 20 mM). Each symbol shape represents a distinct oocyte. (**c**) Representative TEVC and pH$_{in}$ traces of the response of MFSD1/GLMP oocytes to Lys-Glu. (**d**) Acidification and current responses normalized to His-Ala and normalized acidification/current ratios. Data are means ± SEM of 4 oocytes. (**e**) A model accounting for the uptake of Glu-Lys by MFSD1/GLMP. Only uptake of the minor cationic form, Glu$^0$-Lys$^+$, can be detected in this electrophysiological technique. Source numerical data are available in source data.

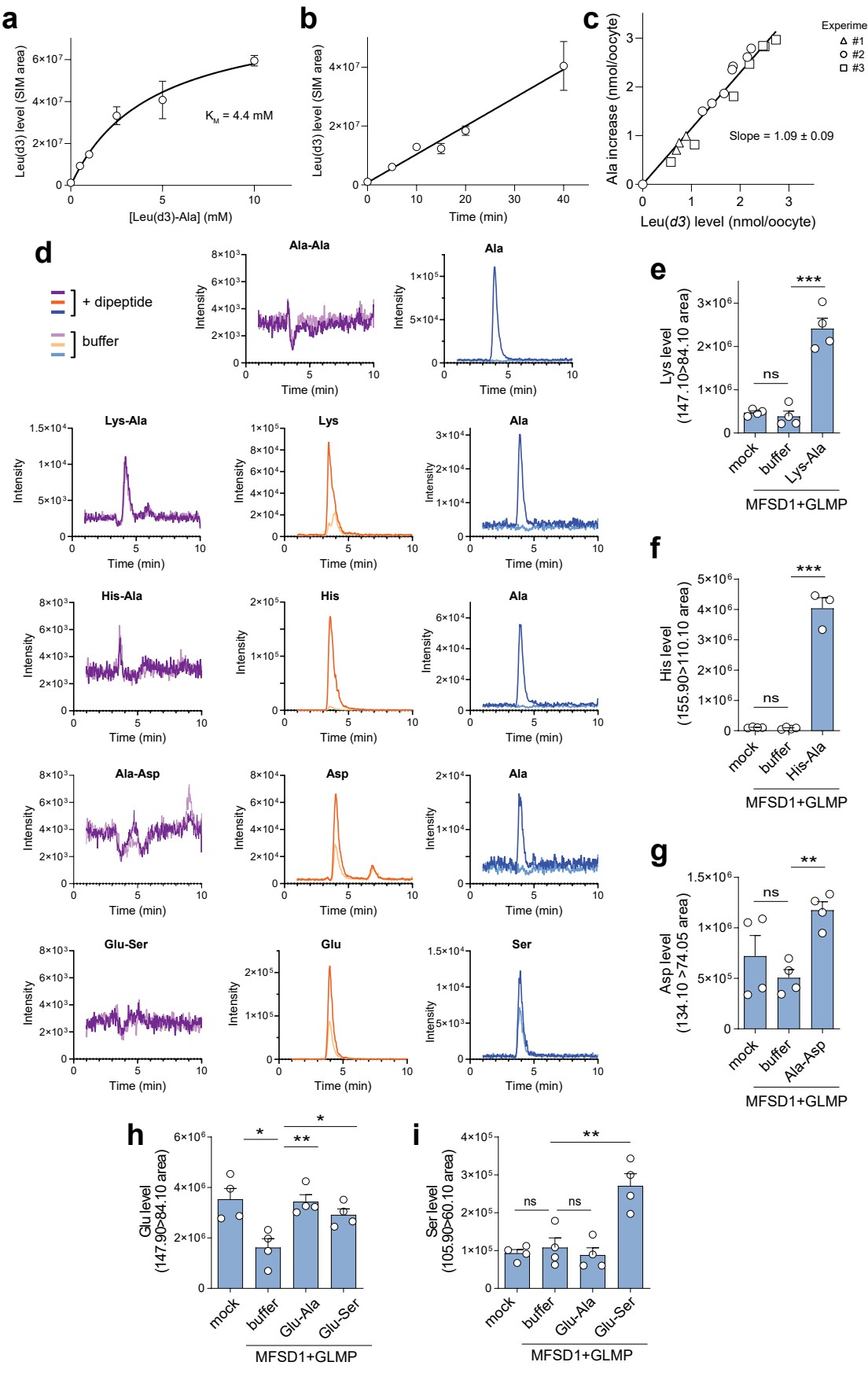

**Extended Data Fig. 4 | See next page for caption.**

**Extended Data Fig. 4 | Additional evidence for Leu-Ala uptake by MFSD1 and Substrate selectivity of MFSD1/GLMP in the LC-MS/MS assay.** (**a**) Dose-response relationship of the accumulation of Leu($d3$) in MFSD1/GLMP oocytes exposed to Leu($d3$)-Ala (means ± SEM of 3 oocytes). The line shows a hyperbolic curve fit with a $K_M$ value of 4.4 mM ($R^2$ = 0.989). (**b**) Time course of Leu($d3$) accumulation in the presence of 10 mM Leu($d3$)-Ala (means ± SEM of 3 oocytes). Linear regression $R^2$ = 0.980. (**c**) Relationship between the accumulation of Leu($d3$) and the increase of 'light' Ala over its endogenous level. Data shown in Fig. 5c were replotted to show the equimolar ratio between these two proxies of Leu($d3$)-Ala uptake. Linear regression of the pooled data yielded a ratio of 1.15 Ala molecule co-released with each Leu($d3$) molecule ($R^2$ = 0.980), or a mean ratio of 1.09 ± 0.09 when the 3 experiments were analyzed separately. (**d-i**) Substrate selectivity of MFSD1/GLMP in the LC-MS/MS assay. (**d**) Representative LC-MS chromatograms of 6 to 8 oocytes per condition from 2 independent experiments. (**e-i**) Relative quantification of the chromatographic peak area of Lys, His, Asp, Glu, and Ser in extracts from mock and MFSD1/GLMP oocytes, incubated or not, with the indicated dipeptides (10 mM) for 23 min at pH 5.0. Data are means ± SEM; (e + g): n = 4 oocytes, (f): n = 3 oocytes from the same experiment. Two-tailed unpaired t-tests: ns = not significant; * p ≤ 0.05; ** ≤ 0.01; *** p ≤ 0.001. Source numerical data are available in source data.

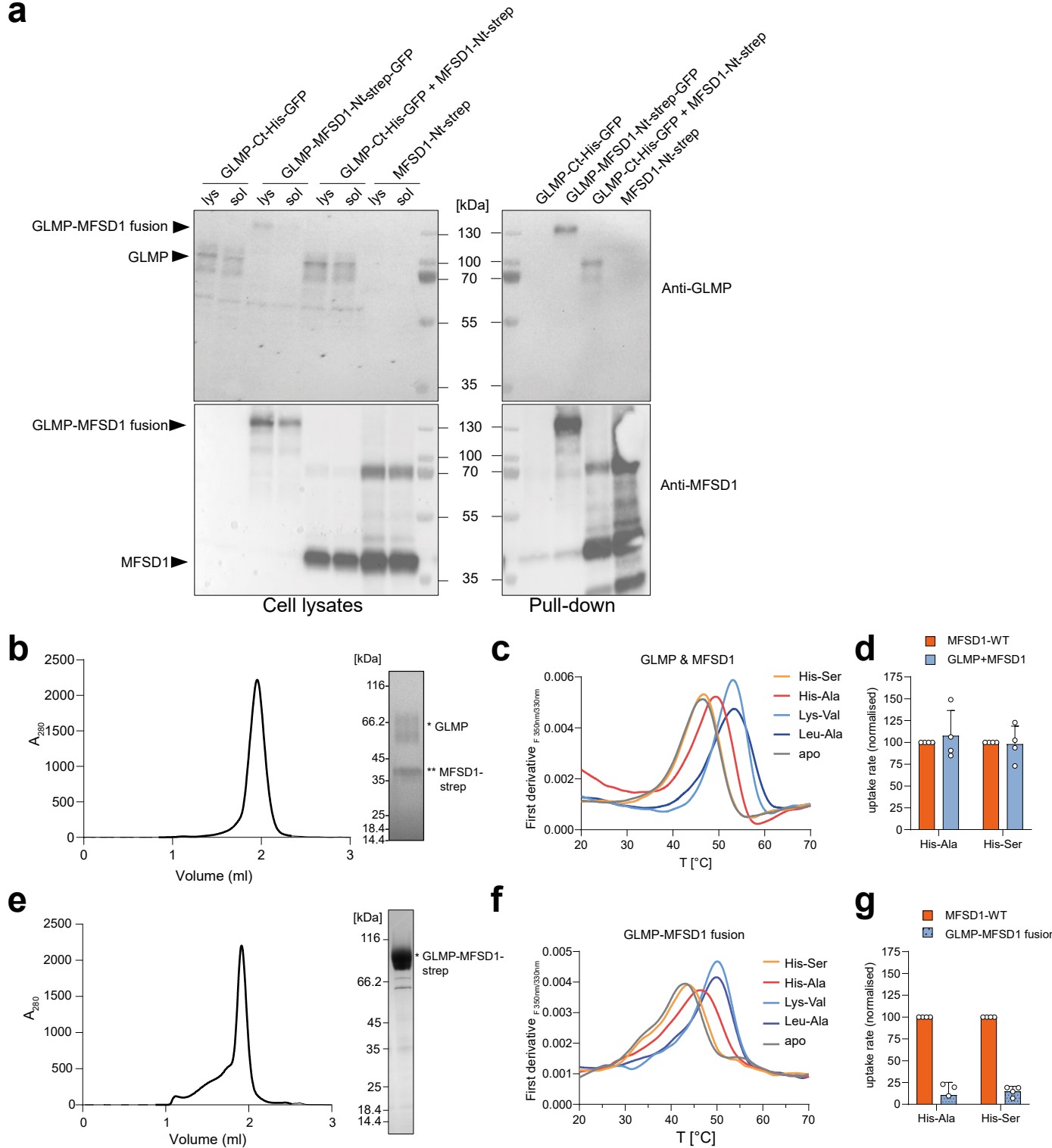

**Extended Data Fig. 5 | The recombinantly expressed proteins interact in vitro.** (**a**) Pull-down assays of Twin-Strepavidin (strep) tagged MFSD1 (MFSD1-Nt-strep), GFP-strep-tagged GLMP-MFSD1 (GLMP-MFSD1-Nt-strep-GFP) and GFP-8×His-tagged GLMP (GLMP-Ct-His-GFP) and GLMP-MFSD1-Nt-strep-GFP. Each protein was individually over-expressed in Expi293F, and additionally, MFSD1-Nt-strep was co-expressed with GLMP-Ct-His-GFP. MFSD1 and GLMP were detected in Western blot using specific primary antibodies against either MFSD1 or GLMP. Samples either contained crude lysate (lys) or the soluble fraction (sol) of each construct over-expressed in Expi293F cells (left panel) or the elution fraction after pull-down over Strep-Tactin beads (right panel). Bands corresponding to GLMP, GLMP-MFSD1, or MFSD1 are indicated. The experiment was performed once. (**b**) SEC chromatogram and SDS-PAGE gel of purified

GLMP in complex with MFSD1 carrying a twin-streptavidin-tag (MFSD1-strep). (**c**) Thermal stability of GLMP in complex with MFSD1-strep in the absence (apo) or presence of 5 mM His-Ser, His-Ala, Lys-Val, or Leu-Ala. (**d**) Normalized initial uptake rates of the dipeptides His-Ala or His-Ser during liposome-based assays by MFSD1$_{WT}$ and GLMP/MFSD1 complex. n = 4, of two reconstitution batches; Error bars are shown as SD. (**e**) SEC chromatogram and SDS-PAGE gel of purified GLMP-MFSD1-fusion protein carrying a twin-streptavidin-tag. (**f**) Thermal stability of GLMP-MFSD1-fusion protein in the absence (apo) or presence of 5 mM His-Ser, His-Ala, Lys-Val, or Leu-Ala. (**g**) Normalized initial uptake rates of the dipeptides His-Ala or His-Ser during liposome-based assays by MFSD1$_{WT}$ and GLMP-MFSD1 fusion protein. n = 4, Error bars are shown as SD. Source numerical data and unprocessed blots are available in source data.

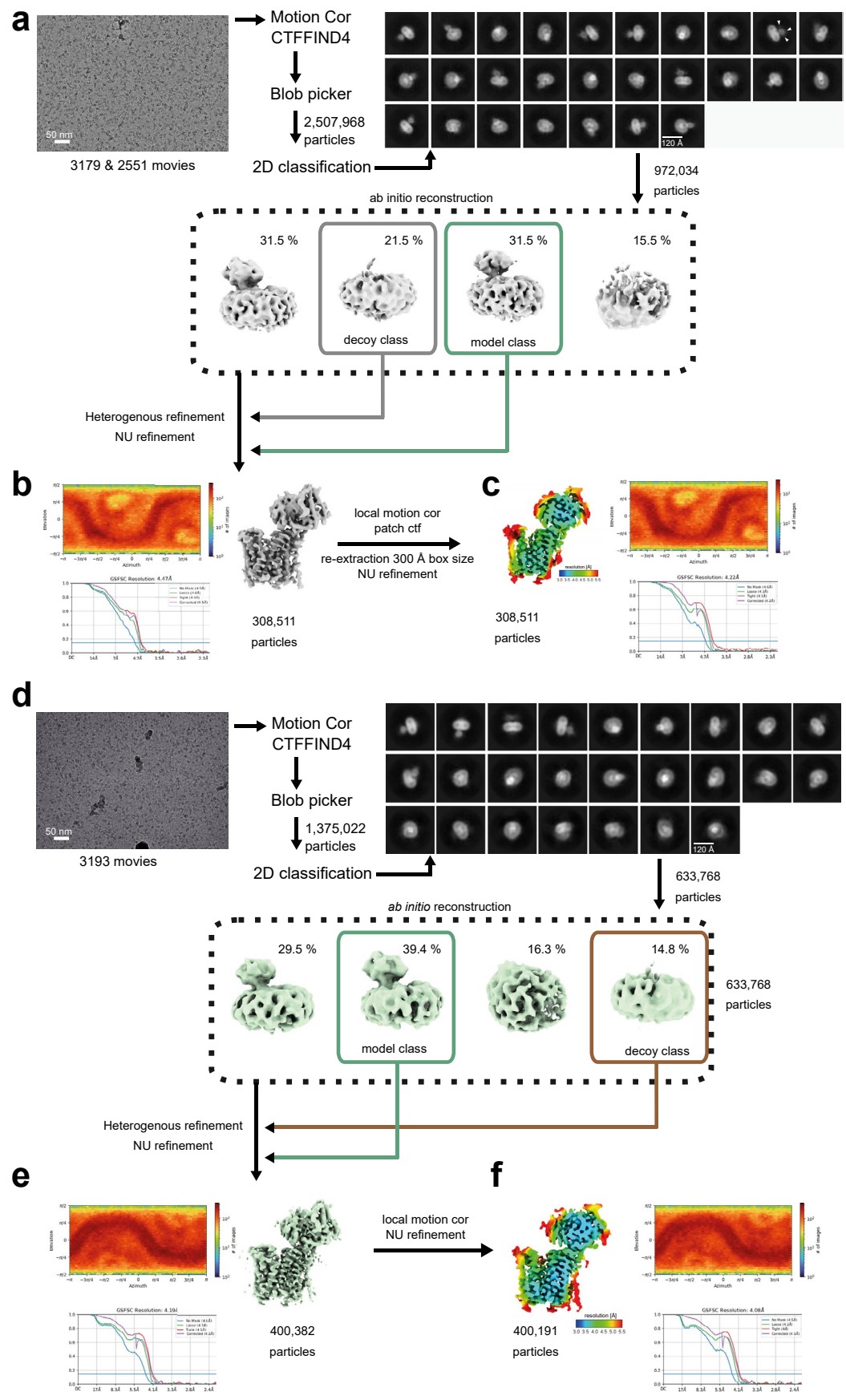

**Extended Data Fig. 6 | See next page for caption.**

**Extended Data Fig. 6 | Cryo-EM data collection and processing of the GLMP-MFSD1$_{apo}$ and GLMP-MFSD1$_{His-Ala}$ data sets.** (**a**) Image data processing workflow with a representative micrograph and 2D classes of the GLMP-MFSD1$_{apo}$ data set. All data were processed in cryoSPARC. (**b**) Angular distribution plot, GSFSC plot, and cryoEM map of initial reconstruction before further refinement. White arrowheads denote densities corresponding to N-glycans. (**c**) Angular distribution plot, GSFSC plot, and cryo-EM map of the final GLMP-MFSD1$_{apo}$ reconstruction colored by local resolution. (**d-f**) Cryo-EM data collection and processing of the GLMP-MFSD1$_{His-Ala}$ data set. (**d**) Image data processing workflow with a representative micrograph and 2D classes of the GLMP-MFSD1$_{His-Ala}$ data set. All data were processed in cryoSPARC. (**e**) Angular distribution plot, GSFSC plot, and Cryo-EM map of initial reconstruction before further refinement. White arrowheads in 2D class references denote densities corresponding to N-glycans. (**f**) Angular distribution plot, GSFSC plot, and Cryo-EM map of the final GLMP-MFSD1$_{His-Ala}$ reconstruction colored by the local resolution.

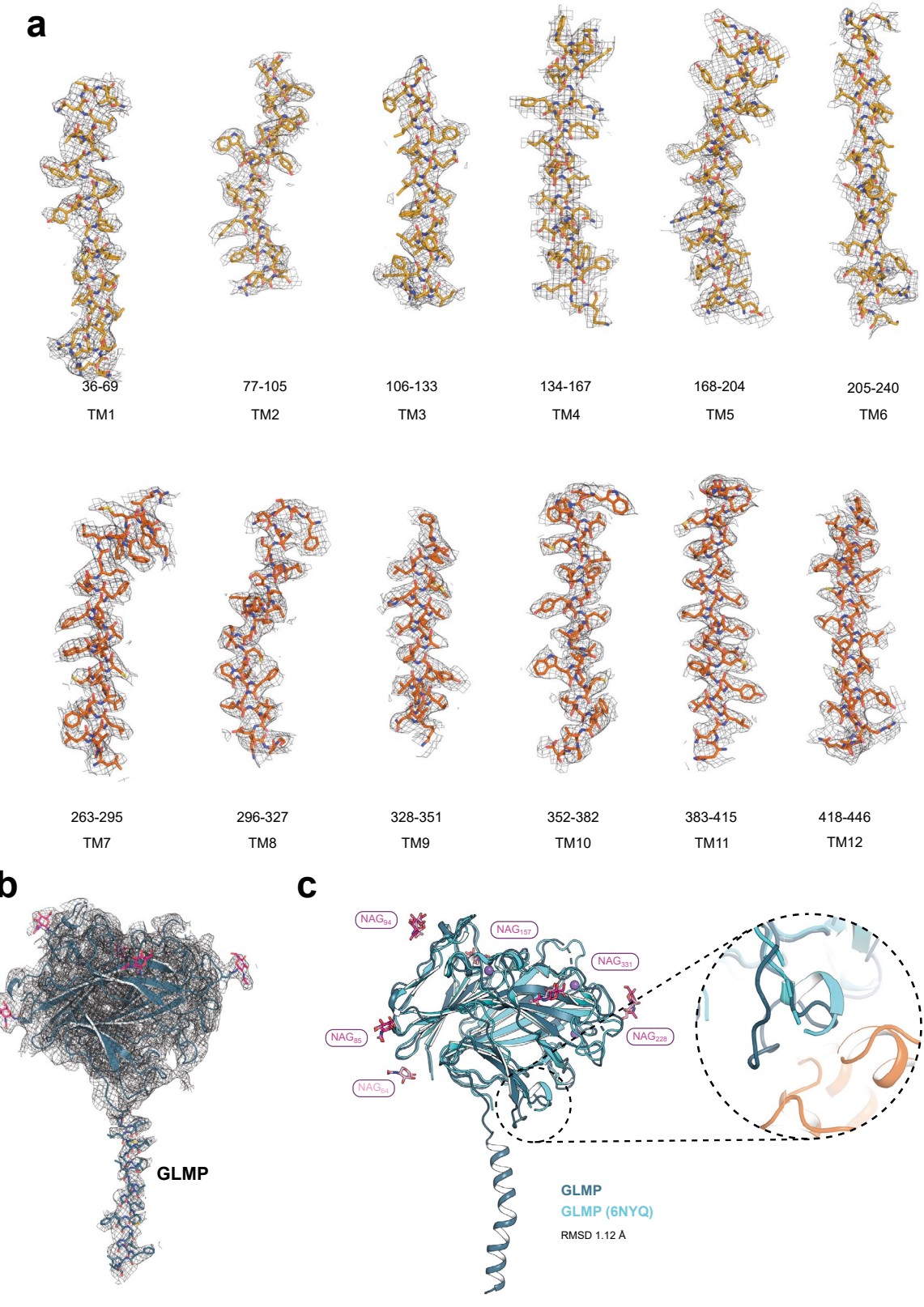

**a**

| 36-69 | 77-105 | 106-133 | 134-167 | 168-204 | 205-240 |
| TM1 | TM2 | TM3 | TM4 | TM5 | TM6 |

| 263-295 | 296-327 | 328-351 | 352-382 | 383-415 | 418-446 |
| TM7 | TM8 | TM9 | TM10 | TM11 | TM12 |

**b** GLMP

**c**

NAG94
NAG157
NAG331
NAG85
NAG228
NAG64

GLMP
GLMP (6NYQ)
RMSD 1.12 Å

**Extended Data Fig. 7 | See next page for caption.**

**Extended Data Fig. 7 | Density map of the Cryo-EM structure of GLMP-MFSD1$_{His-Ala}$. (a)** Cyro-EM map (grey mesh) is shown and depicts a density within a 2.5 Å radius of any modelled atom. Maps are shown for individual helices of MFSD1, with individual residues shown as sticks (yellow and orange). **(b)** Cryo-EM map (grey mesh) is shown and depicts a density within a 2.5 Å radius around the model of GLMP (blue). Individual residues of the transmembrane helix and the five NAG molecules (pink) are shown as sticks. **(c)** Overlay of the Cryo-EM structure of GLMP (blue) with the X-ray structure of GLMP (light blue, PDB-ID: 6NYQ). The RMSD$_{C\alpha}$ of the superimposition is 1.12 Å over 271 residues of the luminal GLMP domain. Five of the NAG molecules (pink) identified in the Cryo-EM structure overlap with the six NAG molecules (light pink) found in the X-ray structure of GLMP (PDB-ID: 6NYQ). Additionally, the X-ray structure of GLMP contains three sodium ions (purple spheres). The zoom-in highlights the loop region, which is responsible for the interaction of GLMP with MFSD1, which has not been modeled in the crystal structure and is structured in the EM-derived model.

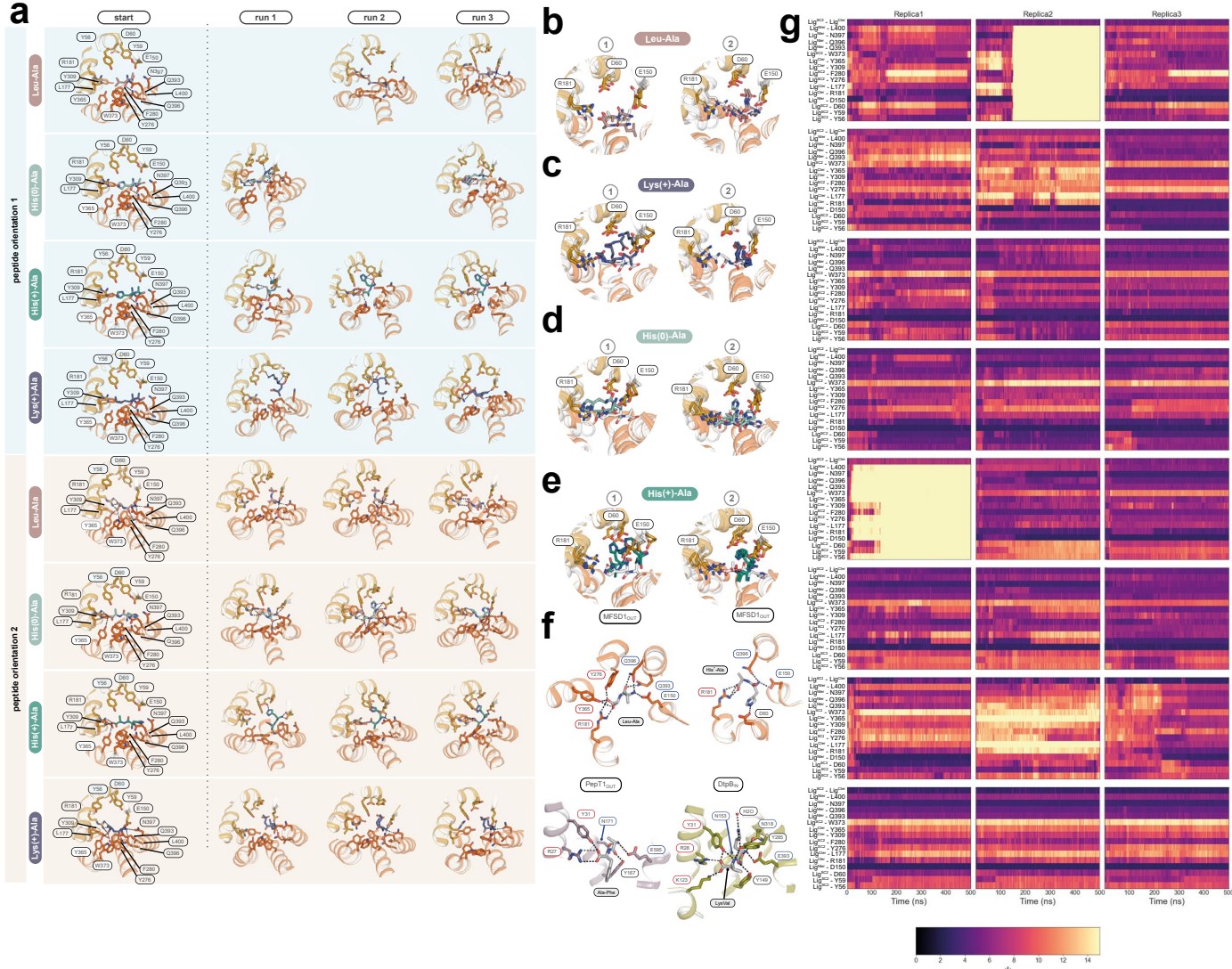

**Extended Data Fig. 8 | MD simulations of dipeptide-bound MFSD1 and Flexibility of dipeptide binding in MFSD1 during MD simulations.** (**a**) MD simulations were performed on MFSD1 in complex with the dipeptides Leu-Ala (rose), His-Ala in its neutral (His(0)-Ala, pale teal) and charged (His(+)-Ala, teal) state and Lys(+)-Ala, purple). The basis of the binding mode for each peptide was the initial non-protein density found in the GLMP-MFSD1$_{His-Ala}$ map. The dipeptide His-Ala was placed in two different binding poses, denoted peptide orientation 1 (pale blue background) and peptide orientation 2 (light orange background). Based on this pose, the remaining ligands were oriented. Shown and labeled are critical binding site residues for each starting structure and the same view for the binding site of each simulation run after 500 ns. Additional interacting residues appearing at the endpoint of the simulation are highlighted in the respective panels. (**b-g**) Flexibility of dipeptide binding in MFSD1 during MD simulations. (**b**) The binding site of MFSD1 represents the starting pose of Leu-Ala (grey) and the final pose of the peptide after 500 ns of MD simulation (light purple) for each of the two peptide orientations (1 and 2). Below are RMSD plots of distant changes of the N- and C-terminus of the peptide with respect to residues E150 and R181. Plots show the results for each peptide orientation (orientation 1-blue, orientation 2-orange). MD simulations were run in triplicates. (**c**) Illustration of the MFSD1 binding site of MFSD1 with the starting pose of Lys-Ala pose of the dipeptide His(0)-Ala (grey) and the final pose after 500 ns of triplicate MD simulation (pale teal) for each of the two peptide orientations (1&2). (**d**) Binding

site of MFSD1 showing the starting pose of the dipeptide His(0)-Ala (grey) and the final pose after 500 ns of MD simulation (pale teal) for each of the two peptide orientations (1&2). Below are RMSD plots of distant changes of the N- and C-terminus of the substrate with respect to residues E150 and R181. RMSD plots highlight the results for each peptide orientation (orientation 1-blue, orientation 2-orange) run in triplicates. (**e**) The Starting pose of the dipeptide His(+)-Ala (grey) and the final pose in the MFSD1 binding site after 500 ns of MD simulation (dark teal) are shown for each of the two peptide orientations (1&2). MD simulations were run in triplicates for each peptide orientation (orientation 1-blue, orientation 2-orange). (**f**) Comparison of dipeptide binding sites of MFSD1 in the outward-open conformation (orange) either in complex with Leu-Ala (MD simulation run 3, peptide orientation 2) or His(+)-Ala (MD simulation run 1, peptide orientation 2), the Cryo-EM structure of PepT1 (PDB-ID: 7PMX) in the outward-open conformation (pale purple) in complex with Ala-Phe, and the X-ray structure of DtpB (PDB-ID: 8B1H) in the inward-open conformation (pale green) bound to the dipeptide Lys-Val. Critical residues important for the coordinating of the N-terminus of the substrate are framed in blue and the C-terminus in red. Hydrogen bonds are shown as black dashed lines. (**g**) Plots of the distance changes between the N- (Lig$^{Nter}$), C-terminus (Lig$^{Cter}$), and the sidechain of the 2nd amino acid (Lig$^{SC2}$) in the ligand and sixteen residues located in the MFSD1 binding site.

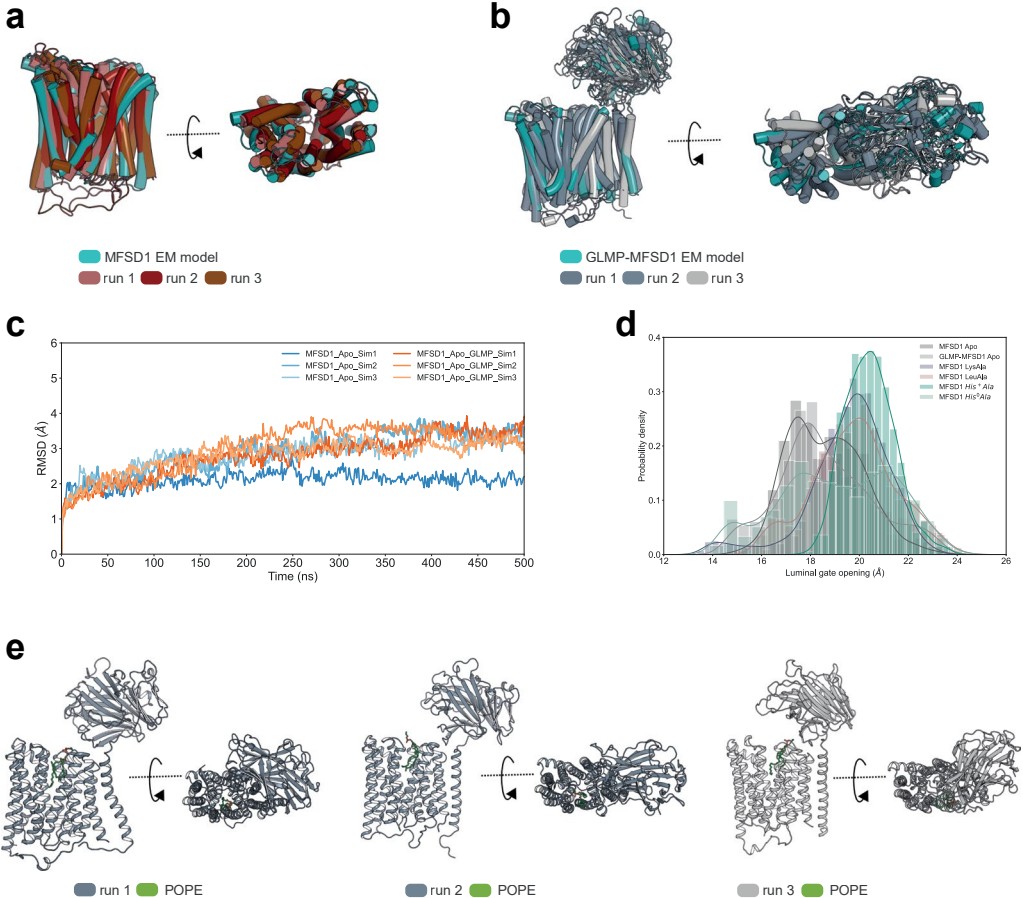

**Extended Data Fig. 9 | Analysis of MD simulations of GLMP-MFSD1$_{apo}$ and MFSD1$_{apo}$.** (**a**) Superposition of MFSD1 starting model (in blue, derived from the cryoEM model) and the structure after 500 ns of MD simulations of the three replicates (shades of red). (**b**) Superposition of GLMP-MFSD1 starting models (in blue, representing the cryoEM structure) and the complex after 500 ns of MD simulations of the three replicates (shades of grey). (**c**) RMSD (MFSD1 in relation to the starting CryoEM model) changes over the course of the MD simulation. The change in RMSD of MFSD1 in the apo form is shown in red, and the RMSD change of MFSD1 in the apo form as part of the complex with GLMP is given in grey blue.

Each model was run in triplicates (Sim 1-3). (**d**) Conformational dynamics of the gate open to the lysosomal lumen (luminal gate) of MFSD1 in the absence/presence of the substrates and GLMP+MFSD1$_{apo}$. The width of the opening of the luminal gate is defined as the distance between the centre of mass of two TM groups (group 1: TM1, TM2, and TM5; group 2: TM7, TM8, and TM11) and is plotted against its probability density. (**e**) A POPE lipid molecule (green) is only found between TMs of MFSD1 during simulations (run1-3) of the GLMP-MFSD1$_{apo}$ complex but not when simulations are run on MFSD1 only.

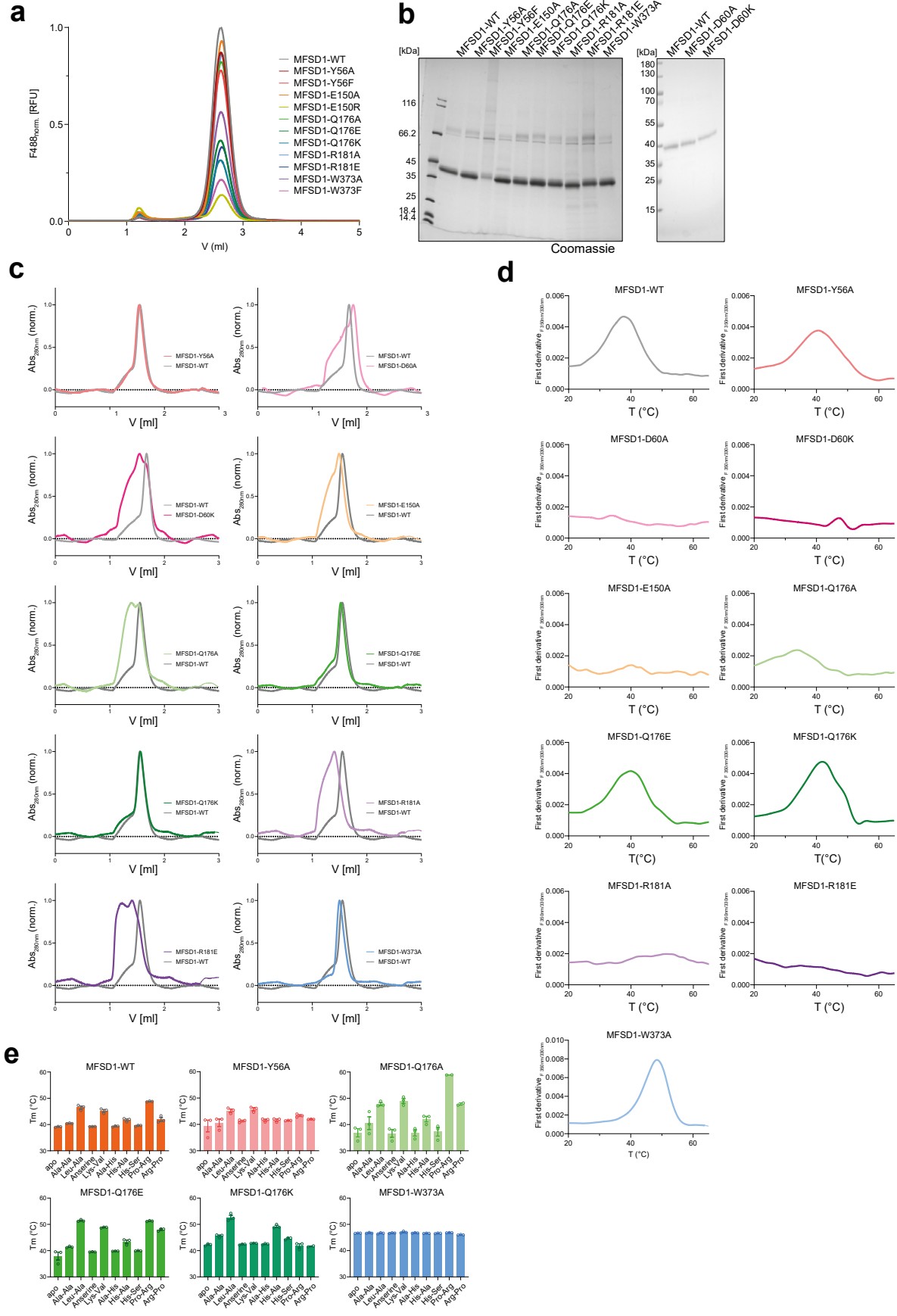

**Extended Data Fig. 10 | See next page for caption.**

**Extended Data Fig. 10 | The binding-site mutations in MFSD1 lose binding to peptides in the thermal stability assay.** (**a**) FSEC analysis of MFSD1$_{WT}$ and mutants normalized to the fluorescent signal at $\lambda_{ex}$ = 488 nm/$\lambda_{em}$ = 510 nm of GFP (F488) of MFSD1$_{WT}$. The supernatant of soluble fraction after whole-cell solubilization was loaded onto a Superose 6 5/150 column. (**b**) SDS-PAGE of purified MFSD1$_{WT}$ and binding site mutants. For each lane, 2 μg of protein were loaded. (**c**) Comparison of SEC traces of binding site mutants (colored) to MFSD1$_{WT}$ (grey) of each mutant. Each mutant was purified once for subsequent experiments. (**d**), (**e**) Melting temperatures derived from thermal stability experiments of each mutant in the absence (apo, grey) or presence of 5 mM of selected peptides. n = 3 independent experiments with data shown as mean ± SD. For mutants for which no bar graph is given, unfolding traces are given for the apo state to show that no T$_M$ value could be determined. Source numerical data and unprocessed blots are available in source data.

# Reporting Summary

## Statistics

For all statistical analyses, confirm that the following items are present in the figure legend, table legend, main text, or Methods section.

| n/a | Confirmed | |
|---|---|---|
| ☐ | ☒ | The exact sample size (*n*) for each experimental group/condition, given as a discrete number and unit of measurement |
| ☐ | ☒ | A statement on whether measurements were taken from distinct samples or whether the same sample was measured repeatedly |
| ☐ | ☒ | The statistical test(s) used AND whether they are one- or two-sided *Only common tests should be described solely by name; describe more complex techniques in the Methods section.* |
| ☒ | ☐ | A description of all covariates tested |
| ☒ | ☐ | A description of any assumptions or corrections, such as tests of normality and adjustment for multiple comparisons |
| ☒ | ☐ | A full description of the statistical parameters including central tendency (e.g. means) or other basic estimates (e.g. regression coefficient) AND variation (e.g. standard deviation) or associated estimates of uncertainty (e.g. confidence intervals) |
| ☐ | ☒ | For null hypothesis testing, the test statistic (e.g. *F*, *t*, *r*) with confidence intervals, effect sizes, degrees of freedom and *P* value noted *Give P values as exact values whenever suitable.* |
| ☒ | ☐ | For Bayesian analysis, information on the choice of priors and Markov chain Monte Carlo settings |
| ☒ | ☐ | For hierarchical and complex designs, identification of the appropriate level for tests and full reporting of outcomes |
| ☐ | ☒ | Estimates of effect sizes (e.g. Cohen's *d*, Pearson's *r*), indicating how they were calculated |

*Our web collection on statistics for biologists contains articles on many of the points above.*

## Software and code

Policy information about availability of computer code

| Data collection | The Cryo-EM data was collected using EPU2.8.0.1256REL (Thermo Fisher Scientific);<br>The Thermal Stability data was collected using PR.ThermControl v.2.1.2 (Nanotemper);<br>The fluorescent data for liposome-based assay was collected using i-control™ (Tecan);<br>Multi-target LC-MS/MS analysis of oocyte extracts was performed using the LabSolutions v 5.118 software (Shimadzu Corporation)<br>Electrophysiological recordings were made with Clampex V. 11.2.2.17 (Molecular Devices);<br>The confocal immunofluorescence microscopy data was collected using a LSM980 + AiryScan (Zeiss) with Zen Blue software V. 3.2<br>Molecular Dynamics simulations were performed using GROMACS 2021.3. |
|---|---|
| Data analysis | The Cryo-EM data was processed using CryoSPARCv3. Refinement and validation: Isolde v1.6, Phenix v.1.20.1; Coot v.0.9.8.1; ChimeraX 1.3; PyMOL v2.5.5; MolProbity 4.2<br>For sequence alignments ClustalOmega (no version) and visualization ESPript 3.<br>Structure modeling was performed using AlphaFold2 and AlphaFold2 Multimer;<br>Thermal Stability data was analyzed using PR.ThermControl v.2.1.2 (Nanotemper), MoltenProt v0.2.1; Visualized using Graphpad Prism v.9.5.1<br>Targeted LC-MS/MS chromatograms were analysed using LabSolutions v 5.118 (Shimazu )<br>Electrophysiological data were analyzed with Clampfit v. 11.2.2.17 (Molecular Devices)<br>ImageJ 1.52 was used for adjustment of images<br>Graphpad Prism v.9.5.1 was used for general statistics |

For manuscripts utilizing custom algorithms or software that are central to the research but not yet described in published literature, software must be made available to editors and reviewers. We strongly encourage code deposition in a community repository (e.g. GitHub). See the Nature Portfolio guidelines for submitting code & software for further information.

## Data

Policy information about availability of data

  All manuscripts must include a data availability statement. This statement should provide the following information, where applicable:
    - Accession codes, unique identifiers, or web links for publicly available datasets
    - A description of any restrictions on data availability
    - For clinical datasets or third party data, please ensure that the statement adheres to our policy

The EM data and fitted models for GLMP-MFSD1 have been deposited in the Electron Microscopy Data Bank under accession code EMD-19005 and the PDB under accession code 8R8Q. Raw data used for data plotting are available as a supplementary table. The crystal structure of GLMP used for comparative analysis in this study can be found in the PDB under accession code 6NYQ AlphaFold2 predictions of MFSD1 as well as the models of MFSD1 and GLMP-MFSD1 after 500 ns of molecular dynamics simulations and metabolomics raw data were deposited to Zenodo (Alphafold models: 10.5281/zenodo.10276738; MFSD1 apo/with ligands in initial poses and after 500 ns MD: 10.5281/zenodo.10276760;). All protein sequences used in this study are publicly available at Uniprot (https://www.uniprot.org/). The metabolomics data are available at 10.5281/zenodo.10839783. Source data have been provided in Source Data. All other data supporting the findings of this study are available from the corresponding author on reasonable request.

## Research involving human participants, their data, or biological material

Policy information about studies with human participants or human data. See also policy information about sex, gender (identity/presentation), and sexual orientation and race, ethnicity and racism.

| | |
|---|---|
| Reporting on sex and gender | N/A |
| Reporting on race, ethnicity, or other socially relevant groupings | N/A |
| Population characteristics | N/A |
| Recruitment | N/A |
| Ethics oversight | N/A |

Note that full information on the approval of the study protocol must also be provided in the manuscript.

# Field-specific reporting

Please select the one below that is the best fit for your research. If you are not sure, read the appropriate sections before making your selection.

☒ Life sciences          ☐ Behavioural & social sciences          ☐ Ecological, evolutionary & environmental sciences

For a reference copy of the document with all sections, see nature.com/documents/nr-reporting-summary-flat.pdf

# Life sciences study design

All studies must disclose on these points even when the disclosure is negative.

| | |
|---|---|
| Sample size | No sample-size calculation was performed. The sample size for each experiment is included in the respective figure legend and statistical methods were used to calculate standard deviation as noted in figure legends. Experiments were performed with at least n = 3; n-numbers are provided in the figures/figure legends. Sample size was determined based on similar studies in this field and our own experience obtaining consistent data throughout replicating the experiments. |
| Data exclusions | Data were only excluded if obvious technical problems occurred during the experiments. Generally, no data was excluded. |
| Replication | Experiments were performed with at least n = 3. For each representative image/dataset, at least 2 other independent experiments were successfully repeated. |
| Randomization | Samples were not randomized for this study because all experiments were internally controlled. |
| Blinding | Blinding was not performed as is not applicable to the study for many experiments (purification of the recombinant proteins, cryo-EM, nano-DSF experiments with recombinant proteins, oocyte overexpressing MFSD1/GLMP). |

# Reporting for specific materials, systems and methods

We require information from authors about some types of materials, experimental systems and methods used in many studies. Here, indicate whether each material, system or method listed is relevant to your study. If you are not sure if a list item applies to your research, read the appropriate section before selecting a response.

## Materials & experimental systems

| n/a | Involved in the study |
|---|---|
| ☐ | ☒ Antibodies |
| ☐ | ☒ Eukaryotic cell lines |
| ☒ | ☐ Palaeontology and archaeology |
| ☐ | ☒ Animals and other organisms |
| ☒ | ☐ Clinical data |
| ☒ | ☐ Dual use research of concern |
| ☒ | ☐ Plants |

## Methods

| n/a | Involved in the study |
|---|---|
| ☒ | ☐ ChIP-seq |
| ☒ | ☐ Flow cytometry |
| ☒ | ☐ MRI-based neuroimaging |

# Antibodies

| Antibodies used | LAMP1 clone 1D4B  (purified rat monoclonal, Developmental Studies Hybridoma Bank); LAMP1 clone 1D4B (purified rat monoclonal, conjugated to AlexaFluor 647, #121609, BioLegend); HA clone 3F10 (rat monoclonal, ROAHAHA/11867423001; Sigma-Aldrich / Merck), ); HA clone 3F10 (rat monoclonal,  conjugated to FITC, 11988506001; Sigma-Aldrich / Merck), GFP ( 11814460001, mouse monoclonal, Roche Molecular Biochemicals), mKate2 (rabbit polyclonal, TA150072, Origene), KDEL (clone 10C3, mouse monoclonal, ADI-SPA-827-D, Enzo Life Sciences), Cox IV (rabbit polyclonal, ab16056, Abcam), Golgin 97 (clone CDF4, mouse monoclonal, A-21270, Thermo Scientific Fisher). The antibody against cathepsin D (CTSD) was custom-made against a synthetic peptide (CKSDQSKARGIKVEKQIFGEATKQP) and immunization of rabbits, followed by affinity purification against the immunization peptide. The custom-made MFSD1- and GLMP-specific rabbit polyclonal antibodies were described before (Massa Lopez et al., 2019 / PMID: 31661432).<br>Secondary antibodies: HRP-coupled goat anti rat (112-035-143, Dianova); goat anti mouse (115-035-146, Dianova), goat anti rabbit (111-035-144, Dianova) |
|---|---|
| Validation | Well established commercial antibodies were used throughout the study. Whenever possible, monoclonal antibodies were used. MFSD1, GLMP and CTSD specific antibodies were in house knockout validated (MFSD1/GLMP:  PMID: 31661432; CTSD: unpublished). LAMP1 1D4B mAb is knockout validated (PMID: 10212251). Tag-specific monoclonal antibodies (3F10 / HA; GFP; mKate2) were validated upon the overexpression of tagged proteins with the corresponding tag; Golgin 97 (clone CDF4) is well-established with >100 citations. Cox IV is well-established with >200 citations. KDEL (clone 10C3) is well-established with >40 citations. Additional antibody validation can be found on the manufacturer's website. |

# Eukaryotic cell lines

Policy information about cell lines and Sex and Gender in Research

| Cell line source(s) | Expi293F Thermo Fisher (Cat. number: A14527), Mouse embryonic fibroblasts (MEF); Winnie Eskild lab |
|---|---|
| Authentication | None of the cell lines used were authenticated. |
| Mycoplasma contamination | We confirmed that Expi293F cells were  tested negative for mycoplasma contamination. |
| Commonly misidentified lines (See ICLAC register) | No commonly misidentified cell line was used |

# Animals and other research organisms

Policy information about studies involving animals; ARRIVE guidelines recommended for reporting animal research, and Sex and Gender in Research

| Laboratory animals | Mice: C57Bl/6N-Mfsd1tm1dHhtg/Damme; age 6 months; Mice were housed under standard laboratory conditions with a 12-hour light/dark cycle and constant room temperature and humidity. Food and water were available ad libitum.<br>Xenopus laevis oocytes. Female Xenopus frogs  (age: 6- 10 years) were used for the production of oocytes. |
|---|---|
| Wild animals | No wild animals were used. |
| Reporting on sex | Female and male mice were used for the study and the gender was not considered in the study design. |
| Field-collected samples | The study did not involve samples collected from the field. |
| Ethics oversight | Mice: ethical agreement Ministerium für Energiewende, Klimaschutz, Umwelt und Natur V242-13648/2018<br>Xenopus: ethical agreement Ministère de l'Enseignement Supérieur et de la Recherche, France APAFiS #14316-2017112311304463 v4 |

Note that full information on the approval of the study protocol must also be provided in the manuscript.

## Plants

| | |
|---|---|
| Seed stocks | N/A |
| Novel plant genotypes | N/A |
| Authentication | N/A |

