## [Peer Review File · Nature Cell Biology]

Peer Review Information

Journal: Nature Cell Biology

Manuscript Title: MFSD1 with its accessory subunit GLMP functions as a general dipeptide uniporter in lysosomes

Corresponding author name(s): Professor Markus Damme

Editorial Notes:

**Redactions –
published data**

Reviewer Comments & Decisions:

Decision Letter, initial version:

Please delete the link to your author homepage if you wish to forward this email to co-authors.

Dear Professor Damme,

Thank you again for submitting your manuscript, "MFSD1 in complex with its accessory subunit GLMP functions as a general dipeptide uniporter in lysosomes", to Nature Cell Biology and for your patience with the peer review process. Your manuscript has now been seen by 5 referees, who are experts in lysosomes, cell biology (Referee #1); structural biology (Referee #2); metabolomics (Referee #3); molecular dynamics simulations (Referee #4); and electrophysiology, channels (Referee #5). As you will see from their comments (attached below), they found the work of potential interest but have raised substantial concerns, which in our view would need to be addressed with considerable revisions before we can consider publication in Nature Cell Biology.

Nature Cell Biology editors discuss the referee reports in detail within the editorial team, including the chief editor, to identify key referee points that should be addressed with priority, as opposed to requests that are overruled as being beyond the scope of the current study. To guide the scope of the revisions, I have listed these points below. Our standard revision process is six months, and we are committed to providing a fair and constructive peer-review process, so please feel free to contact me if you would like to discuss any of the referee comments further.

In our view, it would be essential to address the following points:

1- The reviewers asked for a deeper understanding of the types of dipeptides transported and bound:
Rev#1 point #1

Rev#2 points #3, #4, #8, #9, #10

Note also that Rev#5 disagrees with the interpretation that MFSD1 is a 'simple' dipeptide uniporter and believes that it may function as a uniporter and as a coupled transporter for some but not all peptides. The referee indicates that, in the absence of a rigorous test of this hypothesis, the conclusions need to be adjusted throughout at a minimum.

2- Please address Rev#3's points #4-5-6 about metabolite quantifications and the standards and quality control processes involved. Their other questions about the methods and analyses done should be addressed rigorously.

3- Rev#4's concerns about the types of lipids used in vitro and in molecular dynamics analyses should be addressed in full: points #1, #2, #3 including with new modelling analyses. Please also follow their recommendation to perform more than 300ns of MD simulations (#5), a recommendation shared by Rev#5 in their last point.

4- All other concerns about the strength of the current data, requests for clarifications, text edits, methodological details, controls and statistical analyses should be tackled in full.

5- Finally, please pay close attention to our guidelines on statistical and methodological reporting (listed below) as failure to do so may delay the reconsideration of the revised manuscript. In particular, please provide:

We would be happy to consider a revised manuscript that would satisfactorily address these points, unless a similar paper is published elsewhere or is accepted for publication in Nature Cell Biology in the meantime.

- ensure that it conforms to our format instructions and publication policies (see below and <https://www.nature.com/nature/for-authors>).

- provide a point-by-point rebuttal to the full referee reports verbatim, as provided at the end of this letter.

- provide the completed Reporting Summary (found here <https://www.nature.com/documents/nr-reporting-summary.pdf>). This is essential for reconsideration of the manuscript will be available to

editors and referees in the event of peer review. For more information see <http://www.nature.com/authors/policies/availability.html> or contact me.

When submitting the revised version of your manuscript, please pay close attention to our [Digital Image Integrity Guidelines](https://www.nature.com/nature-portfolio/editorial-policies/image-integrity). and to the following points below:

Nature Cell Biology is committed to improving transparency in authorship. As part of our efforts in this direction, we are now requesting that all authors identified as 'corresponding author' on published papers create and link their Open Researcher and Contributor Identifier (ORCID) with their account on the Manuscript Tracking System (MTS), prior to acceptance. ORCID helps the scientific community achieve unambiguous attribution of all scholarly contributions. You can create and link your ORCID from the home page of the MTS by clicking on 'Modify my Springer Nature account'. For more information please visit www.springernature.com/orcid.

This journal strongly supports public availability of data. Please place the data used in your paper into a public data repository, or alternatively, present the data as Supplementary Information. If data can only be shared on request, please explain why in your Data Availability Statement, and also in the correspondence with your editor. Please note that for some data types, deposition in a public repository is mandatory - more information on our data deposition policies and available repositories appears below.

[Redacted]

We hope that you will find our referees' comments and editorial guidance helpful. Please do not hesitate to contact me if there is anything you would like to discuss. Thank you again for considering the journal for your work,

Best wishes,

Melina

Melina Casadio, PhD
Senior Editor, Nature Cell Biology
ORCID ID: <https://orcid.org/0000-0003-2389-2243>

Reviewers' Comments:

Reviewer #1:

Remarks to the Author:

This outstanding manuscript identifies the function of an orphan lysosomal transporter MFSD1 in the export of cationic and neutral dipeptides. As highlighted by the authors, the current knowledge of the transport mechanisms through the lysosomal membrane is still limited. The paper goes long way to not only ascribe a function in the efflux of dipeptides to the MFSD1/GLMP complex, but also presents a very thorough investigation of the underlying mechanism, culminating in the determination of the complex structure by cryo-EM. Whilst the expertise of this reviewer lies in the area of cell and particularly lysosomal biology with lesser knowledge of the transport assays, structural biology and MD, it is nevertheless very clear that the overall study is robust. Indeed, it is a rarity to see such a compelling dataset where discrepancies arising from different experimental approaches are systematically addressed by an additional set of orthogonal assays to produce a significant insight into the function of a previously uncharacterised protein. The importance of the findings is also apparent, with the model proposed in Fig. 7E, and which is strongly supported by the data, is essentially describing a novel mechanism by which lysosomal function is maintained during periods of high activity. Clearly, extensive further investigations will be required to establish the role of MFSD1/GLMP-mediated transport in cellular processes coordinated by lysosomes in physiological and stress conditions. However, this study will undoubtedly serve as a strong foundation for future functional studies. For this reviewer, only minor changes are required before publication as outlined below.

1. It is less clear from the dataset if MFSD1/GLMP has a function in transporting anionic dipeptides which, based on data in Suppl Fig. 2D, E, show competition effect. Are these transported by this MFSD1, and if not would it be possible to explain this and discuss a potential alternative mechanism?
2. It would be helpful to indicate the positions of zoomed areas in Fig. 7D using white boxes.
3. Some figures are incorrectly referenced in the text: Suppl. Fig. 6A-C is 7A-C, Suppl. Fig. 15 is 16.
4. Remove ".) In fact," on page 21.

Reviewer #2:

Remarks to the Author:

Jungnickel et al. present findings on the deorphanization, structure and function of the lysosomal heterodimeric transporter complex MFSD1-GLMP. The authors isolated lysosomes for LC-MS/MS-based metabolomics analysis and functionally identified MFSD1-GLMP as a peptide transporter. The identification/deorphanization was achieved by recombinantly expressing the complex in frog oocytes and human cells. From the latter, protein was purified using detergent for biochemical, biophysical and structural analysis, and functional reconstitution into liposomes. Structures of both the apo- and His-Ala bound forms of MFSD1-GLMP were determined by cryo-electron microscopy (cryo-EM) at resolutions of 4.2 Å and 4.1 Å, respectively.

The submitted work represents a significant contribution with new and important findings for the scientific community - congratulations to the authors. However, there is room for improvement: The manuscript is written in a very concise and compact manner, and it is very dense in information and results, likely due to limitations imposed by the target journal. More detailed descriptions would enhance comprehension. Additionally, the figures often contain multiple panels, which can impede readability. Overall, the manuscript is well-written, but certain sections, especially in discussion and some captions, can benefit from adaptation/rephrasing. Please refer to the outlined issues below that will help improving presentation of the data and readability.

(1) Please mention the detergent (DDM/CHS) used for purification, nanoDSF and structure determination in main part. This will make reading easier and fluent not having to stop to check Materials and Methods for such an important information.

(2) Figure 1 is suboptimal: Too many panels and unclear, e.g., too small. A possibility would be to split the Figure into two Figures, e.g., Figure 1 (panels A-E) and Figure 2 (panels F-H) - would also make thematically sense. See also specific feedback regarding panels G and H, below:

- Fig. 1G: Please make short horizontal colored lines thicker, colors are not clearly distinguishable.

- Fig. 1H: Please split this panel into two rows or put into Supplementary Information.

Currently, names of amino acids and dipeptides are difficult to read, too small, significant zooming-in necessary.

(3) Page 22, top, sentence: The strongest stabilization effects were observed for neutral dipeptides (e.g., Leu-Leu, $\Delta T_m = 14$ °C) and dipeptides that possess at least one positively-charged residue (e.g., Pro-Arg, $\Delta T_m = 12.1$ °C and His-Lys, $\Delta T_m = 12$ °C).

Based on this finding, dipeptides with different physicochemical properties bind to MFSD1. However, only positively charged dipeptides are taken up (Figure 2).

What is the mechanistic sense behind binding efficiently neutral ($\Delta T_m = 14$ °C) and negatively charged dipeptides (Suppl. Figure 2D, E), but not transporting them?

(4) Page 25: Here the determined K_m -values of His-Ser (119 μM) and His-Ala (24 μM) are presented. Affinities are relatively high (i.e., low K_m -values), compared to other shown kinetic parameters.

How are these two K_m -values correlated to the observed ΔT_m from Figure 1H, e.g., His-Ser almost doesn't show thermostability, but has a significant and for peptide transporters relative high affinity ?

(5) Fig. 3C: Please make short horizontal colored lines thicker, colors are not clearly distinguishable.

(6) Suppl. Fig. 1, panel C: Please indicate in Figure legend SEC column type used.

Furthermore, there is an important void resp. aggregates between 1-1.5 ml. Do the authors have an explanation for this? It is not a typical void peak, but remains almost constant until elution volume 1.5 ml.

(7) In the context of wording: i) please avoid expressions such as "powerful hydrolysis" and "powerfully driven". Use more neutral adjectives and adverbs. ii) If "The first major" is mentioned, then the "second" and even "third" is expected by the reader. Please adapt. iii) Fig. 6H: Rather a lengthy caption, the information might be better placed in the main text.

(8) In general, there are numerous dipeptides tested, but it is not clear to the reader why certain

dipeptides have been chosen for assays, structure and some not. Please provide some rationale, where possible, so that the reader understands the logic behind the procedure.

(9) As mentioned in point (3), Leu-Leu has a high ΔT_m , but also Leu-Ala ($\Delta T_m \sim 12-13^\circ\text{C}$).

- Why did the author not determine the K_D of these interesting inhibitory compounds?
- What was the rationale to choose His-Ala (Fig. 1H, $\Delta T_m \sim 6^\circ\text{C}$) for structure solution and not, e.g., Leu-Leu and/or Leu-Ala that showed higher ΔT_m , being most probably more protein stabilizing and more optimal for structure solution (e.g., at higher resolution) ?

(10) Page 39: "...MFSD1 has a low affinity for its substrates in the mM range"

This statement is misleading, since K_m values are indicated with ~ 24 to ~ 120 μM (Fig. 3E), i.e., values comparable to PHT1 and PHT2. Please revisit this statement.

(11) Comments regarding Figures:

- Suppl. Fig. 6A and 6B: GLMP migrates as a diffuse band, because glycosylation. It would be beneficial to mention this information for less experienced readers.
- Suppl. Fig. 6C,F and Suppl. Fig. 13D: y-axis description is not clear "first derivative (Ratio)" or "Ration (first derivative)": of what? Furthermore, use one consistent y-axis naming/description in both Suppl. Figs.
- Suppl. Fig. 9, comment: "Cryo-EM map (grey mesh) within a 2.5 \AA radius around". Please clarify what is meant by " 2.5 \AA radius around".
- Fig. 6D: For better understanding, please color/highlight the "additional binding site density".
- Fig. 6E and Suppl. Fig. 10: The interaction D60 to His(+)-Ala is an important interaction. It would be beneficial to also display this residue in Fig. 6E and Suppl. Fig. 10.
- Fig. 6G: In the plot, certain mutants cannot be distinguished from each other, because the bar heights are very low (color of bar not visible). For better view, mutants might be labeled (also) below the bars.
- Fig. 7A, B and C: Image quality/resolution low, please be sure to provide better resolved panels in this Figure.
- Fig. 7B and C, legend: Not a "salt bridge" between residue Y416 (MFSD1) and R292 (GLMP), rather H-bond - please check. Please also revisit the legend to Fig. 7: There is space for improvement.

(12) Typos:

- Page 21: "In fact, The..." - lower case T.
- Page 33: "... Arg181, respectively (Suppl. Figure 1)" - probably Suppl. Figure 10 (not 1) ?

Reviewer #3:

Remarks to the Author:

I am reviewing primarily as a metabolomics technical expert.

1. LC-MS/MS-based analysis of dipeptides from tissues: sample prep. no protease inhibitors were used in sample prep - might the dipeptides be generated during and/or after homogenization and before ethanol addition (which denatures and deactivates proteins)?

2. LC-MS/MS-based analysis of dipeptides from tissues: AccQ acetonitrile solution - what is this? Is this commercial accq-tag kit? Please describe in more detail. It appears there are also numerous other

products in the methods which are ambiguously defined. (i.e. 96-black chimney deep well plate).

3. LC-MS/MS-based analysis of dipeptides from tissues: LC-MS/MS analyses - "detection of monovalent and divalent ions". can you clarify what this means? Generally we describe molecular ions as singly or doubly charged, not by valency, which generally refers to atoms. But i am not fully clear on what you mean.

4. Untargeted metabolomics and targeted metabolite quantitation: '1.5-3 μ L' injections - why a range? What samples used different injection volumes and why? Did you account for differences in injection volume before plotting and performing statistical analysis (Fig 1 E)? You mention full scan mode with polarity switching - what was the duty cycle under these conditions? That is, how many positive mode (or negative mode) scans per second did you observe?

5. Untargeted metabolomics and targeted metabolite quantitation: "known metabolite standards (MSMLS, Sigma-Aldrich) for targeted metabolite quantification" were any isotopically labelled internal standards used? It isn't clear that they were. Please provide more detail as to how quantification was performed. I would have to guess that you are using a calibration curve in neat solution and quantifying signal from complex matrix (sample extract). This is generally not a reliable quantitation method, as the sample may generate appreciable matrix effects (signal suppression, generally, but signal enhancement, at times). isotopically labelled internal standards are commonly used to account for this and enable more accurate quantification. Additionally, was a dose response curve used? If so, how many points? Please describe the quality of the curve. There is far too little information provided to know whether we can trust the quantitative values at all.

6. There is no mention of any sort of metabolomics quality control. pooled quality control samples, internal standard, solvent or sample preparation blanks, replicate injections, standard reference materials - these are all methods to demonstrate data quality, and the authors appear to have not reported use of any of them. I do realize that the metabolomics served as the discovery set, and the results from it were validated through extensive additional experimentation, so the lack of apparent quality control may not be too detrimental. However the authors should describe any quality control that they did perform. Also, recognize that strong quality control approaches in the future will ensure you pursue fewer artifactual results. I highly recommend depositing metabolomics data in an established repository such as MetaboLights, Metabolomics Workbench, MetaboBank, etc.

Reviewer #4:

Remarks to the Author:

Jungnickel and coauthors proposed an interdisciplinary approach combining metabolomics, in vitro electrophysiology, Cryo-EM structure determination, modeling, and Molecular Dynamics (MD) simulations to characterize the function of MFSD1 protein, in interaction with GLMP, as a dipeptide transporter. This is a very well designed work which allows understanding from the function of this transporter to its structural characterization at the molecular scale. This reviewer expertise mainly lies in molecular dynamics simulations and structural biology even if they have coordinated interdisciplinary projects. Thus, they can appreciate the extent of this work.

Overall, this work is very interesting and well conducted. The manuscript is clear. The MD simulations were properly set up and ran. 300ns is not a very extensive time to explore protein-ligand interaction

but the replicas are useful to assess the stability of these interactions. This reviewer has few points that need to be clarified before considering this manuscript acceptable for publication.

Major points:

1- It is mentioned in the introduction that lysosomal membrane contains a large fraction of glycolipids yet neither in vitro assays (reconstituted liposomes) nor molecular dynamics simulations include glycolipids. This glycolipids can affect protein function, by shielding some surface residues, or even limit the diffusion of peptides, especially the charged ones. So, at least, it would be important to discuss why these lipids were not included.

2- Related to question 1, can the authors explicit a bit more the choice of lipid compositions in liposomes and in MD simulations.

- For the former, why the choice of POPE:POPG:CHS at a 3:3:1 ratio ? Furthermore can the authors precise what is CHS ? Cholesteryl hemisuccinate ? If yes, why not have used directly cholesterol ?
- For the latter, the composition of POPE:POPC:CHOL:PSM at a ratio of 2:3:3:2 does not correspond neither to liposome lipid composition nor lysosome membrane composition (as glycolipids are missing).

3- Same idea as glycolipids, why the glycosylation was not fully modelled especially when studying the stability of MFSD1-GLMP interactions using MD simulations (figure S16). Indeed, as presented in figure 6-C, there is one N-glycosylation at the GLMP surface that can potentially point towards MSFD1 binding site. If this reviewer is right, the CryoEM approach only allowed determining a small part of this long (up to 14 sugars ?) oligosaccharide. Again,

4- In figure S11, only few distances were monitored along the course of the simulation. It would be more insightful to monitor distances between different parts of the peptide with all the residues presented in the "start" column in figure S10. This can be done by using matrix colored in function of distance. This will really help understanding which residues are involved in stable interactions.

5- In figure S14, it would have been useful to perform at least 300ns of MD simulations to check the stability of the AlphaFold model and assess the opening/closing of the canal using tools such as Hole (which is included in MDAnalysis:

https://docs.mdanalysis.org/1.1.1/documentation_pages/analysis/hole2.html).

6- It is necessary to release more data from the modelling. Only the AlphaFold model and the last structure is not enough. This reviewer was not able to find the Zenodo pages associated to these models.

Minor points:

Table S3, there are only 3 replicates for each system. As mentioned in figure S10, there are 2 positions for each peptide. So, for the peptide systems is it more 2 (each pose) x 3 x 300 ns ?

Reviewer #5:

Remarks to the Author:

This paper reports on a program to determine the function of the lysosomal MFS transporter MFSD1. I have been asked to review the electrophysiological portions of the experiments, and will focus most efforts on those parts of the paper.

Briefly, metabolomic screening of lysosomes from MFSD1 KO mice demonstrated an increase in the dipeptides Arg-Pro and Pro-Lys, suggesting an involvement in dipeptide transport, a conclusion supported by thermal stabilization of the purified MFSD1 protein by a series dipeptides. Targeting the protein (and its partner GLMP) to the plasma membrane in *Xenopus* oocytes allowed measurements of transport currents in the presence of some dipeptides, which are dose-dependent and activated at low extracellular (luminal) pH. Further experiments with reconstituted, purified protein in liposomes revealed electrogenic acidification induced by certain titratable dipeptides. Further oocyte experiments detect simultaneous acidification and transport currents in the presence of titratable peptides. CryoEM structures at limited resolution reveal the expected MFS fold, along with possible substrate density, and MD experiments suggest a possible orientation for the substrate. Final experiments (somewhat superficially) explore the interaction between MFSD1 and GLMP.

Overall, the conclusion that MFSD1 is a dipeptide transporter is very strongly supported. The initial metabolomic mass spec experiments point strongly in this direction and then are nicely reinforced by the experiments in *Xenopus* oocytes and with purified protein reconstituted into lipid vesicles. However, I disagree with the interpretation of the experiments concluding that the protein is a simple dipeptide uniporter. Indeed, much of the data presented here suggests otherwise—especially the results in Figure 3 and 4 demonstrating a decrease in pH accompanying transport of Histidine- and Glutamate-containing dipeptides. These results suggest to me that transport of these titratable peptides is indeed proton-coupled, though I agree with the authors that such coupling is unlikely for the lysine-containing peptides where no pH change is seen. If true, this would a very interesting finding—that transport is coupled to the proton gradient for some peptides but not for others. To truly distinguish between a uniporter and a sometimes-coupled transporter would require assessing the ability of a pH gradient to DRIVE the uptake of a peptide, like His-Ser. A true uniporter could never concentrate its substrate, it could only mediate transport down an electrochemical gradient. For example the liposome system could be used with equal K⁺ on both sides (no potential), equal concentrations of His-Ser on both sides, and a large inwardly directed pH gradient, ideally measuring substrate concentration alongside the pH. This clearly would represent substantial effort and is important, but not necessarily for the current paper, which already represents an impressive and substantial body of work. It would acceptable to acknowledge the ambiguity, soften the conclusion, and recognize that the distinction between uniport and substrate-dependent active transport requires further work.

A second major issue is the use of ΔpH in Figure 4 to estimate stoichiometry. In contrast to the currents, which are 1:1 with charges moved, the ΔpH is highly dependent on the cytoplasmic buffer capacity. Because of buffering, there is no reason to expect that the ΔpH is linearly related to the actual number of protons transported, much less to expect a proportionality constant of 1. This is particularly true if the ΔpH is near 1 unit, as suggested by the ~ 40 mV changes in voltage. The authors are vague about the calibration in the methods “The potential difference between the two inputs tested in diverse buffers...” but this fails to account for the buffer capacity of the oocyte cytoplasm. The stoichiometry arguments presented here are not central to the main conclusions and should be removed.

Other comments:

Fig 2 2C The data are convincing that most current comes with coexpression. However, the presentation of the MFSD1-only data is confusing. I suspect that the two entries for MFSD1 alone show a representative trace for a positive oocyte, and representative trace for a negative oocyte, with the numbers corresponding to the number of oocytes where this is seen, however, this is far from clear from reading the figure and legend. In addition, examining the data points on the bar graph makes it look like maybe only two points have nonzero current. These results need better explanation/presentation.

Figure 2E: What is the odd deflection early in the pH 7.0 trace? Is that reproducible?

Figure 2/3. The K_m s shown in Figure 2 (for Lys-Ala) and those in Figure 3E/F for His-Ser and His-Ala are 20-fold apart. Is there a substrate for which the two assays can be compared? Also, in Figure 3, the v_{max} values, in units of $\Delta F_{norm}/sec$ don't mean anything since these are arbitrary units—these should be removed. It would be interesting to know the Hill coefficients of these fits if the data are well-determined enough to reliably estimate them. The individual measurements need to be shown on these plots in addition to the means/std dev.

Figure 4 I'm not sure why the pH traces are shown here in mV when conversion to ΔpH is straightforward

Figure 5. The presentation of the putative substrate density is limited and unconvincing. It's pretty difficult to see that density directly in the EM maps, and Figure 5D is a bit confusing. For a start, perhaps it would be easier to visualize if the orientation of the structure in Fig 5D was the same as that in 5C (right panel). Could difference density be presented to highlight the area of interest? Why are the maps in the accompanying mrc files not aligned to each other to make comparisons easier for the reader?

Molecular Dynamics. These simulations are for pretty short times (300ns) and though in the presented data one orientation does seem more stable, it's hard to be confident that even the "stable" state would be that way. A minimum of maybe 500 us would be more convincing, as would a quantitative analysis. For Lys-Ala, though the conf1 distance is larger, it doesn't appear any less stable than conf2, and in two of the traces the substrate seems to move closer to E150 and becomes quite stable there; similarly the Cterminal of the peptide in that simulation seems equally unstable in both configurations.

READABILITY OF MANUSCRIPTS – Nature Cell Biology is read by cell biologists from diverse backgrounds, many of whom are not native English speakers. Authors should aim to communicate their findings clearly, explaining technical jargon that might be unfamiliar to non-specialists, and

avoiding non-standard abbreviations. Titles and abstracts should concisely communicate the main findings of the study, and the background, rationale, results and conclusions should be clearly explained in the manuscript in a manner accessible to a broad cell biology audience. Nature Cell Biology uses British spelling.

REFERENCES – are limited to a total of 70 for Articles, Resources, Technical Reports; and 40 for Letters. This includes references in the main text and Methods combined. References must be numbered sequentially as they appear in the main text, tables and figure legends and Methods and must follow the precise style of Nature Cell Biology references. References only cited in the Methods should be numbered consecutively following the last reference cited in the main text. References only associated with Supplementary Information (e.g. in supplementary legends) do not count toward the total reference limit and do not need to be cited in numerical continuity with references in the main text. Only published papers can be cited, and each publication cited should be included in the

numbered reference list, which should include the manuscript titles. Footnotes are not permitted.

Methods should be written concisely, but should contain all elements necessary to allow interpretation and replication of the results. As a guideline, Methods sections typically do not exceed 3,000 words. The Methods should be divided into subsections listing reagents and techniques. When citing previous methods, accurate references should be provided and any alterations should be noted. Information must be provided about: antibody dilutions, company names, catalogue numbers and clone numbers for monoclonal antibodies; sequences of RNAi and cDNA probes/primers or company names and catalogue numbers if reagents are commercial; cell line names, sources and information on cell line identity and authentication. Animal studies and experiments involving human subjects must be reported in detail, identifying the committees approving the protocols. For studies involving human subjects/samples, a statement must be included confirming that informed consent was obtained. Statistical analyses and information on the reproducibility of experimental results should be provided in a section titled "Statistics and Reproducibility".

All Nature Cell Biology manuscripts submitted on or after March 21 2016 must include a Data availability statement as a separate section after Methods but before references, under the heading "Data Availability". For Springer Nature policies on data availability see <http://www.nature.com/authors/policies/availability.html>; for more information on this particular policy see <http://www.nature.com/authors/policies/data/data-availability-statements-data-citations.pdf>. The Data availability statement should include:

- Accession codes for primary datasets (generated during the study under consideration and designated as "primary accessions") and secondary datasets (published datasets reanalysed during the study under consideration, designated as "referenced accessions"). For primary accessions data should be made public to coincide with publication of the manuscript. A list of data types for which submission to community-endorsed public repositories is mandated (including sequence, structure, microarray, deep sequencing data) can be found here <http://www.nature.com/authors/policies/availability.html#data>.
- Unique identifiers (accession codes, DOIs or other unique persistent identifier) and hyperlinks for datasets deposited in an approved repository, but for which data deposition is not mandated (see here for details <http://www.nature.com/sdata/data-policies/repositories>).
- At a minimum, please include a statement confirming that all relevant data are available from the authors, and/or are included with the manuscript (e.g. as source data or supplementary information), listing which data are included (e.g. by figure panels and data types) and mentioning any restrictions on availability.
- If a dataset has a Digital Object Identifier (DOI) as its unique identifier, we strongly encourage including this in the Reference list and citing the dataset in the Methods.

We recommend that you upload the step-by-step protocols used in this manuscript to the Protocol Exchange. More details can be found at www.nature.com/protocolexchange/about.

All imaging data should be accompanied by scale bars, which should be defined in the legend. Cropped images of gels/blots are acceptable, but need to be accompanied by size markers, and to retain visible background signal within the linear range (i.e. should not be saturated). The boundaries of panels with low background have to be demarked with black lines. Splicing of panels should only be considered if unavoidable, and must be clearly marked on the figure, and noted in the legend with a statement on whether the samples were obtained and processed simultaneously. Quantitative comparisons between samples on different gels/blots are discouraged; if this is unavoidable, it should only be performed for samples derived from the same experiment with gels/blots were processed in parallel, which needs to be stated in the legend.

The total number of Supplementary Figures (not including the "unprocessed scans" Supplementary Figure) should not exceed the number of main display items (figures and/or tables (see our Guide to

Authors and March 2012 editorial <http://www.nature.com/ncb/authors/submit/index.html#suppinfo>; <http://www.nature.com/ncb/journal/v14/n3/index.html#ed>). No restrictions apply to Supplementary Tables or Videos, but we advise authors to be selective in including supplemental data.

GUIDELINES FOR EXPERIMENTAL AND STATISTICAL REPORTING

REPORTING REQUIREMENTS – We are trying to improve the quality of methods and statistics reporting in our papers. To that end, we are now asking authors to complete a reporting summary that collects information on experimental design and reagents. The Reporting Summary can be found here <https://www.nature.com/documents/nr-reporting-summary.pdf> If you would like to reference the guidance text as you complete the template, please access these flattened versions at <http://www.nature.com/authors/policies/availability.html>.

Author Rebuttal to Initial comments

Prof. Dr. Markus Damme, Olshausenstraße 40, 24098 Kiel

Editorial Board
Nature Cell Biology
Melina Casadio, Senior Editor

Bearbeiter/in, Zeichen
PVO

Mail, Telefon, Fax
mdamme@biochem.uni-kiel.de
tel +49(0)431-880-2218

Biochemisches Institut

Geschäftsführender Direktor:
Prof. Dr. Becker-Pauly

Paketanschrift:
Eduard-Buchner-Haus
Otto-Hahn-Platz 9, 24118 Kiel

Postanschrift:
Olshausenstraße 40, 24098 Kiel

<https://www.uni-kiel.de/Biochemie/scripte/dynamic/groups/damme/damme.php>

Datum
25.03.2024

Resubmission of our revised research article to *Nature Cell Biology* (NCB-A53205A)

POINT-BY-POINT RESPONSE

Editors' / Reviewers' Comments:

In our view, it would be essential to address the following points:

1- The reviewers asked for a deeper understanding of the types of dipeptides transported and bound:

Rev#1 point #1

Rev#2 points #3, #4, #8, #9, #10

Note also that Rev#5 disagrees with the interpretation that MFSD1 is a 'simple' dipeptide uniporter and believes that it may function as a uniporter and as a coupled transporter for some but not all peptides. The referee indicates that, in the absence of a rigorous test of this hypothesis, the conclusions need to be adjusted throughout at a minimum.

2- Please address Rev#3's points #4-5-6 about metabolite quantifications and the standards and quality control processes involved. Their other questions about the methods and analyses done should be addressed rigorously.

3- Rev#4's concerns about the types of lipids used in vitro and in molecular dynamics analyses should be addressed in full: points #1, #2, #3 including with new modelling analyses. Please also follow their recommendation to perform more than 300ns of MD simulations (#5), a recommendation shared by Rev#5 in their last point.

Response: We addressed all of the points mentioned above as carefully as possible and included new data on the transport of anionic dipeptides (a point raised by reviewers #1 and #2), added controls, and explained experimental details for the metabolomics experiments, and added new molecular dynamics modeling analyses with extended simulation time (500 ns).

4- All other concerns about the strength of the current data, requests for clarifications, text edits, methodological details, controls and statistical analyses should be tackled in full.

Response: We made changes throughout the manuscript for points raised by the reviewers to increase readability (e.g., Figure 1), edited the text, and added details regarding controls (especially for metabolomics) and statistics.

5- Finally, please pay close attention to our guidelines on statistical and methodological reporting (listed below), as failure to do so may delay the reconsideration of the revised manuscript. In particular, please provide:

Response: A Supplementary Figure with unprocessed images of gels/blots is now included (Supplementary Figure 18).

Response: A Supplementary Table with numerical source data, with data for different figures provided as different sheets within a single Excel file.

We would be happy to consider a revised manuscript that would satisfactorily address these points, unless a similar paper is published elsewhere or is accepted for publication in Nature Cell Biology in the meantime.

Reviewer #1:

Remarks to the Author:

This outstanding manuscript identifies the function of an orphan lysosomal transporter MFSD1 in the export of cationic and neutral dipeptides. As highlighted by the authors, the current knowledge of the transport mechanisms through the lysosomal membrane is still limited. The paper goes long way to not only ascribe a

function in the efflux of dipeptides to the MFSD1/GLMP complex, but also presents a very thorough investigation of the underlying mechanism, culminating in the determination of the complex structure by cryo-EM. Whilst the expertise of this reviewer lies in the area of cell and particularly lysosomal biology with lesser knowledge of the transport assays, structural biology and MD, it is nevertheless very clear that the overall study is robust. Indeed, it is a rarity to see such a compelling dataset where discrepancies arising from different experimental approaches are systematically addressed by an additional set of orthogonal assays to produce a significant insight into the function of a previously uncharacterised protein. The importance of the findings is also apparent, with the model proposed in Fig. 7E, and which is strongly supported by the data, is essentially describing a novel mechanism by which lysosomal function is maintained during periods of high activity. Clearly, extensive further investigations will be required to establish the role of MFSD1/GLMP-mediated transport in cellular processes coordinated by lysosomes in physiological and stress conditions. However, this study will undoubtedly serve as a strong foundation for future functional studies. For this reviewer, only minor changes are required before publication as outlined below.

1. It is less clear from the dataset if MFSD1/GLMP has a function in transporting anionic dipeptides which, based on data in Suppl Fig. 2D, E, show competition effect. Are these transported by this MFSD1, and if not would it be possible to explain this and discuss a potential alternative mechanism?

Response: To address this point, we took advantage of the robust increase in 'light' Ala following the uptake of Leu(d3)-Ala to compare this substrate with neutral (Ala-Ala), cationic (Lys-Ala; His-Ala) and anionic (Glu-Ala; Ala-Asp) dipeptides in the LC/MS-based assay. These experiments showed that all dipeptides are transported by MFSD1 (albeit to a lesser extent for Ala-Asp). Targeted detection of Glu and Asp also showed a significant increase over their (high) endogenous background, confirming that MFSD1 also transports anionic dipeptides. Another anionic dipeptide, Glu-Ser, is also transported.

These data are shown in new panels of Figure 5 and a new supplementary Figure (Supplementary Figure 6).

During this analysis, we noticed that using unfragmented ions (SIM) to monitor Ala was often confounded by endogenous oocyte compounds. We thus used MS/MS fragmentation (MRM detection mode) to monitor single amino acids. The Ala graph in Figure 5C was modified accordingly.

2. It would be helpful to indicate the positions of zoomed areas in Fig. 7D using white boxes.

Response: Boxes for the zoomed-in areas were included in the figure 7D.

3. Some figures are incorrectly referenced in the text: Suppl. Fig. 6A-C is 7A-C, Suppl. Fig. 15 is 16.

Response: We corrected the wrong figure citations throughout the manuscript.

4. Remove ".) In fact," on page 21.

Response: "In fact" was removed because it was an unnecessary sentence fragment.

Reviewer #2:

Remarks to the Author:

Jungnickel et al. present findings on the deorphanization, structure and function of the lysosomal heterodimeric transporter complex MFSD1-GLMP. The authors isolated lysosomes for LC-MS/MS-based metabolomics analysis and functionally identified MFSD1-GLMP as a peptide transporter. The identification/deorphanization was achieved by recombinantly expressing the complex in frog oocytes and human cells. From the latter, protein was purified using detergent for biochemical, biophysical and structural analysis, and functional reconstitution into liposomes. Structures of both the apo- and His-Ala bound forms of MFSD1-GLMP were determined by cryo-electron microscopy (cryo-EM) at resolutions of 4.2 Å and 4.1 Å, respectively.

The submitted work represents a significant contribution with new and important findings for the scientific community - congratulations to the authors. However, there is room for improvement: The manuscript is written in a very concise and compact manner, and it is very dense in information and results, likely due to limitations imposed by the target journal. More detailed descriptions would enhance comprehension. Additionally, the figures often contain multiple panels, which can impede readability. Overall, the manuscript is well-written, but certain sections, especially in discussion and some captions, can benefit from adaptation/rephrasing. Please refer to the outlined issues below that will help improving presentation of the data and readability.

(1) Please mention the detergent (DDM/CHS) used for purification, nanoDSF and structure determination in main part. This will make reading easier and fluent not having to stop to check Materials and Methods for such an important information.

Response: We included this important information in the introduction of the purified protein used for NanoDSF experiments, liposome-based experiments, and structure determination.

(2) Figure 1 is suboptimal: Too many panels and unclear, e.g., too small. A possibility would be to split the Figure into two Figures, e.g., Figure 1 (panels A-E) and Figure 2 (panels F-H) - would also make thematically sense. See also specific feedback regarding panels G and H, below:

Response: We agree with the reviewer that this Figure is pretty full of data. We split figure panel 1H (nano-DSF screen) into two rows to increase readability. After consultation with the editor, all illustrations should be used in full size if possible. We, therefore, would like to stick to the first Figure version.

- Fig. 1G: Please make short horizontal colored lines thicker, colors are not clearly distinguishable.

Response: The colored lines were adjusted accordingly for better readability.

- Fig. 1H: Please split this panel into two rows or put into Supplementary Information. Currently, names of amino acids and dipeptides are difficult to read, too small, significant zooming-in necessary.

Response: We followed the reviewer's suggestion and split panel 1H into two rows for better readability and increased lettering size.

(3) Page 22, top, sentence: The strongest stabilization effects were observed for neutral dipeptides (e.g., Leu-Leu, $\Delta T_m = 14$ °C) and dipeptides that possess at least one positively-charged residue (e.g., Pro-Arg, $\Delta T_m = 12.1$ °C and His-Lys, $\Delta T_m = 12$ °C). Based on this finding, dipeptides with different physicochemical properties bind to MFSD1. However, only positively charged dipeptides are taken up (Figure 2). What is the mechanistic sense behind binding efficiently neutral ($\Delta T_m = 14$ °C) and negatively charged dipeptides (Suppl. Figure 2D, E), but not transporting them?

Response: We do not agree with the statement that only 'positively charged dipeptides are taken up' as we had shown by LC-MS that Leu-Ala is transported in the first submission. By using the thermal shift method, we could show that neutral and positively charged dipeptides induce the strongest stabilization effect and are, therefore, likely to bind to MFSD1. To investigate whether these peptides are also transported, we used three different transport assays: (i) oocytes (TEVC), (ii) oocytes (Mass spec), and (iii) liposomes (fluorescence).

These assays have their strengths and limitations, but our data (see Figure 5) now show convincing uptake of the neutral dipeptides Leu-Ala, Ala-Ala, and negatively charged dipeptides (Glu-Ala; Ala-Asp; Glu-Ser), supporting our statement that MFSD1 acts as a general dipeptide uniporter.

We also want to highlight that the interaction of a dipeptide with MFSD1 does not necessarily mean that a particular dipeptide is also taken up by the transporter. Therefore, we were careful in our analysis to highlight that the compound screen we present here was performed to give us an idea of which compounds MFSD1 might interact with and that we had to perform uptake experiments to confirm these findings. In a recent systematic study on the bacterial di- and tripeptide transporter DtpB (Kotov et al., 2023) it has been shown that 'binding' of peptides to the transporter not necessarily translate into 'transport'.

(4) Page 25: Here the determined K_m -values of His-Ser (119 μM) and His-Ala (24 μM) are presented. Affinities are relatively high (i.e., low K_m -values), compared to other shown kinetic parameters. How are these two K_m -values correlated to the observed ΔT_m from Figure 1H, e.g., His-Ser almost doesn't show thermostability, but has a significant and for peptide transporters relative high affinity?

Response: We do not observe a direct correlation between ΔT_m , K_m , or K_d , although the latter is directly determined based on thermal shift data. For example, the dipeptide Leu-Ala has an estimated K_d value of only 2 mM, although this ligand was highly stabilizing in the nanoDSF experiments. It should be noted that thermal

shift data were monitored using detergent-solubilized protein, whereas K_m measurements were performed in a lipid environment. Therefore, it is difficult to compare these numbers directly. We are careful to interpret ΔT_m values only as initial indicators of what type of compounds MFSD1 might interact with. The different transport assays presented in the manuscript provide evidence that dipeptides are substrates of MFSD1 and are thus used for further analysis.

(5) Fig. 3C: Please make short horizontal colored lines thicker, colors are not clearly distinguishable.

Response: The colored lines were adjusted accordingly for better readability.

(6) Suppl. Fig. 1, panel C: Please indicate in Figure legend SEC column type used. Furthermore, there is an important void resp. aggregates between 1-1.5 ml. Do the authors have an explanation for this? It is not a typical void peak, but remains almost constant until elution volume 1.5 ml.

Response: The column type has been added to the figure legend. The 'not typical void peak' contains aggregated MFSD1 and impurities. Furthermore, to avoid an additional concentration step before grid freezing for cryo-EM, the sample is typically concentrated to high concentrations before gel filtration, so higher oligomeric species cannot be excluded.

(7) In the context of wording: i) please avoid expressions such as "powerful hydrolysis" and "powerfully driven". Use more neutral adjectives and adverbs. ii) If "The first major" is mentioned, then the "second" and even "third" is expected by the reader. Please adapt. iii) Fig. 6H: Rather a lengthy caption, the information might be better placed in the main text.

Response: We removed " powerful " wording accordingly and adjusted the "First ... second" wording. We moved parts of the caption of Figure 6H to the main text.

(8) In general, there are numerous dipeptides tested, but it is not clear to the reader why certain dipeptides have been chosen for assays, structure and some not. Please provide some rationale, where possible, so that the reader understands the logic behind the procedure.

Response: During this study, a large amount of data was generated. While it is technically not feasible to study all available peptides (400 from the 20 proteinogenic amino acids) with all assays presented here, we decided to focus/follow up (cryo-EM, MD) on peptides that are transported in the oocyte and liposome-based assay. We tried to cover a broad spectrum of dipeptides with different chemical properties. In addition, the different assays needed dipeptides in various amounts; therefore, the commercial availability of the selected peptides in relatively high quantities from commercial vendors was also considered.

We tried to explain the rationale better:

" In contrast, several cationic dipeptides such as Ala-Lys, Arg-Ala, His-Ser, Arg-Pro (a dipeptide that was identified in $Mfsd1^{tm1d/tm1d}$ lysosomes) and, to a lesser extent, Lys-Pro and Pro-Arg, evoked a robust current, whereas neutral dipeptides (Leu-Ala; Ala-Ala) and an anionic dipeptide (Glu-Ser) had no effect (Figure 2G, Suppl. Figure 2B)."

We also collected single particle cryo-EM data on MFSD1-GLMP bound to the dipeptides His-Lys, Leu-Ala, and Lys-Val (see Reviewer Figure below). The reconstructions are of similar or lower quality compared to the presented His-Ala data set. In all cases, additional density is visible in the binding site (colored green), but the dipeptide's unambiguous placement is not possible. K_D -determination also revealed that despite a high thermal shift, the affinity of the neutral dipeptide Leu-Ala is in the mM range, likely explaining the diffuse density of this peptide in the reconstruction.

Reviewer Figure: Data set of GLMP-MFSD1 in the presence of 5 mM ligands (His-Lys, Leu-Ala, and Lys-Val). Local resolution is depicted for each CryoEM map (left column). Additional density appearing in the binding site with potential ligand fitted into the density is shown in green, while the protein density is shown in light blue (middle) column. GSFSC plots for each data set, the number of particles of the final reconstruction is given in the middle right column, and the K_D values determined via thermal shift assays are shown in the right column.

The reason why we used the His-Ala dipeptide for structure and MD work is stated in the main text.

(9) As mentioned in point (3), Leu-Leu has a high ΔT_m , but also Leu-Ala ($\Delta T_m \sim 12-13^\circ\text{C}$).

- Why did the author not determine the K_D of these interesting inhibitory compounds?

Response: We present K_D data on Leu-Ala in Supplementary Figure 1D.

- What was the rationale to choose His-Ala (Fig. 1H, $\Delta T_m \sim 6^\circ\text{C}$) for structure solution and not, e.g., Leu-Leu and/or Leu-Ala that showed higher ΔT_m , being most probably more protein stabilizing and more optimal for structure solution (e.g., at higher resolution) ?

Response: Please see comments to point 8 of Reviewer 2.

(10) Page 39: "...MFSD1 has a low affinity for its substrates in the mM range" This statement is misleading, since K_m values are indicated with ~24 to ~120 μM (Fig. 3E), i.e., values comparable to PHT1 and PHT2. Please revisit this statement.

Response: We changed the phrasing of this part to include all affinity measures reported in our study to reflect the diverse affinities we found for a subset of substrates where K_m values were determined.

Now it reads: ' Third, MFSD1 has affinities ranging from 24 μM to 4 mM depending on its substrates, whereas PHT1 and PHT2 operate in the 10 to 100 μM range (Custodio et al., 2023; Dong et al., 2023).'

(11) Comments regarding Figures:

- Suppl. Fig. 6A and 6B: GLMP migrates as a diffuse band, because glycosylation. It would be beneficial to mention this information for less experienced readers.

Response: We added the information that GLMP migrates as a diffuse band due to N-glycosylation to the main text.

- Suppl. Fig. 6C,F and Suppl. Fig. 13D: y-axes description is not clear "first derivative (Ratio)" or "Ration (first derivative)": of what? Furthermore, use one consistent y-axis naming/description in both Suppl. Figs.

Response: The axis description has been changed and explained ("First derivative F 350nm/330nm") and is now consistent throughout the figures.

- Suppl Fig. 9, comment: "Cyro-EM map (grey mesh) within a 2.5 Å radius around". Please clarify what is meant by "2.5 Å radius around".

Response: We rephrased this statement and hope it is clearer now. We are referring to the limits of the displayed EM map by using the Pymol command carve=2.5, thus limiting the display to only 2.5 Å around each atom of the displayed helices for better visualization.

- Fig. 6D: For better understanding, please color/highlight the "additional binding site density".

Response: We now aligned both maps and showed them in mesh (apo) and surface (His-Ala data set) representation at the same sigma level to highlight the additional density.

- Fig. 6E and Suppl. Fig. 10: The interaction D60 to His(+)-Ala is an important interaction. It would be beneficial to also display this residue in Fig. 6E and Suppl. Fig.10.

Response: The residue D60 is now included in Figure 6E and all panels of Suppl. Fig. 10.

- Fig. 6G: In the plot, certain mutants cannot be distinguished from each other, because the bar heights are very low (color of bar not visible). For better view, mutants might be labeled (also) below the bars.

Response: Every mutant was labeled accordingly, additionally below the bars.

- Fig. 7A, B and C: Image quality/resolution low, please be sure to provide better resolved panels in this Figure.

Response: We solved the issues with the image resolution, and high-resolution images are now provided in Figure 7.

- Fig. 7B and C, legend: Not a "salt bridge" between residue Y416 (MFSD1) and R292 (GLMP), rather H-bond - please check. Please also revisit the legend to Fig. 7: There is space for improvement.

Response: We corrected the mistake: The text of the figure legend now reads: "Besides the salt-bridge hydrogen-bond between residue Y416 (MFSD1) and R292 (GLMP)"

(12) Typos:

- Page 21: "In fact, The..." - lower case T.

Response: "In fact" was removed because it was unnecessary.

- Page 33: "... Arg181, respectively (Suppl. Figure 1)" - probably Suppl. Figure 10 (not 1) ?

Response: This has been corrected.

Reviewer #3:

Remarks to the Author:

I am reviewing primarily as a metabolomics technical expert.

1. LC-MS/MS-based analysis of dipeptides from tissues: sample prep. no protease inhibitors were used in sample prep - might the dipeptides be generated during and/or after homogenization and before ethanol addition (which denatures and deactivates proteins)?

Response: Purified intact lysosomes were immediately frozen in liquid nitrogen and kept at -80 °C. For metabolomic analysis, we used cold 80% methanol to quench metabolism and any enzymatic reaction immediately after taking cells from the incubator. The harsh chemical nature of methanol is even more potent than protease inhibitors. While we cannot formally rule out this possibility, we think it's very unlikely.

2. LC-MS/MS-based analysis of dipeptides from tissues: AccQ acetonitrile solution - what is this? Is this commercial accq-tag kit? Please describe in more detail. It appears there are also numerous other products in the methods which are ambiguously defined. (i.e. 96-black chimney deep well plate).

Response: 6-aminoquinolyl-N Hydroxysuccinimidyl carbamate (AccQ) was purchased from Toronto Research Chemicals (Toronto, ON, Canada). It is not AccQ-tag kit. These details are indicated in the methods section. The AccQ powder was dissolved in acetonitrile giving 3 mg/mL.

For clarity, we added details (manufacturer) for several products in the methods section.

3. LC-MS/MS-based analysis of dipeptides from tissues: LC-MS/MS analyses - "detection of monovalent and divalent ions". can you clarify what this means? Generally we describe molecular ions as singly or doubly charged, not by valency, which generally refers to atoms. But i am not fully clear on what you mean.

Response: We followed the reviewer's comment and changed the wording to singly and doubly charged ionic species.

4. Untargeted metabolomics and targeted metabolite quantitation: '1.5-3 μ L' injections - why a range? What samples used different injection volumes and why? Did you account for differences in injection volume before plotting and performing statistical analysis (Fig 1 E)? You mention full scan mode with polarity switching - what was the duty cycle under these conditions? That is, how many positive mode (or negative mode) scans per second did you observe?

Response: We would like to apologize for not specifying the exact injection volumes for the metabolomics samples. The injection was 1 μ l, and the range of "1.5-3 μ l" was stated before because it is our normal range of injection for samples run in our lab. We have corrected this technical information in the manuscript.

For full scan mode with polarity switching, we applied the standard polarity switching setting from Orbitrap Tribrid Mass Spectrometer (ThermoFisher Scientific), which used 200 ms for polarity switching and 1 s for scan at each polarity.

5. Untargeted metabolomics and targeted metabolite quantitation: "known metabolite standards (MSMLS, Sigma-Aldrich) for targeted metabolite quantification" were any isotopically labelled internal standards used? It isn't clear that they were. Please provide more detail as to how quantification was performed. I would have to guess that you are using a calibration curve in neat solution and quantifying signal from complex matrix (sample extract). This is generally not a reliable quantitation method, as the sample may generate appreciable matrix effects (signal suppression, generally, but signal enhancement, at times). isotopically labelled internal standards are commonly used to account for this and enable more accurate quantification. Additionally, was a dose response curve used? If so, how many points? Please describe the quality of the curve. There is far too little information provided to know whether we can trust the quantitative values at all.

Response: We appreciate the reviewer's comment about the use of isotopically labeled standards. Indeed, we performed ILAA standardization as the reviewer suggested. We have also edited the manuscript to reflect these clarifications. We have also prepared dilutions for samples to evaluate quantitation responses. To be specific, dilutions at 0.01x, 0.1x, and undiluted samples were used as we did in other published work (Laqtom et al., Nature, 2022). The quality of this assessment is good as the R^2 values are generally > 0.99 , as shown in the following technical figures.

6. There is no mention of any sort of metabolomics quality control. pooled quality control samples, internal standard, solvent or sample preparation blanks, replicate injections, standard reference materials - these are all methods to demonstrate data quality, and the authors appear to have not reported use of any of them. I do realize that the metabolomics served as the discovery set, and the results from it were validated through extensive additional experimentation, so the lack of apparent quality control may not be too detrimental. However the authors should describe any quality control that they did perform. Also, recognize that strong quality control approaches in the future will ensure you pursue fewer artifactual results. I highly

recommend depositing metabolomics data in an established repository such as MetaboLights, Metabolomics Workbench, MetaboBank, etc.

Response: As the reviewer suggested, we performed quality control on different aspects of the metabolomic study. We mentioned previously that pooled quality control samples and internal standards were used. The solvent and sample preparation blanks were also rigorously monitored, and we have stated this in the method section. Replicate injections were not performed due to limited instrumental time and the reason specified by the reviewer – results from this discovery step were later validated by extensive follow-up characterizations. We have edited the manuscript where necessary to reflect these updates. Last but not least, we will follow the reviewer's suggestion and deposit our raw data to Zenodo, a publicly available repository similar to the ones mentioned by the reviewer (10.5281/zenodo.10839783).

“Untargeted metabolomics and targeted metabolite quantitation”

Three replicates of lysosome-enriched samples from each genotype were submitted for untargeted metabolomics. The polar metabolites were extracted using cold 80% methanol (v/v) with isotopically labeled amino acids (Cambridge Isotope Laboratories MSK-A2-S) as internal standards and profiled using a Thermo Fisher Scientific ID-X Tribrid mass spectrometer with an ESI probe.

Reviewer #4:

Remarks to the Author:

Jungnickel and coauthors proposed an interdisciplinary approach combining metabolomics, in vitro electrophysiology, Cryo-EM structure determination, modeling, and Molecular Dynamics (MD) simulations to characterize the function of MFSD1 protein, in interaction with GLMP, as a dipeptide transporter. This is a very well designed work which allows understanding from the function of this transporter to its structural characterization at the molecular scale. This reviewer expertise mainly lies in molecular dynamics simulations and structural biology even if they have coordinated interdisciplinary projects. Thus, they can appreciate the extent of this work.

Overall, this work is very interesting and well conducted. The manuscript is clear. The MD simulations were properly set up and ran. 300ns is not a very extensive time to explore protein-ligand interaction but the replicas are useful to assess the stability of these interactions. This reviewer has few points that need to be clarified before considering this manuscript acceptable for publication.

Major points:

1- It is mentioned in the introduction that lysosomal membrane contains a large fraction of glycolipids yet neither in vitro assays (reconstituted liposomes) nor molecular dynamics simulations include glycolipids. This glycolipids can affect protein function, by shielding some surface residues, or even limit the diffusion of peptides, especially the charged ones. So, at least, it would be important to discuss why these lipids were not included.

Response: The effect of glycolipids on MFSD1's activity is interesting and should be studied in future experiments. However, with our assays, we anticipated obtaining proton-tight liposomes. Therefore, we used established lipid mixtures that were proven to work for this purpose and at the temperatures we used for the experiments. However, we agree that mimicking the lipid composition to resemble the membrane composition of a lysosomal membrane as closely as possible would be an ideal scenario; it is, however, not feasible for the questions we tried to answer with these assays. Our approach was to monitor possible proton-coupling by MFSD1 using a pH-sensitive dye. Since this approach worked in this study system (as seen by little to no proton leakage under a membrane potential of roughly -100 mV in our assays throughout the experiment), we continued this path forward. We also want to highlight that there is an agreement between the oocyte assays, where the protein resides in a more natural lipid mixture, and our liposome assays, which is an artificially created lipid environment.

It should also be mentioned that the lysosomal membrane contains glycolipids, but very little is known about the exact (glyco-) lipid composition (exact glycolipid species). Moreover, glycolipids are unevenly distributed between intraluminal membranes and the limiting membrane. Therefore, proper modeling of a "physiological" lysosomal membrane is challenging.

We included a sentence in the Results part:

*"As an alternative in vitro approach, the transport activity was characterized using purified MFSD1WT purified in DDM/CHS and then reconstituted into **established POPE:POPG:CHS (3:1:1) lipid vesicles (Figure 4a)**, because the exact lipid composition of the lysosomal membrane is poorly understood."*

2- Related to question 1, can the authors explicit a bit more the choice of lipid compositions in liposomes and in MD simulations.

- For the former, why the choice of POPE:POPG:CHS at a 3:3:1 ratio ?

Response: We used the stated POPE:POPG mixture to obtain proton-tight liposomes since our experiments are monitoring possible proton-influx events by the studied transporter. Although this lipid mixture resembles that of E. coli lipid composition, it has repeatedly been shown to result in proton-tight liposomes feasible for uptake assays using pH-sensitive dyes (see <https://pubs.acs.org/doi/10.1021/bi201897t#>, <https://www.nature.com/articles/s41467-023-38120-5#Sec9>, <https://doi.org/10.1016/j.celrep.2023.112831>). We chose to add cholesterol to the mixture to mimic the presence of cholesterol in the lysosomal membrane. We know that the liposome composition does not fully reflect the cell's natural lipid environment of MFSD1. We used the chosen ratio of 3:1:1 on the one hand from the usual ratio of POPE:POPG of 3:1 from previous experiments and used CHS at 20% (w/w) based on liposome assays performed to study EAAT3 (<https://www.nature.com/articles/s41467-023-38120-5#Sec9>).

Furthermore can the authors precise what is CHS ? Cholesteryl hemisuccinate ? If yes, why not have used directly cholesterol ?

Response: CHS refers indeed to Cholesteryl hemisuccinate. Our reasoning behind this decision was twofold. We initially tried the addition of cholesterol to the liposome

mixture but needed help to get a homogeneous mixture. When searching for an alternative, we found that CHS (=Cholesteryl hemisuccinate) can be used as a substitute instead (<https://www.ncbi.nlm.nih.gov/pmc/articles/PMC8111416/>) and indeed could successfully prepare the lipid mixture with CHS present. Additionally, it has the benefit that we have CHS present in our detergent-solubilized protein sample so that at least this compound is present in our liposome assays.

- For the latter, the composition of POPE:POPC:CHOL:PSM at a ratio of 2:3:3:2 does not correspond neither to liposome lipid composition nor lysosome membrane composition (as glycolipids are missing).

Response: We thank the reviewer for bringing this issue to our attention. The significance of lipid composition is widely acknowledged and agreed upon. Nevertheless, there are two crucial aspects to consider. Our primary objective in both the simulation and experiment was to recreate a native-like environment, although not with an exact replication of the lysosomal membrane. Furthermore, and of greater importance, our simulations primarily investigate the dynamics of ligands within the binding site rather than the ligand binding process itself. This way, there is a significant distance between the ligand in the binding site and the lipid molecules in the membrane (the minimum distance between the ligand and any lipid molecule in the membrane is around 11 Å). Consequently, we anticipate that the presence of a specific lipid species would have no/minimal impact on the outcomes of our simulations.

See also point 1 regarding the glycolipids in the lysosomal membrane.

3- Same idea as glycolipids, why the glycosylation was not fully modelled especially when studying the stability of MFSD1-GLMP interactions using MD simulations (Figure S16). Indeed, as presented in figure 6-C, there is one N-glycosylation at the GLMP surface that can potentially point towards MSFD1 binding site. If this reviewer is right, the CryoEM approach only allowed determining a small part of this long (up to 14 sugars ?) oligosaccharide.

Response: We agree with the reviewer that using the glycosylation sites in the MD simulation would be interesting in probing their effect on the protein. Additionally, the reviewer is correct in assuming that we did not determine the full glycosylation pattern of GLMP in the structure and our data. Therefore, the glycosylation sites were not used in the MD simulations.

4- In figure S11, only few distances were monitored along the course of the simulation. It would be more insightful to monitor distances between different parts of the peptide with all the residues presented in the "start" column in Figure S10. This can be done by using matrix colored in function of distance. This will really help understanding which residues are involved in stable interactions.

Response: We followed the reviewer's suggestion and included a matrix-colored plot to visualize distance as a function of distance (new supplemental Figure 12d).

5- In Figure S14, it would have been useful to perform at least 300ns of MD simulations to check the stability of the AlphaFold model and assess the opening/closing of the canal using tools such as Hole (which is included in

MDAnalysis:

https://docs.mdanalysis.org/1.1.1/documentation_pages/analysis/hole2.html).

Response: The AlphaFold models shown in this study were subjected to a relaxation step as part of the prediction procedure of AF2 (<https://www.nature.com/articles/s41586-021-03819-2>). The authors do not see the rationale behind additional MD simulations for these models.

6- It is necessary to release more data from the modelling. Only the Alphafold model and the last structure is not enough.

This reviewer was not able to find the Zenodo pages associated to these models.

Response: The access restrictions to the Zenodo pages have been released. We apologize.

Minor points:

Table S3, there are only 3 replicates for each system. As mentioned in Figure S10, there are 2 positions for each peptide. So, for the peptide systems is it more 2 (each pose) x 3 x 300 ns?

Response: We thank the reviewer for pointing out the mistake in the table. We have corrected the table in the revised form to reflect the correct number of simulations (now 2 x 3 x 500 ns for each system setup).

Reviewer #5

Comments to the Authors

This paper reports on a program to determine the function of the lysosomal MFS transporter MFSD1. I have been asked to review the electrophysiological portions of the experiments, and will focus most efforts on those parts of the paper.

Briefly, metabolomic screening of lysosomes from MFSD1 KO mice demonstrated an increase in the dipeptides Arg-Pro and Pro-Lys, suggesting an involvement in dipeptide transport, a conclusion supported by thermal stabilization of the purified MFSD1 protein by a series dipeptides. Targeting the protein (and its partner GLMP) to the plasma membrane in *Xenopus* oocytes allowed measurements of transport currents in the presence of some dipeptides, which are dose-dependent and activated at low extracellular (luminal) pH. Further experiments with reconstituted, purified protein in liposomes revealed electrogenic acidification induced by certain titratable dipeptides. Further oocyte experiments detect simultaneous acidification and transport currents in the presence of titratable peptides. CryoEM structures at limited resolution reveal the expected MFS fold, along with possible substrate density, and MD experiments suggest a possible orientation for the substrate. Final experiments (somewhat superficially) explore the interaction between MFSD1 and GLMP.

Overall, the conclusion that MFSD1 is a dipeptide transporter is very strongly

supported. The initial metabolomic mass spec experiments point strongly in this direction and then are nicely reinforced by the experiments in *Xenopus* oocytes and with purified protein reconstituted into lipid vesicles. However, I disagree with the interpretation of the experiments concluding that the protein is a simple dipeptide uniporter. Indeed, much of the data presented here suggests otherwise—especially the results in Figure 3 and 4 demonstrating a decrease in pH accompanying transport of Histidine- and Glutamate-containing dipeptides. These results suggest to me that transport of these titratable peptides is indeed proton-coupled, though I agree with the authors that such coupling is unlikely for the lysine-containing peptides where no pH change is seen. If true, this would be a very interesting finding—that transport is coupled to the proton gradient for some peptides but not for others. To truly distinguish between a uniporter and a sometimes-coupled transporter would require assessing the ability of a pH gradient to DRIVE the uptake of a peptide, like His-Ser. A true uniporter could never concentrate its substrate, it could only mediate transport down an electrochemical gradient. For example the liposome system could be used with equal K⁺ on both sides (no potential), equal concentrations of His-Ser on both sides, and a large inwardly directed pH gradient, ideally measuring substrate concentration alongside the pH. This clearly would represent substantial effort and is important, but not necessarily for the current paper, which already represents an impressive and substantial body of work. It would be acceptable to acknowledge the ambiguity, soften the conclusion, and recognize that the distinction between uniport and substrate-dependent active transport requires further work.

Response: See the full response to the next point, which also addresses this (first major) point.

A second major issue is the use of ΔpH in Figure 4 to estimate stoichiometry. In contrast to the currents, which are 1:1 with charges moved, the ΔpH is highly dependent on the cytoplasmic buffer capacity. Because of buffering, there is no reason to expect that the ΔpH is linearly related to the actual number of protons transported, much less to expect a proportionality constant of 1. This is particularly true if the ΔpH is near 1 unit, as suggested by the ~40 mV changes in voltage. The authors are vague about the calibration in the methods "The potential difference between the two inputs tested in diverse buffers..." but this fails to account for the buffer capacity of the oocyte cytoplasm. The stoichiometry arguments presented here are not central to the main conclusions and should be removed.

*Response: Regarding the second point, we did not expect that every proton entering the oocyte would change the internal pH without buffering by cytosolic components nor without stimulating proton export by the endogenous Na⁺/H⁺ exchanger at the plasma membrane. This is the reason why we used His-Ala applied to the **same oocyte** with the ion-selective electrode impaled at the **same position** to normalize the acidification responses of the tested dipeptides. On the other hand, we showed in a previous study that the acidification rate recorded by this technique varies linearly with the proton influx above a ~100-nA current threshold.*

[Redacted]

See Figure 1C in Leray et al, PNAS 2021 (<https://doi.org/10.1073/pnas.2025315118>):

Beyond this threshold, there is a linear correlation between the acidification rate and the transport current. The linear part of this relationship justifies the use of our approach to detect differences in the level of proton release across distinct electrogenic MFSD1 substrate. Therefore, the finding that His-Glu induces acidification/current ratio about twice higher than His-Ala and His-Ser (Fig.4C-F) is a substantial piece of evidence, which cannot be ignored before drawing conclusions about the transport mode of MFSD1.

This leads us to the first point raised by Reviewer #5. This point challenges our interpretation that MFSD1 operates as a classical ('simple') uniporter, suggesting that it could instead shift between a uniport mode and a H⁺ symport mode depending on the dipeptide. We agree with Reviewer #5 that a bioenergetical criterion, that is probing the capacity of MFSD1 to concentrate, or not, translocated substrates, would provide key evidence to discriminate between the two models. However, this criterion cannot be easily tested in cellular assays, because cytosolic cleavage of the dipeptide prevents reaching thermodynamic equilibrium, while the proteoliposome assay would require to measure internal dipeptide concentrations instead of a pH-sensitive dye readout.

With this caution in mind, we can however use the parsimony principle (Occam's razor) and the fact that the acidification/current ratio of His-Glu is about twice that of His-Ala and His-Ser to help prioritize the two models.

Indeed, to account for these differences in the acidification/current ratio, a 'sometimes-coupled' transporter should posit that MFSD1 shifts between three, not two, transport modes depending on the substrate: a uniport mode for Lys-Ala; a 1:1 H⁺/dipeptide symport for His-Ala and His-Ser; and a 2:1 H⁺/dipeptide symport for His-Glu. In contrast, our 'simple' uniport model, where protons are carried by the dipeptide rather than through a substrate-coupled H⁺ pathway, would easily account for the three levels of this ratio since Lys-Ala, His-Ser and His-Glu (ratios of -0.05 ± 0.03 ; 1.2 ± 0.1 ; and 2.4 ± 0.3 , respectively) have zero, one and two titratable residues (His: pKa = 6.0; Glu: pKa = 4.1) able to protonate in the extracellular pH 5.0 buffer and deprotonate when they face the cytosol at pH 7.2.

A 'simple' uniport model thus represents the simplest explanation for currently available data.

These two points are addressed in the revised manuscript in the following manner. We now present the rationale behind the experiments shown in Fig. 4C-G in a more accurate manner, with explicit mention of the acidification/current relationship found in our previous study. We also describe the His-Ala-normalized acidification/current ratio as a 'rough estimate' rather than a 'proxy' of the proton stoichiometry to account for the existence of a threshold in this relationship. The existence of this threshold is also considered when interpreting the acidification/current ratio of His-Glu is higher (2.4 ± 0.3) than the predicted ratio of 2.0 for the cationic (fully protonated) form of the dipeptide:

The deviation above the theoretical ratio of 2.0 may result either from the aforementioned threshold in the acidification/current relationship or from significant transport of His-Glu in the predominant zwitterionic form, which would carry additional protons in an electroneutral manner (Figure 4G).

We also highlight that the uniporter model is the simplest, not the exclusive, interpretation of our data:

"Taken together, these data show that MFSD1 transports cationic dipeptides with or without a concomitant acidification whose presence and intensity depend on the number of titratable side chains in the dipeptide. The simplest interpretation is that MFSD1 is not coupled to protons, as initially thought, and that it operates instead as a dipeptide uniporter."

And we acknowledge the word of caution of the reviewer and soften the conclusion is in the Discussion by mentioning that the 'sometimes-coupled' transporter model cannot be excluded.

We thank the reviewer for raising these points and offering an opportunity to improve the manuscript and provide a more rigorous presentation of our data.

Other comments: Fig 2 2C The data are convincing that most current comes with coexpression. However, the presentation of the MFSD1-only data is confusing. I suspect that the two entries for MFSD1 alone show a representative trace for a positive oocyte, and representative trace for a negative oocyte, with the numbers corresponding to the number of oocytes where this is seen, however, this is far from clear from reading the Figure and legend. In addition, examining the data points on the bar graph makes it look like maybe only two points have nonzero current. These results need better explanation/presentation.

Response: We agree, and for clarity, we removed the trace of the MFSD1-only oocyte, which showed a small response because, indeed, only two out of 14 oocytes showed a significant response to Lys-Ala. The non-responders are thus more representative. The non-responders are mentioned in the figure legend:

"Only two out of 14 oocytes expressing only MFSD1L11A/L12A-EGFP responded to Lys-Ala."

Figure 2E: What is the odd deflection early in the pH 7.0 trace? Is that reproducible?

Response: The odd deflection was a perfusion artefact, probably a bubble, during the exchange of solutions. We replaced the paired pH 7.0 vs. pH 5.0 traces by another representative oocyte.

Figure 2/3. The Kms shown in Figure 2 (for Lys-Ala) and those in Figure 3E/F for His-Ser and His-Ala are 20-fold apart. Is there a substrate for which the two assays can be compared? Also, in Figure 3, the v_{max} values, in units of $\Delta F_{norm}/sec$ don't mean anything since these are arbitrary units—these should be removed. It would be interesting to know the Hill coefficients of these fits if the data are well-determined enough to reliably estimate them. The individual measurements need to be shown on these plots in addition to the means/std dev.

Response: We reanalyzed the transport data with the Hill equation and obtained Hill coefficients of $n = 1.11 \pm 0.29$ (for His-Ala) and $n = 0.73 \pm 0.53$ (for His-Ser, slightly poorer data quality). Based on these data, we suggest keeping the current MM-analysis. We have removed the v_{max} values as suggested. Since the transport of the dipeptide Lys-Ala cannot be monitored in the liposome assay (no deprotonation event) we currently do not have KM-data for direct comparison between the two assays. The observed difference (20-fold Km between Ala-Lys and His-Ala/His-Ser)

might also originate from the different setups used (oocytes vs. liposome, different buffer conditions, etc.). The individual data points (instead of only the mean) are now included in Figure panels 2d and 3e.

Figure 4 I'm not sure why the pH traces are shown here in mV when conversion to ΔpH is straightforward

*Response: We tried in the past to calibrate the pH electrode within the oocyte but failed to do so because, among other reasons, ionophores required for this calibration are poorly efficient in these cells. We are unaware of any publication reporting the intracellular calibration of ion-selective electrodes in *Xenopus* oocytes. This is why we express all H^+ -selective electrode data in mV or mV/s rather than in pH or $[\text{H}^+]$ values deduced from the calibration in solution, which may differ from the response in the cytosol.*

Figure 5. The presentation of the putative substrate density is limited and unconvincing. It's pretty difficult to see that density directly in the EM maps, and Figure 5D is a bit confusing. For a start, perhaps it would be easier to visualize if the orientation of the structure in Fig 5D was the same as that in 5C (right panel). Could difference density be presented to highlight the area of interest? Why are the maps in the accompanying mrc files not aligned to each other to make comparisons easier for the reader?

Response: We have modified Figure 5D. Please also see the response to Reviewer 2. We apologize that we could not provide the mrc files in an aligned format since this is not conventionally done, as we directly used the output of the maps of CryoSparc. Additionally, the deposition to the PDB requires us to also submit the half maps of the data processing jobs from CryoSparc, which we would then have to align with the subsequent models. If we were to realign all our data to either one of the maps, we would have to redo the entire pdb deposition. We apologize for any inconvenience this might cause.

Molecular Dynamics. These simulations are for pretty short times (300ns) and though in the presented data one orientation does seem more stable, it's hard to be confident that even the "stable" state would be that way. A minimum of maybe 500 us would be more convincing, as would a quantitative analysis. For Lys-Ala, though the conf1 distance is larger, it doesn't appear any less stable than conf2, and in two of the traces the substrate seems to move closer to E150 and becomes quite stable there; similarly the Cterminal of the peptide in that simulation seems equally unstable in both configurations.

Response: We thank the reviewer for raising this issue. To enhance our sampling, we have extended each replica to 500 ns, which shows a similar trend. The reviewer's observation about the distances is on the spot. However, to interpret the data, we looked at all the simulations. First, we see partial or complete dissociation of the ligand only in Conf1 (3 instances out of 12 replicas: LA replica 1, H0A replica 2, and Ka replica 2), while no dissociation was observed in Conf2. Furthermore, in all simulations of Conf2, E150-Nter is stable, while at least 2 replicas (apart from the partial/full dissociated ones) do not have stable E150-Nter interactions (H0A replica 1 and KA replica 1). About the R181-Cter, we agree with the reviewer that the

interaction is not as stable as the E150-Nter interaction. However, this interaction, to some extent, is ligand-dependent. While it is more stable in H0A and H+A, it is less stable in LA, especially in KA. In the case of KA, it is mainly because of the formation of an internal hydrogen bond between the sidechain of Lys and the Cter.

Thank you for your time and consideration! We look forward to hearing back from you.

With our best regards,

Markus Damme, PhD

and

Christian Löw, PhD

and

Bruno Gasnier, PhD

Decision Letter, first revision:

Our ref: NCB-A53205B

10th April 2024

Dear Dr. Damme,

Thank you for submitting your revised manuscript "MFSD1 with its accessory subunit GLMP functions as a general dipeptide uniporter in lysosomes" (NCB-A53205B). It has now been seen by the original referees and their comments are below. The reviewers find that the paper has improved in revision, and therefore we'll be happy in principle to publish it in Nature Cell Biology, pending minor revisions to satisfy the referees' final requests and to comply with our editorial and formatting guidelines.

We will now begin performing detailed checks on your paper and will send you a checklist detailing our editorial and formatting requirements in about ~2 weeks. Please do not upload the final materials and make any revisions until you receive this additional information from us.

Thank you again for your interest in Nature Cell Biology. Please do not hesitate to contact me if you have any questions.

Sincerely,

Melina

Melina Casadio, PhD
Senior Editor, Nature Cell Biology
ORCID ID: <https://orcid.org/0000-0003-2389-2243>

Reviewer #1 (Remarks to the Author):

The authors have clarified my queries, and the revised manuscript is now describing MFSD1 as a general lysosomal dipeptide uniporter which is consistent with the data. The dataset is much improved and the manuscript could be recommended for publication in the current form. Proofreading of the text is required to amend inconsistencies in referencing different figure panels and correct small typographical errors.

Reviewer #2 (Remarks to the Author):

The authors have satisfactorily addressed my concerns and open questions. Congratulations to the authors for this heroic contribution.

Reviewer #3 (Remarks to the Author):

The authors have responded to my concerns, and i support publication.

Reviewer #4 (Remarks to the Author):

While the addition of analysis (SI fig 12), the extension of the MD simulations to 500ns, and the release of MD trajectories clearly improved the manuscript, this reviewer is a bit concerned by the limitation of lipid types in MD simulations which may have effect on peptide binding site even if not close to the binding site (allosteric effects). So, for readers interest, I would recommend to mention in the manuscript why the authors chose to model such a simple membrane. Except this minor point, I consider the manuscript acceptable for publication in Nature Chemical Biology from the MD perspective.

Reviewer #5 (Remarks to the Author):

The authors have substantially edited the manuscript and have fully addressed most of my comments. However, there remains a misunderstanding regarding my most substantial input about the MFSD1 being a uniporter. The authors' data clearly demonstrates that the protein acts as a proton-coupled transporter for acidic substrates. I agree with their conclusion that the protons are carried by these substrates—but that still manifests as a form of coupling to protons: a proton gradient should indeed drive these substrates against their gradients. In contrast to the conclusion of the authors, though (rebuttal p17), no additional "modes" of transport are required—only that the transporter prefer the protonated forms of these substrates. Thus, the identity of the substrate determines the coupling without needing any additional modifications of the transport cycle. I persevere on this issue because I believe that this will be a prominent paper and I believe that it's worth nailing this idea so as not to cause confusion in the field. For a particularly dramatic example of this the authors might look at Basilio et al (JGP 2009 doi: 10.1085/jgp.200810170) where the large-bore Anthrax toxin channel acts as a proton-protein symporter because it strongly disfavors entry of negatively charged residues on its proteinaceous substrate and prefers the protonated forms of those sidechains. Protons are therefore picked up on one side of the membrane and deposited on the other, coupling the movement of substrate protein to that of protons.

Decision Letter, final checks:

Our ref: NCB-A53205B

19th April 2024

Dear Dr. Damme,

Thank you for your patience as we've prepared the guidelines for final submission of your Nature Cell

Biology manuscript, "MFSD1 with its accessory subunit GLMP functions as a general dipeptide uniporter in lysosomes" (NCB-A53205B). Please carefully follow the step-by-step instructions provided in the attached file, and add a response in each row of the table to indicate the changes that you have made. Ensuring that each point is addressed will help to ensure that your revised manuscript can be swiftly handed over to our production team.

In recognition of the time and expertise our reviewers provide to Nature Cell Biology's editorial process, we would like to formally acknowledge their contribution to the external peer review of your manuscript entitled "MFSD1 with its accessory subunit GLMP functions as a general dipeptide uniporter in lysosomes". For those reviewers who give their assent, we will be publishing their names alongside the published article.

Nature Cell Biology offers a Transparent Peer Review option for new original research manuscripts submitted after December 1st, 2019. As part of this initiative, we encourage our authors to support increased transparency into the peer review process by agreeing to have the reviewer comments, author rebuttal letters, and editorial decision letters published as a Supplementary item. When you submit your final files please clearly state in your cover letter whether or not you would like to participate in this initiative. Please note that failure to state your preference will result in delays in accepting your manuscript for publication.

Cover suggestions

COVER ARTWORK: We welcome submissions of artwork for consideration for our cover. For more information, please see our guide for cover artwork.

Nature Cell Biology has now transitioned to a unified Rights Collection system which will allow our Author Services team to quickly and easily collect the rights and permissions required to publish your work. Approximately 10 days after your paper is formally accepted, you will receive an email in providing you with a link to complete the grant of rights. If your paper is eligible for Open Access, our Author Services team will also be in touch regarding any additional information that may be required to arrange payment for your article.

Please note that *Nature Cell Biology* is a Transformative Journal (TJ). Authors may publish their research with us through the traditional subscription access route or make their paper immediately open access through payment of an article-processing charge (APC). Authors will not be required to make a final decision about access to their article until it has been accepted. Find out more about

Transformative Journals

Please use the following link for uploading these materials:
[Redacted]

Best regards,

Kendra Donahue
Staff
Nature Cell Biology

On behalf of

Melina Casadio, PhD
Senior Editor, Nature Cell Biology
ORCID ID: <https://orcid.org/0000-0003-2389-2243>

Reviewer #1:

Remarks to the Author:

The authors have clarified my queries, and the revised manuscript is now describing MFSD1 as a general lysosomal dipeptide uniporter which is consistent with the data. The dataset is much improved and the manuscript could be recommended for publication in the current form. Proofreading of the text is required to amend inconsistencies in referencing different figure panels and correct small typographical errors.

Reviewer #2:
Remarks to the Author:

The authors have satisfactorily addressed my concerns and open questions.
Congratulations to the authors for this heroic contribution.

Reviewer #3:
Remarks to the Author:
The authors have responded to my concerns, and i support publication.

Reviewer #4:
Remarks to the Author:
While the addition of analysis (SI fig 12), the extension of the MD simulations to 500ns, and the release of MD trajectories clearly improved the manuscript, this reviewer is a bit concerned by the limitation of lipid types in MD simulations which may have effect on peptide binding site even if not close to the binding site (allosteric effects). So, for readers interest, I would recommend to mention in the manuscript why the authors chose to model such a simple membrane. Except this minor point, I consider the manuscript acceptable for publication in Nature Chemical Biology from the MD perspective.

Reviewer #5:
Remarks to the Author:
The authors have substantially edited the manuscript and have fully addressed most of my comments. However, there remains a misunderstanding regarding my most substantial input about the MFSD1 being a uniporter. The authors' data clearly demonstrates that the protein acts as a proton-coupled transporter for acidic substrates. I agree with their conclusion that the protons are carried by these substrates—but that still manifests as a form of coupling to protons: a proton gradient should indeed drive these substrates against their gradients. In contrast to the conclusion of the authors, though (rebuttal p17), no additional "modes" of transport are required—only that the transporter prefer the protonated forms of these substrates. Thus, the identity of the substrate determines the coupling without needing any additional modifications of the transport cycle. I persevere on this issue because I believe that this will be a prominent paper and I believe that it's worth nailing this idea so as not to cause confusion in the field. For a particularly dramatic example of this the authors might look at Basilio et al (JGP 2009 doi: 10.1085/jgp.200810170) where the large-bore Anthrax toxin channel acts as a proton-protein symporter because it strongly disfavors entry of negatively charged residues on its proteinaceous substrate and prefers the protonated forms of those sidechains. Protons are therefore picked up on one side of the membrane and deposited on the other, coupling the movement of substrate protein to that of protons.

Author Rebuttal, first revision:

Prof. Dr. Markus Damme, Olshausenstraße 40, 24098 Kiel

Editorial Board
Nature Cell Biology
Melina Casadio, Senior Editor

Bearbeiter/in, Zeichen
PVO

Mail, Telefon, Fax
mdamme@biochem.uni-kiel.de
tel +49(0)431-880-2218

Biochemisches Institut

Geschäftsführender Direktor:
Prof. Dr. Becker-Pauly

Paketanschrift:
Eduard-Buchner-Haus
Otto-Hahn-Platz 9, 24118 Kiel

Postanschrift:
Olshausenstraße 40, 24098 Kiel

<https://www.uni-kiel.de/Biochemie/scripte/dynamic/groups/damme/damme.php>

Datum
27.04.2024

Resubmission of our revised research article to *Nature Cell Biology* (NCB-A53205B)

Dear Melissa,

Please find enclosed the resubmitted version of our manuscript entitled "MFSD1 with its accessory subunit GLMP functions as a general dipeptide uniporter in lysosomes" (NCB-A53205B). We would like to thank the five reviewers for the second round of review and the overall very positive feedback in response to our revision. Please find below the response to the remaining points:

Editors' / Reviewers' Comments:

Reviewer #1:

Remarks to the Author:

The authors have clarified my queries, and the revised manuscript is now describing MFSD1 as a general lysosomal dipeptide uniporter which is consistent with the data. The dataset is much improved and the manuscript could be recommended for publication in the current form. Proofreading of the text is required to amend inconsistencies in referencing different figure panels and correct small typographical errors.

We thank the reviewer for the positive feedback. We proofread the manuscript, and the different figure references should now fit.

Reviewer #2:

Remarks to the Author:

The authors have satisfactorily addressed my concerns and open questions.

Congratulations to the authors for this heroic contribution.

We thank the reviewer for the positive feedback.

Reviewer #3:

Remarks to the Author:

The authors have responded to my concerns, and i support publication.

We thank the reviewer for the positive feedback.

Reviewer #4:

Remarks to the Author:

While the addition of analysis (SI fig 12), the extension of the MD simulations to 500ns, and the release of MD trajectories clearly improved the manuscript, this reviewer is a bit concerned by the limitation of lipid types in MD simulations which may have effect on peptide binding site even if not close to the binding site (allosteric effects). So, for readers interest, I would recommend to mention in the manuscript why the authors chose to model such a simple membrane. Except this minor point, I consider the manuscript acceptable for publication in Nature Chemical Biology from the MD perspective.

We mention the use of the (simple) lipid bilayer in the results:

“To further investigate peptide binding, we performed MD simulations in a simple lipid bilayer reflecting that of the liposomes in the presence of different dipeptides.”

Reviewer #5:

Remarks to the Author:

The authors have substantially edited the manuscript and have fully addressed most of my comments. However, there remains a misunderstanding regarding my most substantial input about the MFSD1 being a uniporter. The authors' data clearly demonstrates that the protein acts as a proton-coupled transporter for acidic substrates. I agree with their conclusion that the protons are carried by these substrates—but that still manifests as a form of coupling to protons: a proton gradient should indeed drive these substrates against their gradients. In contrast to the conclusion of the authors, though (rebuttal p17), no additional “modes’ of transport are required—only that the transporter prefer the protonated forms of these substrates. Thus, the identity of the substrate determines the coupling without needing any additional modifications of the transport cycle. I perseverate on this

issue because I believe that this will be a prominent paper and I believe that it's worth nailing this idea so as not to cause confusion in the field. For a particularly dramatic example of this the authors might look at Basilio et al (JGP 2009 doi: 10.1085/jgp.200810170) where the large-bore Anthrax toxin channel acts as a proton-protein symporter because it strongly disfavors entry of negatively charged residues on its proteinaceous substrate and prefers the protonated forms of those sidechains. Protons are therefore picked up on one side of the membrane and deposited on the other, coupling the movement of substrate protein to that of protons.

We indeed misunderstood the point initially raised by Reviewer 5, and the idea behind the term 'sometimes-coupled transporter'. We fully agree that the protonation in cis and deprotonation in trans of titratable dipeptides indirectly couples their transport to the proton gradient if MFSD1 prefers their protonated form (an aspect deserving future investigation).

To avoid conveying misconceptions about the bioenergetics of MSFD1, we edited a sentence in the Results section and redrafted two paragraphs in the Discussion. We thank the reviewer for the clarification.

Thank you for your time and consideration! We look forward to hearing back from you.

With our best regards,

Markus Damme, PhD

and Christian Löw, PhD

and Bruno Gasnier, PhD

Final Decision Letter:

Dear Dr Damme,

I am pleased to inform you that your manuscript, "MFSD1 with its accessory subunit GLMP functions as a general dipeptide uniporter in lysosomes", has now been accepted for publication in Nature Cell Biology.

You may wish to make your media relations office aware of your accepted publication, in case they consider it appropriate to organize some internal or external publicity. Once your paper has been scheduled you will receive an email confirming the publication details. This is normally 3-4 working days in advance of publication. If you need additional notice of the date and time of publication, please let the production team know when you receive the proof of your article to ensure there is sufficient time to coordinate. Further information on our embargo policies can be found here:

<https://www.nature.com/authors/policies/embargo.html>

Please note that *Nature Cell Biology* is a Transformative Journal (TJ). Authors may publish their research with us through the traditional subscription access route or make their paper immediately open access through payment of an article-processing charge (APC). Authors will not be required to make a final decision about access to their article until it has been accepted. Find out more about Transformative Journals

If you have not already done so, we strongly recommend that you upload the step-by-step protocols used in this manuscript to protocols.io (<https://protocols.io>), an open online resource that allows researchers to share their detailed experimental know-how. All uploaded protocols are made freely available and are assigned DOIs for ease of citation. Protocols and Nature Portfolio journal papers in which they are used can be linked to one another, and this link is clearly and prominently visible in the online versions of both. Authors who performed the specific experiments can act as primary authors for the Protocol as they will be best placed to share the methodology details, but the Corresponding Author of the present research paper should be included as one of the authors. By uploading your Protocols onto protocols.io, you are enabling researchers to more readily reproduce or adapt the methodology you use, as well as increasing the visibility of your protocols and papers. You can also establish a dedicated workspace to collect your lab Protocols. Further information can be found at <https://www.protocols.io/help/publish-articles>.

With kind regards,

Melina Casadio, PhD
Senior Editor, Nature Cell Biology
ORCID ID: <https://orcid.org/0000-0003-2389-2243>